# Safety-Gymnasium: A Unified Safe Reinforcement Learning Benchmark

**Jiaming Ji**[1,*], **Borong Zhang**[1,*], **Jiayi Zhou**[1,*], **Xuehai Pan**[1], **Weidong Huang**[1]

**Ruiyang Sun**[1], **Yiran Geng**[1], **Yifan Zhong**[1,2], **Juntao Dai**[1], **Yaodong Yang** [1,†]

[1] Institute for AI, Peking University
[2] Beijing Institute for General Artificial Intelligence (BIGAI)

{jiamg.ji, borongzh}@gmail.com, gaiejj@outlook.com
yaodong.yang@pku.edu.cn

## Abstract

Artificial intelligence (AI) systems possess significant potential to drive societal progress. However, their deployment often faces obstacles due to substantial safety concerns. Safe reinforcement learning (SafeRL) emerges as a solution to optimize policies while simultaneously adhering to multiple constraints, thereby addressing the challenge of integrating reinforcement learning in safety-critical scenarios. In this paper, we present an environment suite called `Safety-Gymnasium`, which encompasses safety-critical tasks in both single and multi-agent scenarios, accepting vector and vision-only input. Additionally, we offer a library of algorithms named Safe Policy Optimization (`SafePO`), comprising 16 state-of-the-art SafeRL algorithms. This comprehensive library can serve as a validation tool for the research community. By introducing this benchmark, we aim to facilitate the evaluation and comparison of safety performance, thus fostering the development of reinforcement learning for safer, more reliable, and responsible real-world applications. The website of this project can be accessed at `https://sites.google.com/view/safety-gymnasium`.

## 1 Introduction

AI systems possess enormous potential to spur societal progress. However, their deployment is frequently hindered by substantial safety considerations [1; 2; 3; 4]. Distinct from pure reinforcement learning (RL), Safe reinforcement learning (SafeRL) seeks to optimize policies while concurrently adhering to multiple constraints, addressing the challenge of employing RL in scenarios with critical safety implications [5; 6; 7; 8; 9]. This strategy proves particularly pertinent in real-world applications such as autonomous vehicles [10] and healthcare [11], where system failures or unsafe actions can result in grave consequences, such as accidents or harm to individuals. In large language models (LLMs), some studies have also shown that the toxicity of the models can be reduced through SafeRL [12; 13]. Incorporating safety constraints ensures adherence to predefined boundaries and regulatory standards, fostering trust and enabling exploration in environments with high-risk potential. Overall, SafeRL is instrumental in guaranteeing the dependable operation of intelligent systems in intricate and high-stake domains.

---

[*]Equal Contribution. [†]Corresponding author.
Work done when Jiayi Zhou visited Peking University.

37th Conference on Neural Information Processing Systems (NeurIPS 2023) Track on Datasets and Benchmarks.

Simulation environments have become instrumental in fostering the advancement of RL. Eminent examples such as Gym [14], Atari [15], and dm-control [16] underline their importance. These versatile platforms permit researchers to swiftly design and execute varied tasks, thus enabling efficient evaluation of algorithmic effectiveness and intrinsic limitations. However, within the sphere of SafeRL, there is a notable dearth of dedicated simulation environments, which impedes comprehensive exploration of SafeRL. In recent years, there have been strides to address this gap. DeepMind presented AI-Safety-Gridworlds, a suite of RL environments showcasing various safety properties of intelligent agents [17]. Afterward, OpenAI introduced the Safety Gym benchmark suite, a collection of high-dimensional continuous control environments incorporating safety-robot tasks [18]. Over the past two years, several additional environments have been developed by researchers, including safe-control-gym [19], MetaDrive [20], etc.

**Compared to Safety Gym**[1] `Safety-Gymnasium` inherits and expands the settings of some tasks of Safety Gym, aiming to bolster the community's growth further. Compared with Safety Gym, we have made the following major improvements:

- **Refactoring of the physics engine.** Safety Gym utilizes *mujoco-py* to enable Python-based customization of MuJoCo components. However, *mujoco-py* stopped updates and support after 2021. In contrast, `Safety-Gymnasium` supports MuJoCo directly, eliminating the reliance on *mujoco-py*. This facilitates access to the latest MuJoCo features (*e.g.*, rendering speed and accuracy improved, etc.) and lowers the entry barrier, particularly due to *mujoco-py's* dependency on specific GCC versions and more.

- **Extension of Agent and Task Components.** Safety Gym initially supports only three agents and tasks. On this basis, `Safety-Gymnasium` has been further expanded, introducing more diverse agents and task components and expanding safety tasks to cover multi-agent domains. Finally, `Safety-Gymnasium` launched a high-dimensional test component based on Issac-Gym [21], further enriching the benchmark.

- **Enhanced Visual Task Support.** The visual components of Safety Gym are simplistic (consisting of basic geometric shapes), and *mujoco-py* relies on OpenGL for visual rendering, which results in significant virtualization performance loss on headless servers. In contrast, Safety-Gymnasium, built on MuJoCo, achieves rendering speeds on CPU that are twice as fast as the former. Additionally, it offers more comprehensive visual component support.

- **Easy Installation and High Customization.** Safety Gym is cumbersome to install and relies heavily on the underlying software. One of the design motivations of `Safety-Gymnasium` is the ease of use so that everyone can focus on algorithm design. `Safety-Gymnasium` can be easily installed with one simple command pip install safety-gymnasium. While benefiting from the highly integrated framework, `Safety-Gymnasium` only needs 100 lines of code to customize the required environment.

In this work, we introduce `Safety-Gymnasium`, a collection of environments specifically for SafeRL, built upon the Gymnasium [14; 22] and MuJoCo [23]. Enhancing the extant Safety Gym framework [18], we address various concerns and expand the task scope to include vision-only and multi-agent scenarios. Additionally, we released `SafePO`, a single-file style algorithm library containing over 16 state-of-the-art algorithms. Collectively, our contributions are enumerated as follows:

- **Environmental Components.** We provide various safety-oriented tasks under the umbrella of `Safety-Gymnasium`. These tasks encompass single-agent, multi-agent, and vision-based challenges, each with varying constraints. Our environments are categorized into two primary types: Gymnasium-based, featuring agents of escalating complexity for algorithm verification and comparison, and Issac-Gym-based, incorporating sophisticated agents that harness the parallel processing power of Issac-gym's GPU. This empowers researchers to explore SafeRL algorithms in complex scenarios. Further details can be found in Section 4.

- **Algorithm Components.** We offer the `SafePO` algorithm library, which comprises a single-file style housing 16 diverse algorithms. These algorithms encompass both single-agent and multi-agent approaches, along with first-order and second-order variants, as well as Lagrangian-based

---

[1]Again, we have no intention of attacking Safety Gym; the contribution of Safety Gym to the SafeRL community cannot be ignored, and Safety Gym also inspired this work. We hope that through our efforts, `Safety-Gymnasium` can further promote the development of SafeRL and give back to the entire RL community.

and Projection-based methods. Through meticulous decoupling, each algorithm's code resides in an individual file. A more in-depth exploration of `SafePO` is presented in Section 5.

- **Insights and Analysis.** Combining `Safety-Gymnasium` and `SafePO`, we conduct a detailed analysis of existing algorithms. Our analysis encompasses 16 algorithms across 54 distinct environments, covering various scenarios such as single-agent and multi-agent setups with varying constraint complexities. This analysis delves into each algorithm's strengths, constraints, and avenues for enhancement. We provide access to all metadata, fostering community verification and encouraging further research. Further details can be found in Section 6.

## 2 Related Work

**Safety Environments** In RL, agents need to explore environments to learn optimal policies by trial and error. It is currently typical to train RL agents mostly or entirely in simulation, where safety concerns are minimal. However, we anticipate that challenges in simulating the complexities of the real world (*e.g.*, human-AI collaborative control [1; 2]) will cause a shift towards training RL agents directly in the real world, where safety concerns are paramount [20; 24; 25]. OpenAI includes safety requirements in the Safety Gym [18], which is a suite of high-dimensional continuous control environments for measuring research progress on SafeRL. Safe-control-gym [19] allows for constraint specification and disturbance injection onto a robot's inputs, states, and inertial properties through a portable configuration system. DeepMind also presents a suite of RL environments, AI-Safety-Gridworlds [17], illustrating various safety properties of intelligent agents.

**SafeRL Algorithms** CMDPs have been extensively studied for different constraint criteria [26; 27; 28; 29]. With the rise of deep learning, CMDPs are also moving to more high-dimensional continuous control problems. CPO [30] proposes the first general-purpose policy search algorithm for SafeRL with guarantees for near-constraint satisfaction at each iteration. However, CPO's policy updates hinge on Taylor approximations and the inversion of high-dimensional Fisher information matrices. These approximations can occasionally lead to inappropriate policy updates. FOCOPS [31] applies a primal-dual approach to solve the constrained trust region problem directly and subsequently projects the solution back into the parametric policy space. Similarly, CUP [32] offers non-convex implementations through a first-order optimizer, thereby not requiring a strong approximation of the convexity of the objective.

## 3 Preliminaries

### 3.1 Constrained Markov decision process

SafeRL [6; 33] is often formulated as a Constrained Markov decision process (CMDP) [6], which is a tuple $\mathcal{M} = (\mathcal{S}, \mathcal{A}, \mathbb{P}, R, \mathcal{C}, \mu, \gamma)$. Here $\mathcal{S}$ and $\mathcal{A}$ are the state space and action space correspondingly. $\mathbb{P}(s'|s, a)$ is the probability of state transition from $s$ to $s'$ after taking action $a$. $R(s'|s, a)$ denotes the reward obtained by the agent performing action $a$ in state $s$ and transitioning to state $s'$. The set $\mathcal{C} = \left\{ (c_i, b_i) \right\}_{i=1}^{m}$, where $c_i$ are cost functions: $c_i : \mathcal{S} \times \mathcal{A} \to \mathbb{R}$ and the cost thresholds are $b_i, i = 1, \cdots, m$. $\mu(\cdot) : \mathcal{S} \to [0, 1]$ is the initial state distribution and the discount factor $\gamma \in [0, 1)$.

A stationary parameterized policy $\pi_\theta$ is a probability distribution defined on $\mathcal{S} \times \mathcal{A}$, $\pi_\theta(a|s)$ denotes the probability of taking action $a$ in state $s$. We use $\Pi_\theta = \{\pi_\theta : \theta \in \mathbb{R}^p\}$ to denote the set of all stationary policies and $\theta$ is the network parameter needed to be learned. Let $\boldsymbol{P}_{\pi_\theta} \in \mathbb{R}^{|S| \times |S|}$ denotes a state transition probability matrix and the components are: $\boldsymbol{P}_{\pi_\theta}[s, s'] = \mathbb{P}_{\pi_\theta}(s'|s) = \sum_{a \in \mathcal{A}} \pi_\theta(a|s)\mathbb{P}(s'|s, a)$, which denotes one-step state transition probability from $s$ to $s'$ by executing $\pi_\theta$. Finally, we let $d_{\pi_\theta}^{s_0}(s) = (1 - \gamma) \sum_{t=0}^{\infty} \gamma^t \mathbb{P}_{\pi_\theta}(s_t = s|s_0)$ to be the stationary state distribution of the Markov chain starting at $s_0$ induced by policy $\pi_\theta$ and $d_{\pi_\theta}^{\mu}(s) = \mathbb{E}_{s_0 \sim \mu(\cdot)}[d_{\pi_\theta}^{\mu}(s)]$ to be the discounted state visitation distribution on initial distribution $\mu$.

The objective function is defined via the infinite horizon discounted reward function where for a given $\pi_\theta$, we have $J^R(\pi_\theta) = \mathbb{E}[\sum_{t=0}^{\infty} \gamma^t R(s_{t+1}|s_t, a_t)|s_0 \sim \mu, a_t \sim \pi_\theta]$. The cost function is similarly specified via the following infinite horizon discount cost function: $J_i^C(\pi_\theta) = \mathbb{E}[\sum_{t=0}^{\infty} \gamma^t C_i(s_{t+1}|s_t, a_t)|s_0 \sim \mu, a_t \sim \pi_\theta]$.

Then, we define the feasible policy set $\Pi_C$ as : $\Pi_C = \cap_{i=1}^m \{\pi_\theta \in \Pi_\theta \text{ and } J_i^C(\pi_\theta) \le b_i\}$. The goal of CMDP is to search the optimal policy $\pi_\star$: $\pi_\star = \arg\max_{\pi_\theta \in \Pi_C} J^R(\pi_\theta)$.

## 3.2 Constrained Markov Game

Safe multi-agent reinforcement learning is often formulated as a Constrained Markov Game $(\mathcal{N}, \mathcal{S}, \mathcal{A}, \mathbb{P}, \mu, \gamma, R, \boldsymbol{C}, \boldsymbol{b})$. Here, $\mathcal{N} = \{1, \ldots, n\}$ is the set of agents, $\mathcal{S}$ and $\mathcal{A} = \prod_{i=1}^n \mathcal{A}^i$ are the state space and the joint action space (*i.e.*, the product of the agents' action spaces), $\mathbb{P} : \mathcal{S} \times \mathcal{A} \times \mathcal{S} \to \mathbb{R}$ is the probabilistic transition function, $\mu$ is the initial state distribution, $\gamma \in [0, 1)$ is the discount factor, $R : \mathcal{S} \times \mathcal{A} \to \mathbb{R}$ is the joint reward function, $\boldsymbol{C} = \left\{C_j^i\right\}_{1 \le j \le m^i}^{i \in \mathcal{N}}$ is the set of sets of cost functions (every agent $i$ has $m^i$ cost functions) of the form $C_j^i : \mathcal{S} \times \mathcal{A}^i \to \mathbb{R}$, and finally the set of corresponding cost threshold is given by $\boldsymbol{b} = \left\{b_j^i\right\}_{1 \le j \le m^i}^{i \in \mathcal{N}}$. At time step $t$, the agents are in a state $s_t$, and every agent $i$ takes an action $a_t^i$ according to its policy $\pi^i\left(a^i \mid s_t\right)$. Together with other agents' actions, it gives a joint action $\mathbf{a}_t = \left(a_t^1, \ldots, a_t^n\right)$ and the joint policy $\boldsymbol{\pi}(\mathbf{a} \mid \mathbf{s}) = \prod_{i=1}^n \pi^i\left(\mathbf{a}^i \mid \mathbf{s}\right)$. The agents receive the reward $R\left(s_t, \mathbf{a}_t\right)$, meanwhile each agent $i$ pays the costs $C_j^i\left(s_t, a_t^i\right), \forall j = 1, \ldots, m^i$. The environment then transits to a new state $s_{t+1} \sim \mathbb{P}\left(\cdot \mid s_t, \mathbf{a}_t\right)$.

The objective of reward function are $J(\boldsymbol{\pi}) \triangleq \mathbb{E}_{s_0 \sim \rho^0, \mathbf{a}_{0:\infty} \sim \boldsymbol{\pi}, s_{1:\infty} \sim p}[\sum_{t=0}^\infty \gamma^t R(\mathbf{s}_t, \mathbf{a}_t)]$, and costs function are $J_j^i(\boldsymbol{\pi}) \triangleq \mathbb{E}_{s_0 \sim \rho^0, \mathbf{a}_{0:\infty} \sim \boldsymbol{\pi}, s_{1:\infty} \sim p}\left[\sum_{t=0}^\infty \gamma^t C_j^i\left(s_t, a_t^i\right)\right] \le c_j^i, \qquad \forall j = 1, \ldots, m^i$.

We are examining a fully cooperative setting where all agents share a common reward function. Consequently, the goal of safe multi-agent RL is to identify the optimal policy that maximizes the expected total reward while simultaneously ensuring that the safety constraints of each agent are satisfied. Then we define the feasible joint policy set $\boldsymbol{\pi}_C = \cap_{i=1}^n \{\pi_\theta \in \Pi_\theta \text{ and } J_j^i(\boldsymbol{\pi}) \le c_j^i, \forall j = 1, \ldots, m^i\}$. The goal of CMG is to search the optimal policy $\boldsymbol{\pi}_\star = \arg\max_{\pi_\theta \in \Pi_C} J(\pi_\theta)$.

# 4 Safety Environments: `Safety-Gymnasium`

`Safety-Gymnasium` provides a seamless installation process and minimalistic code snippets to basic examples, as shown in Figure 1. Due to the limited space of the paper, we provide a more detailed description (*e.g.*, detailed instructions, the composition of the robot's observation space and action space, dynamic structure, physical parameters, etc.) in Appendix B and Online Documentation[2].

```
"""
Install from PyPI:
  pip install safety-gymnasium
"""
import safety_gymnasium
# Create the safety-task environment
env = safety_gymnasium.make("SafetyPointGoal1-v0", render_mode="human")
# Reset the environment
obs, info = env.reset()
while True:
    # Sample a random action
    act = env.action_space.sample()
    # Step the environment: costs are returned
    obs, reward, cost, terminated, truncated, info = env.step(act)
    if terminated or truncated:
        break
```

Figure 1: Using `Safety-Gymnasium` to create, step, render a specific safety-task environment.

## 4.1 Gynasium-based Learning Environments

In this section, we introduce Gymnasium-based environment components from three aspects: (1) the robots (both single-agent and multi-agent); (2) the tasks that are supported within the environment; (3) the safety constraints that are upheld.

---

[2]Online Documentation: `www.safety-gymnasium.com`

**Supported Robots** As shown in Figure 2, `Safety-Gymnasium` inherits three pre-existing agents from Safety Gym [18], namely Point, Car, and Doggo. By meticulously adjusting the model parameters, we have successfully mitigated the issue of excessive oscillations during the runtime of `Point` and `Car` agents. Building upon this foundation, we have introduced two additional robots: `racecar` [34; 35], and `ant` [23], to enrich the single-agent scenarios. As for multi-agent robots, we have leveraged certain configurations from multi-agent MuJoCo [36], deconstructing the original single-agent structure and enabling multiple agents to control distinct body segments. This design choice has been widely adopted in various research works [37; 38; 39].

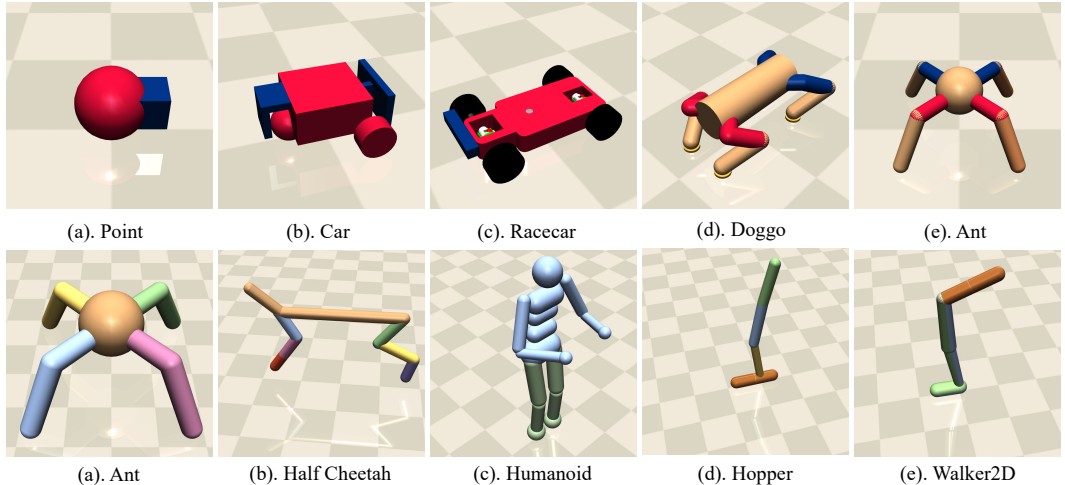

(a). Point    (b). Car    (c). Racecar    (d). Doggo    (e). Ant

(a). Ant    (b). Half Cheetah    (c). Humanoid    (d). Hopper    (e). Walker2D

Figure 2: **Upper:** The Single-Agent Robots of Gymnasium-based Environments. **Lower:** The Multi-Agent Robots of Gymnasium-based Environments.

**Supported Tasks** As shown in Figure 3, the Gymnasium-based learning environments support the following tasks. For a more detailed task specification, please refer to our online documentation[3].

- *Velocity.* The robot aims to facilitate coordinated leg movement of the robot in the forward (right) direction by exerting torques on the hinges.
- *Run.* The robot starts with a random initial direction and a specific initial speed as it embarks on a journey to reach the opposite side of the map.
- *Circle.* The reward is maximized by moving along the green circle and not allowed to enter the outside of the red region, so its optimal path follows the line segments $AD$ and $BC$.
- *Goal.* The robot navigates to multiple goal positions. After successfully reaching a goal, its location is randomly reset while maintaining the overall layout.
- *Push.* The objective is to move a box to a series of goal positions. Like the goal task, a new random goal location is generated after each achievement.
- *Button.* The objective is to activate a series of goal buttons distributed throughout the environment. The agent's goal is to navigate towards and contact the currently highlighted button, known as the goal button.

**Supported Constraints** As shown in Figure 3, the Gymnasium-based environments support the following constraints. For a more detailed task specification, please refer to our online documentation.

- *Velocity-Constraint* involves safety tasks using MuJoCo agents [23]. In these tasks, agents aim for higher reward by moving faster, but they must also adhere to velocity constraints for safety. Specifically, in a two-dimensional plane, the cost is computed as the Euclidean norm of the agent's velocities ($v_x$ and $v_y$).

---

[3]Task Specification Documentation: `https://www.safety-gymnasium.com/en/latest/components_of_environments/tasks.html`

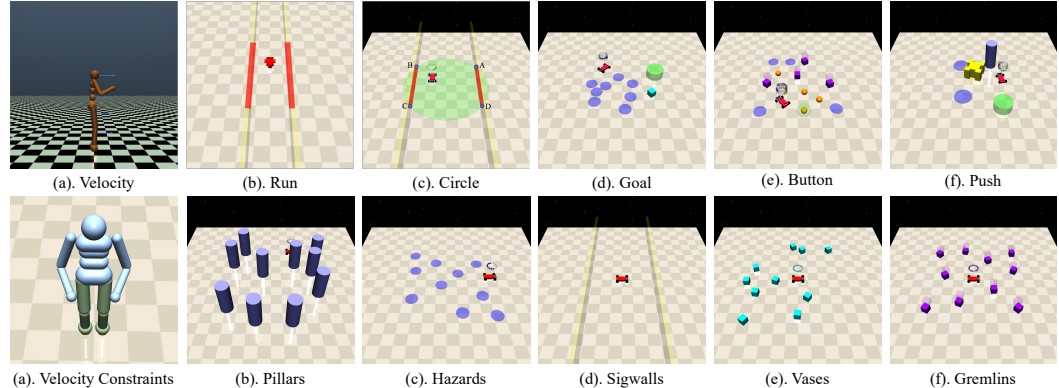

|  |  |  |  |  |  |
|---|---|---|---|---|---|
| (a). Velocity | (b). Run | (c). Circle | (d). Goal | (e). Button | (f). Push |
| (a). Velocity Constraints | (b). Pillars | (c). Hazards | (d). Sigwalls | (e). Vases | (f). Gremlins |

Figure 3: **Upper:** Tasks of Gymnasium-based Environments; **Lower:** Constraints of Gymnasium-based Environments.

- *Pillars* are employed to represent large cylindrical obstacles within the environment. In the general setting, contact with a pillar incurs costs.
- *Hazards* are utilized to model areas within the environment that pose a risk, resulting in costs when an agent enters such areas.
- *Sigwalls* are designed specifically for Circle tasks. Crossing the wall from inside the safe area to the outside incurs costs.
- *Vases* represent static and fragile objects within the environment. Touching or displacing these objects incurs costs for the agent.
- *Gremlins* represent moving objects within the environment that can interact with the agent.

### 4.1.1 Vision-only tasks

Vision-only SafeRL has gained significant attention as a focal point of research, primarily due to its applicability in real-world contexts [40; 41]. While the initial iteration of Safety Gym offered rudimentary visual input support, there is room for enhancing the realism of its environment. To effectively evaluate vision-based SafeRL algorithms, we have devised a more realistic visual environment utilizing MuJoCo. This enhanced environment facilitates the incorporation of both RGB and RGB-D inputs (as shown in Figure 5). An exemplar of this environment is depicted in Figure 4, while comprehensive descriptions are available in Appendix B.5.

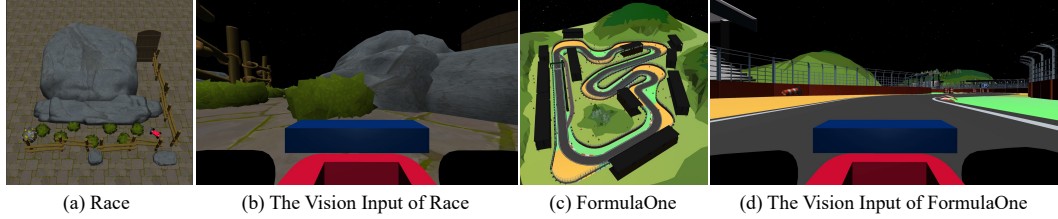

| (a) Race | (b) The Vision Input of Race | (c) FormulaOne | (d) The Vision Input of FormulaOne |
|---|---|---|---|

Figure 4: Vision-only Tasks of Gymnasium-based Environments.

### 4.2 Issac-Gym-based Learning Environments

In this section, we introduce `Safety-DexterousHands`, a collection of environments built upon DexterousHands [42] and the Isaac Gym engine [21]. Leveraging GPU capabilities, Safety-DexterousHands enables large-scale parallel sample collection, significantly accelerating the training process. The environments support both single-agent and multi-agent settings. These environments involve two robotic hands (refer to Figure 6 (a) and (b)). In each episode, a ball randomly descends near the right hand. The right hand needs to grasp and launch the ball toward the left hand, which subsequently catches and deposits it at the target location.

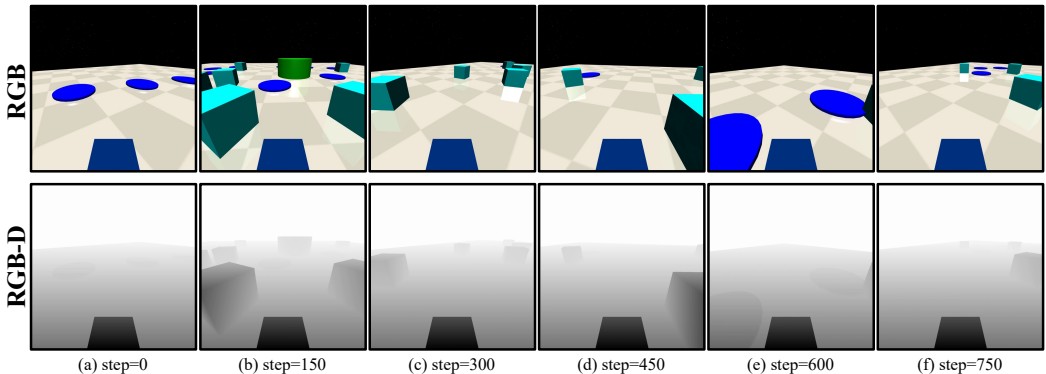

Figure 5: The RGB and RGB-D input of Gymnasium-based Environments.

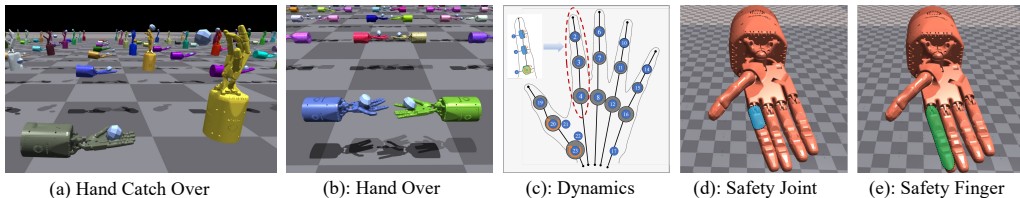

(a) Hand Catch Over     (b): Hand Over     (c): Dynamics     (d): Safety Joint     (e): Safety Finger

Figure 6: Tasks of Safety-DexterousHands.

For timestep $t$, let $x_{b,t}$, $x_{g,t}$ to be the position of the ball and the goal, $d_{p,t}$ to denote the positional distance between the ball and the goal $d_{p,t} = \|x_{b,t} - x_{g,t}\|_2$. Let $d_{a,t}$ denote the angular distance between the object and the goal, and the rotational difference is $d_{r,t} = 2\arcsin\min\{|d_{a,t}|, 1.0\}$. The reward is defined as follows, $r_t = \exp\{-0.2(\alpha d_{p,t} + d_{r,t})\}$, where $\alpha$ is a constant balance of positional and rotational reward.

**Safety Joint** constrains the freedom of joint ④ of the forefinger (refer to Figure 6 (c) and (d)). Without the constraint, joint ④ has freedom of $[-20°, 20°]$. The safety tasks restrict joint ④ within $[-10°, 10°]$. Let ang_4 be the angle of joint ④, and the cost is defined as: $c_t = \mathbb{I}(\texttt{ang\_4} \notin [-10°, 10°])$.

**Safety Finger** constrains the freedom of joints ②, ③ and ④ of forefinger (refer to Figure 6 (c) and (e)). Without the constraint, joints ② and ③ have freedom of $[0°, 90°]$ and joint ④ of $[-20°, 20°]$. The safety tasks restrict joints ②, ③, and ④ within $[22.5°, 67.5°]$, $[22.5°, 67.5°]$, and $[-10°, 10°]$ respectively. Let ang_2, ang_3, ang_4 be the angles of joints ②, ③, ④, and the cost is defined as:

$$c_t = \mathbb{I}(\texttt{ang\_2} \notin [22.5°, 67.5°], \text{ or } \texttt{ang\_3} \notin [22.5°, 67.5°], \text{ or } \texttt{ang\_4} \notin [-10°, 10°]). \quad (1)$$

## 5 Safe Policy Optimization Algorithms: `SafePO`

This section provides a detailed discussion of the design of `SafePO`. Features such as strong performance, extensibility, customization, visualization, and documentation are all presented to demonstrate the advantages and contributions of `SafePO`.

**Correctness** For a benchmark, it is critical to ensure its correctness and reliability. Firstly, each algorithm is implemented strictly according to the original paper (*e.g.*, ensuring consistency with the gradient flow of the original paper, etc.). Secondly, we compare our implementation with those line by line for algorithms with a commonly acknowledged open-source code base to double-check the correctness. Finally, we compare `SafePO` with existing benchmarks (*e.g.*, Safety-Starter-Agents[4] and RL-Safety-Algorithms[5]) and `SafePO` outperforms or achieves comparable performance with other existing implementations, as shown in Table 1.

---

[4]Safety-Starter-Agents: `https://github.com/openai/safety-starter-agents`
[5]RL-Safety-Algorithms: `https://github.com/SvenGronauer/RL-Safety-Algorithms`

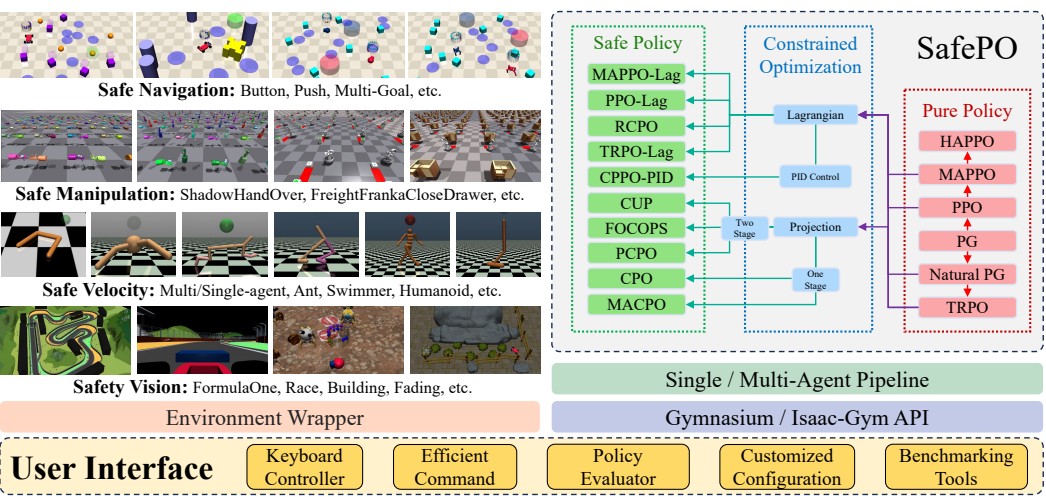

Figure 7: The Architecture of `SafePO`

**Extensibility**  `SafePO` enjoys high extensibility thanks to its architecture (as shown in Figure 7). New algorithms can be integrated into `SafePO` by inheriting from base algorithms and only implementing their unique features. For example, we integrate PPO by inheriting from policy gradient and only adding the clip ratio variable and rewriting the function that computes the loss of policy $\pi$. Similarly, algorithms can be easily added to `SafePO`.

**Logging and Visualization**  Another necessary functionality of `SafePO` is logging and visualization. Supporting both TensorBoard and WandB, we offer code for visualizing more than 40 parameters and intermediate computation results to inspect the training process. Standard parameters and metrics such as KL-divergence, SPS (step per second), and cost variance are visualized universally. Special features of algorithms are also reported, such as the Lagrangian multiplier of Lagrangian-based methods, $g^T H^{-1} g, g^T H^{-1} b, \nu^*$, and $\lambda^*$ of CPO, proportional, integral, and derivative of PID-Lagrangian algorithms, etc. During training, users can inspect the changes of every parameter, collect the log file, and obtain saved checkpoint models. The complete and comprehensive visualization allows easier observation, model selection, and comparison.

**Documentation**  In addition to its code implementation, `SafePO` comes with an extensive documentation[6]. We include detailed guidance on installation and propose solutions to common issues. Moreover, we provide instructions on simple usage and advanced customization of `SafePO`. Official information concerning maintenance, ethical, and responsible use are stated clearly for reference.

Table 1: A comparison between `SafePO` and other implementations. Results are based on 10 evaluation iterations using over 3 seeds under `cost_limit=25.00`. $\bar{J}^R$ stands for normalized reward from PPO's performance, $\bar{J}^C$ signifies normalized cost relative to `cost_limit`, and AvgR/AvgC represents the ratio of the means of both across 10 environments. The $\uparrow$ indicates higher rewards are better, while the $\downarrow$ indicates lower costs (when beyond the threshold of 1.00) are better. *Gray* and *Black* depicts violation and compliance with the `cost_limit`.

| | CPO | | | | | | TRPO-Lag | | | | | | PPO-Lag | | | | FOCOPS | | | |
| | SafePO (Ours) | | Safety Starter Agents | | RL-Safety-Algorithms | | SafePO (Ours) | | Safety Starter Agents | | RL-Safety-Algorithms | | SafePO (Ours) | | Safety Starter Agents | | SafePO (Ours) | | Original Implementation | |
| **Safety Navigation** | $\bar{J}^R\uparrow$ | $\bar{J}^C\downarrow$ | $\bar{J}^R\uparrow$ | $\bar{J}^C\downarrow$ | $\bar{J}^R\uparrow$ | $\bar{J}^C\downarrow$ | $\bar{J}^R\uparrow$ | $\bar{J}^C\downarrow$ | $\bar{J}^R\uparrow$ | $\bar{J}^C\downarrow$ | $\bar{J}^R\uparrow$ | $\bar{J}^C\downarrow$ | $\bar{J}^R\uparrow$ | $\bar{J}^C\downarrow$ | $\bar{J}^R\uparrow$ | $\bar{J}^C\downarrow$ | $\bar{J}^R\uparrow$ | $\bar{J}^C\downarrow$ | $\bar{J}^R\uparrow$ | $\bar{J}^C\downarrow$ |
|---|---|---|---|---|---|---|---|---|---|---|---|---|---|---|---|---|---|---|---|---|
| CarButton1 | 0.08 | 1.75 | 0.34 | 3.65 | -0.06 | 3.30 | -0.04 | 1.08 | 0.02 | 0.78 | -0.05 | 0.63 | 0.01 | 0.47 | 0.02 | 0.67 | 0.04 | 1.21 | 0.53 | 6.02 |
| CarGoal1 | 0.78 | 1.63 | 0.94 | 2.49 | 0.46 | 1.25 | 0.82 | 1.09 | 0.72 | 1.04 | 0.72 | 0.91 | 0.43 | 0.39 | 0.52 | 0.52 | 0.52 | 0.93 | 0.79 | 2.45 |
| PointButton1 | 0.12 | 1.61 | 0.70 | 3.01 | 0.03 | 3.25 | 0.27 | 1.29 | 0.21 | 0.92 | 0.04 | 0.87 | 0.22 | 1.32 | 0.17 | 0.96 | 0.25 | 1.53 | 0.70 | 3.74 |
| PointGoal1 | 0.78 | 1.10 | 0.81 | 1.99 | 0.28 | 2.05 | 0.72 | 0.91 | 0.65 | 0.94 | 0.33 | 0.72 | 0.47 | 1.50 | 0.66 | 0.77 | 0.56 | 1.32 | 0.81 | 1.53 |
| **Safety Velocity** | $\bar{J}^R\uparrow$ | $\bar{J}^C\downarrow$ | $\bar{J}^R\uparrow$ | $\bar{J}^C\downarrow$ | $\bar{J}^R\uparrow$ | $\bar{J}^C\downarrow$ | $\bar{J}^R\uparrow$ | $\bar{J}^C\downarrow$ | $\bar{J}^R\uparrow$ | $\bar{J}^C\downarrow$ | $\bar{J}^R\uparrow$ | $\bar{J}^C\downarrow$ | $\bar{J}^R\uparrow$ | $\bar{J}^C\downarrow$ | $\bar{J}^R\uparrow$ | $\bar{J}^C\downarrow$ | $\bar{J}^R\uparrow$ | $\bar{J}^C\downarrow$ | $\bar{J}^R\uparrow$ | $\bar{J}^C\downarrow$ |
| AntVel | 0.52 | 0.56 | 0.31 | 0.93 | 0.40 | 1.09 | 0.53 | 0.15 | 0.32 | 0.76 | 0.44 | 0.70 | 0.54 | 0.22 | 0.31 | 0.61 | 0.55 | 0.60 | 0.52 | 0.39 |
| HalfCheetahVel | 0.40 | 0.23 | 0.30 | 1.13 | 0.31 | 0.97 | 0.43 | 1.01 | 0.25 | 0.79 | 0.43 | 0.67 | 0.44 | 0.04 | 0.30 | 0.93 | 0.42 | 0.12 | 0.44 | 0.04 |
| HopperVel | 0.73 | 0.48 | 0.35 | 0.93 | 0.26 | 0.68 | 0.59 | 0.71 | 0.41 | 1.11 | 0.24 | 0.57 | 0.58 | 0.89 | 0.29 | 1.20 | 0.66 | 0.30 | 0.74 | 0.53 |
| HumanoidVel | 0.71 | 0.01 | 0.05 | 0.19 | 0.36 | 0.83 | 0.72 | 2.38 | 0.05 | 0.01 | 0.71 | 0.79 | 0.72 | 0.76 | 0.07 | 0.09 | 0.71 | 0.93 | 0.73 | 0.43 |
| SwimmerVel | 0.51 | 0.82 | 0.38 | 1.11 | 0.41 | 0.82 | 0.66 | 0.84 | 0.43 | 1.67 | 0.41 | 1.02 | 0.57 | 1.11 | 0.38 | 1.18 | 0.47 | 1.30 | 0.68 | 0.71 |
| Walker2dVel | 0.39 | 0.81 | 0.44 | 1.85 | 0.05 | 0.67 | 0.51 | 0.77 | 0.46 | 0.67 | 0.51 | 1.34 | 0.44 | 0.20 | 0.47 | 0.81 | 0.50 | 0.68 | 0.48 | 0.74 |
| **AvgR/AvgC** | **0.56** | | 0.27 | | 0.17 | | **0.51** | | 0.40 | | 0.46 | | **0.64** | | 0.41 | | **0.52** | | 0.39 | |

---

[6] `SafePO`'s Documentation: https://safe-policy-optimization.readthedocs.io

# 6 Experiments and Analysis

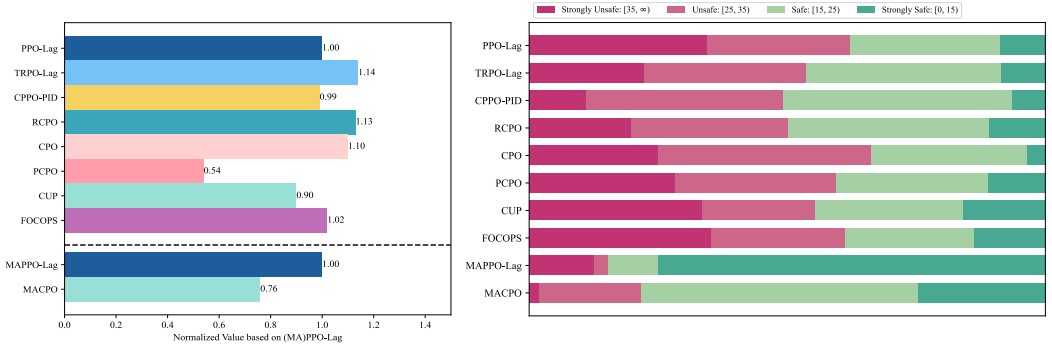

(a) Average Episodic Reward of Algorithms    (b) Bar Chart Categorizing Algorithms into Four Classes Based on Average Episodic Cost

Figure 8: A bar chart analyzing the performance of different algorithms. The left graph compares episodic reward with PPO-Lag [18] (or MAPPO-Lag [39] for multi-agent). The right graph shows episodic costs proportionally under varying constraints. Single-agent data is from 40 navigation and 6 velocity tasks, and multi-agent data is from all 8 velocity tasks in `Safety-Gymnasium`.

Table 2: The performance of single-agent algorithms. $\bar{J}^R$ stands for normalized reward from PPO's performance, and $\bar{J}^C$ signifies normalized cost relative to `cost_limit`. The ↑ indicates higher rewards are better, while the ↓ indicates lower costs (when beyond the threshold of 1.00) are better. *Gray* and *Black* depicts breach and compliance with the `cost_limit`, while *Green* represents the optimal policy, maximizing reward within safety constraints.

| | PPO | | PPO-Lag | | TRPO-Lag | | CPPO-PID | | RCPO | | CPO | | PCPO | | CUP | | FOCOPS | |
|---|---|---|---|---|---|---|---|---|---|---|---|---|---|---|---|---|---|---|---|
| **Safety Navigation** | $\bar{J}^R\uparrow$ | $\bar{J}^C\downarrow$ | $\bar{J}^R\uparrow$ | $\bar{J}^C\downarrow$ | $\bar{J}^R\uparrow$ | $\bar{J}^C\downarrow$ | $\bar{J}^R\uparrow$ | $\bar{J}^C\downarrow$ | $\bar{J}^R\uparrow$ | $\bar{J}^C\downarrow$ | $\bar{J}^R\uparrow$ | $\bar{J}^C\downarrow$ | $\bar{J}^R\uparrow$ | $\bar{J}^C\downarrow$ | $\bar{J}^R\uparrow$ | $\bar{J}^C\downarrow$ | $\bar{J}^R\uparrow$ | $\bar{J}^C\downarrow$ |
| AntButton1 | 1.00 | 4.42 | 0.09 | 0.86 | 0.23 | 1.95 | 0.10 | 0.70 | 0.16 | 2.07 | 0.12 | 4.01 | 0.03 | 1.01 | 0.03 | 0.17 | 0.01 | 0.46 |
| AntCircle1 | 1.00 | 16.81 | 0.79 | 2.56 | 0.65 | 1.05 | 0.69 | 1.90 | 0.63 | 1.04 | 0.47 | 1.07 | 0.28 | 1.87 | 0.60 | 0.82 | 0.02 | 1.22 |
| AntGoal1 | 1.00 | 1.81 | 0.26 | 0.94 | 0.25 | 0.74 | 0.47 | 1.94 | 0.29 | 0.78 | 0.19 | 0.55 | 0.09 | 0.42 | 0.34 | 1.33 | 0.09 | 0.67 |
| AntPush1 | 1.00 | 1.90 | 0.13 | 0.00 | 0.30 | 0.00 | 0.13 | 0.00 | 0.33 | 0.00 | 0.17 | 0.00 | 0.07 | 0.00 | 0.20 | 0.00 | -0.30 | 0.03 |
| CarButton1 | 1.00 | 16.09 | 0.01 | 0.47 | -0.04 | 1.08 | -0.10 | 0.40 | -0.19 | 1.73 | 0.08 | 1.75 | 0.02 | 1.90 | 0.04 | 5.50 | 0.04 | 1.21 |
| CarCircle1 | 1.00 | 8.42 | 0.81 | 0.82 | 1.69 | 2.77 | 1.61 | 1.79 | 1.70 | 3.11 | 1.67 | 3.13 | 1.41 | 1.99 | 0.76 | 1.04 | 0.84 | 1.12 |
| CarGoal1 | 1.00 | 2.38 | 0.43 | 0.39 | 0.82 | 1.09 | 0.03 | 2.47 | 0.55 | 0.86 | 0.78 | 1.63 | 0.61 | 1.42 | 0.19 | 0.63 | 0.52 | 0.93 |
| CarPush1 | 1.00 | 7.16 | 0.46 | 0.78 | 1.38 | 0.70 | 0.03 | 0.47 | 1.11 | 1.42 | 0.83 | 1.14 | 0.64 | 2.36 | 0.32 | 0.95 | 0.29 | 0.36 |
| DoggoButton1 | 1.00 | 7.57 | 0.01 | 0.03 | 0.00 | 1.27 | 0.01 | 0.07 | 0.01 | 0.09 | 0.00 | 0.15 | 0.00 | 0.25 | 0.02 | 0.45 | 0.06 | 3.68 |
| DoggoCircle1 | 1.00 | 33.14 | 0.77 | 0.46 | 0.67 | 1.37 | 0.82 | 2.16 | 0.55 | 1.32 | 0.66 | 1.22 | 0.31 | 0.55 | 0.80 | 2.04 | 0.73 | 4.49 |
| DoggoGoal1 | 1.00 | 2.28 | 0.05 | 0.00 | 0.18 | 0.69 | 0.00 | 0.00 | 0.16 | 2.08 | 0.30 | 0.50 | 0.00 | 0.00 | 0.00 | 0.90 | 0.04 | 1.27 |
| DoggoPush1 | 1.00 | 1.31 | 0.09 | 0.00 | 0.53 | 0.78 | 0.32 | 0.44 | 0.54 | 1.55 | 0.46 | 0.00 | 0.36 | 0.00 | 0.30 | 0.68 | 0.64 | 3.40 |
| PointButton1 | 1.00 | 6.06 | 0.22 | 1.32 | 0.27 | 1.29 | 0.00 | 0.84 | 0.12 | 1.13 | 0.12 | 1.61 | 0.08 | 2.19 | 0.18 | 1.26 | 0.25 | 1.53 |
| PointCircle1 | 1.00 | 8.10 | 0.86 | 0.93 | 1.67 | 1.35 | 1.72 | 2.09 | 1.66 | 1.42 | 1.69 | 1.74 | 1.33 | 2.26 | 0.82 | 0.62 | 0.84 | 0.89 |
| PointGoal1 | 1.00 | 1.93 | 0.47 | 1.50 | 0.72 | 0.91 | 0.31 | 1.05 | 0.53 | 0.99 | 0.78 | 1.10 | 0.71 | 0.82 | 0.46 | 0.73 | 0.56 | 1.32 |
| PointPush1 | 1.00 | 2.31 | 0.98 | 1.33 | 0.85 | 1.00 | 0.35 | 0.35 | 5.30 | 0.94 | 2.22 | 0.80 | 1.72 | 1.25 | 2.32 | 0.80 | 1.13 | 2.51 |
| RacecarButton1 | 1.00 | 13.73 | -0.01 | 1.94 | -0.02 | 1.77 | -0.16 | 2.06 | -0.07 | 1.19 | 0.00 | 2.44 | 0.02 | 1.82 | 0.00 | 5.23 | -0.10 | 3.37 |
| RacecarCircle1 | 1.00 | 15.87 | 0.83 | 1.90 | 0.80 | 2.18 | 0.58 | 1.33 | 0.83 | 2.07 | 0.79 | 0.81 | 0.22 | 2.87 | 0.74 | 3.53 | 0.77 | 2.11 |
| RacecarGoal1 | 1.00 | 4.26 | 0.26 | 0.51 | 1.19 | 0.77 | -0.04 | 1.07 | 0.88 | 0.83 | 1.18 | 2.58 | 0.33 | 0.24 | 0.13 | 1.22 | 0.31 | 0.62 |
| RacecarPush1 | 1.00 | 2.34 | -0.40 | 0.00 | 0.74 | 1.79 | -0.84 | 2.87 | 0.58 | 1.92 | 0.94 | 0.13 | -0.16 | 0.18 | -0.06 | 3.79 | 0.30 | 2.04 |
| **Safety Velocity** | $\bar{J}^R\uparrow$ | $\bar{J}^C\downarrow$ | $\bar{J}^R\uparrow$ | $\bar{J}^C\downarrow$ | $\bar{J}^R\uparrow$ | $\bar{J}^C\downarrow$ | $\bar{J}^R\uparrow$ | $\bar{J}^C\downarrow$ | $\bar{J}^R\uparrow$ | $\bar{J}^C\downarrow$ | $\bar{J}^R\uparrow$ | $\bar{J}^C\downarrow$ | $\bar{J}^R\uparrow$ | $\bar{J}^C\downarrow$ | $\bar{J}^R\uparrow$ | $\bar{J}^C\downarrow$ | $\bar{J}^R\uparrow$ | $\bar{J}^C\downarrow$ |
| AntVel | 1.00 | 38.33 | 0.54 | 0.22 | 0.53 | 0.15 | 0.51 | 0.41 | 0.52 | 0.56 | 0.52 | 0.56 | 0.38 | 0.41 | 0.55 | 0.94 | 0.55 | 0.60 |
| HalfCheetahVel | 1.00 | 36.77 | 0.44 | 0.00 | 0.43 | 1.01 | 0.48 | 0.04 | 0.36 | 0.56 | 0.40 | 0.23 | 0.25 | 0.63 | 0.40 | 0.17 | 0.42 | 0.12 |
| HopperVel | 1.00 | 22.00 | 0.58 | 0.89 | 0.59 | 0.71 | 0.73 | 0.44 | 0.58 | 0.59 | 0.58 | 0.48 | 0.65 | 0.51 | 0.73 | 0.21 | 0.66 | 0.30 |
| HumanoidVel | 1.00 | 38.42 | 0.72 | 0.76 | 0.72 | 2.38 | 0.73 | 0.00 | 0.68 | 0.82 | 0.71 | 0.01 | 0.64 | 0.01 | 0.68 | 0.80 | 0.71 | 0.93 |
| SwimmerVel | 1.00 | 6.61 | 0.57 | 1.11 | 0.66 | 0.84 | 0.91 | 0.92 | 0.54 | 0.90 | 0.51 | 0.82 | 0.50 | 0.69 | 0.59 | 0.96 | 0.47 | 1.30 |
| Walker2dVel | 1.00 | 36.11 | 0.44 | 0.20 | 0.51 | 0.77 | 0.27 | 0.36 | 0.49 | 0.15 | 0.39 | 0.81 | 0.27 | 0.71 | 0.44 | 0.18 | 0.50 | 0.16 |

**Reward and Cost.** Episodic reward and cost exhibit a trade-off relationship. Unconstrained algorithms aim to maximize reward through risky behaviors. HAPPO [37] achieves higher rewards compared to MAPPO [38] across 8 velocity-based tasks, accompanied by a simultaneous increase in average costs. SafeRL algorithms tend to maximize reward while adhering to constraints. As depicted in Table 2, in the velocity task, compared to PPO [43], PPO-Lag [18] achieves a reduction of 98% in cost while only experiencing a decrease of 45% in reward.

**Randomness and Oscillation.** The randomness of tasks is correlated with the oscillation of algorithms' performance. All SafeRL algorithms achieve average episodic costs within the `cost_limit` for velocity tasks. The divergence in episodic rewards between algorithms is negligible, and the distribution of optimal policies is tightly clustered. However, pronounced oscillations are present in navigation tasks characterized by high stochasticity. Out of the 20 navigation tasks examined,

optimal policies are spread out more, leading to observable differences in algorithm performance across various tasks.

**Lagrangian vs. Projection.** In contrast to projection-based methods, the Lagrangian-based methods tend to display more oscillation. A notable disparity becomes apparent upon examining the oscillatory patterns in the episodic cost around the designated safety constraints during training, as presented in Figure 8(b). Both CPO [30] and PPO-Lag [18] demonstrate oscillations; however, those exhibited by PPO-Lag are more conspicuous. This discrepancy is manifested in a higher proportion of instances clas-

Table 3: The normalized performance of `SafePO`'s multi-agent algorithms on `Safety-Gymnasium`.

| Safety Velocity | MAPPO | | HAPPO | | MAPPO-Lag | | MACPO | |
|---|---|---|---|---|---|---|---|---|
| | $\bar{J}^R\uparrow$ | $\bar{J}^C\downarrow$ | $\bar{J}^R\uparrow$ | $\bar{J}^C\downarrow$ | $\bar{J}^R\uparrow$ | $\bar{J}^C\downarrow$ | $\bar{J}^R\uparrow$ | $\bar{J}^C\downarrow$ |
| 2x4AntVel | 1.00 | 35.76 | 1.26 | 39.12 | 0.57 | 0.00 | 0.51 | 0.14 |
| 4x2AntVel | 1.00 | 38.01 | 1.07 | 34.34 | 0.50 | 0.00 | 0.50 | 0.01 |
| 2x3HalfCheetahVel | 1.00 | 39.02 | 1.11 | 37.70 | 0.35 | 0.01 | 0.49 | 1.28 |
| 6x1HalfCheetahVel | 1.00 | 39.23 | 1.09 | 37.74 | 0.28 | 0.02 | 0.36 | 0.37 |
| 3x1HopperVel | 1.00 | 22.58 | 1.04 | 22.05 | 0.47 | 0.00 | 0.22 | 1.03 |
| 9|8HumanoidVel | 1.00 | 6.34 | 2.79 | 17.18 | 0.54 | 0.84 | 0.53 | 1.30 |
| 2x3Walker2dVel | 1.00 | 22.99 | 1.55 | 33.67 | 0.60 | 0.01 | 0.27 | 1.21 |

sified as *Strongly Unsafe* and *Strongly Safe* for PPO-Lag, while CPO maintains a more centered distribution. Nevertheless, an excessively cautious policy has the potential to undermine performance. In contrast, the projection-based method PCPO [3] exhibits lower average costs and rewards in navigation and velocity tasks than CPO. This distinction is further accentuated when examining the contrast between MACPO and MAPPO-Lag.

**Lagrangian vs. PID-Lagrangian.** Incorporating a PID controller within the Lagrangian-based framework proves to be effective in mitigating inherent oscillations. As shown in Figure 8, CPPO-PID [44] displays episodic rewards during training that closely resemble those of PPO-Lag. However, CPPO-PID demonstrates a reduced frequency of instances entering the *Strongly Unsafe* region, resulting in a more significant proportion of *Safe* states and improved safety performance.

# 7 Limitations and Future Works

Ensuring safety remains a paramount concern. Across various tasks, safety concerns can be transformed into corresponding constraints. However, a limitation of this study is its inability to encompass all forms of constraints. For instance, safety constraints related to human-centric considerations are paramount in human-AI collaboration, yet these considerations have not been fully integrated within the scope of this study. This work focuses on safety tasks within a simulated environment and introduces an extensive testing component. However, the transferability of the results to complex real-world safety-critical applications may be limited. A promising work for the future involves transferring policy refined within the `Safety-Gymnasium` to physical robotic platforms, which holds profound implications.

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

# A    Details of Experimental Results

## A.1    Hyperparameters Analysis

This section presents the disclosure of `SafePO` hyperparameters settings and their rationales. We employed the Generalized Advantage Estimation (`GAE`)[45] method to estimate the values of rewards and cost advantages and used Adam[46] for learning the neural network parameters.

**Single-agent Algorithm Settings.** The models employed in the single-agent algorithms were 3-layer MLPs with `Tanh` activation functions and hidden layer sizes of [64, 64], for more intricate navigation agents Ant and Doggo, hidden layers of [256, 256] were employed.

**Multi-agent Algorithms Settings.** The models employed in the multi-agent algorithms were 3-layer MLPs with `ReLU` activation functions and hidden layer sizes of [128, 128].

Table 4: Hyperparameters of `SafePO` algorithms in `Safety-Gymnasium` tasks. Second-order algorithms set the parameters to the actor model directly, instead of iterative gradient descent, so the *Actor Learning Rate* of them are marked Gray.

| PG/PPO/PPO-Lag | Value |
| --- | --- |
| Discount Factor $\gamma$ | 0.99 |
| Target KL | 0.02 |
| GAE $\lambda$ | 0.95 |
| Number of SGD Iterations | 40 |
| Training Batch Size | 20000 |
| Actor Learning Rate | 0.0003 |
| Critic Learning Rate | 0.0003 |
| Cost Limit | 25.00 |
| Clip Coefficient | 0.20 |
| Lagrangian Initial Value | 0.001 |
| Lagrangian Learning Rate | 0.035 |
| Lagrangian Optimizer | Adam |

| TRPO/TRPO-Lag | Value |
| --- | --- |
| Discount Factor $\gamma$ | 0.99 |
| Target KL | 0.01 |
| GAE $\lambda$ | 0.95 |
| Number of SGD Iterations | 10 |
| Training Batch Size | 20000 |
| Actor Learning Rate | None |
| Critic Learning Rate | 0.001 |
| Cost Limit | 25.00 |
| Conjugate Gradient Iterations | 15 |
| Lagrangian Initial Value | 0.001 |
| Lagrangian Learning Rate | 0.035 |
| Lagrangian Optimizer | Adam |

| CPPO-PID | Value |
| --- | --- |
| Discount Factor $\gamma$ | 0.99 |
| Target KL | 0.02 |
| GAE $\lambda$ | 0.95 |
| Number of SGD Iterations | 40 |
| Training Batch Size | 20000 |
| Actor Learning Rate | 0.0003 |
| Critic Learning Rate | 0.0003 |
| Cost Limit | 25.00 |
| Clip Coefficient | 0.20 |
| PID Controller Kp | 0.10 |
| PID Controller Ki | 0.01 |
| PID Controller Kd | 0.01 |

| NPG/RCPO | Value |
| --- | --- |
| Discount Factor $\gamma$ | 0.99 |
| Target KL | 0.01 |
| GAE $\lambda$ | 0.95 |
| Number of SGD Iterations | 10 |
| Training Batch Size | 20000 |
| Actor Learning Rate | None |
| Critic Learning Rate | 0.001 |
| Cost Limit | 25.00 |
| Conjugate Gradient Iterations | 15 |
| Lagrangian Initial Value | 0.001 |
| Lagrangian Learning Rate | 0.035 |
| Lagrangian Optimizer | Adam |

| HAPPO/MAPPO/MAPPO-Lag | Value |
| --- | --- |
| Discount Factor $\gamma$ | 0.99 |
| Target KL | 0.016 |
| GAE $\lambda$ | 0.95 |
| Number of SGD Iterations | 5 |
| Training Batch Size | 10000 |
| Actor Learning Rate | 0.0005 |
| Critic Learning Rate | 0.0005 |
| Cost Limit | 25.00 |
| Clip Coefficient | 0.20 |
| Lagrangian Initial Value | 0.00001 |
| Lagrangian Learning Rate | 0.78 |
| Lagrangian Optimizer | SGD |

| CPO | Value |
| --- | --- |
| Discount Factor $\gamma$ | 0.99 |
| Target KL | 0.01 |
| GAE $\lambda$ | 0.95 |
| Number of SGD Iterations | 10 |
| Training Batch Size | 20000 |
| Actor Learning Rate | None |
| Critic Learning Rate | 0.001 |
| Cost Limit | 25.00 |
| Conjugate Gradient Iterations | 15 |
| CPO Searching Steps | 15 |
| Step Fraction | 0.80 |

| PCPO | Value |
| --- | --- |
| Discount Factor $\gamma$ | 0.99 |
| Target KL | 0.01 |
| GAE $\lambda$ | 0.95 |
| Number of SGD Iterations | 10 |
| Training Batch Size | 20000 |
| Actor Learning Rate | None |
| Critic Learning Rate | 0.001 |
| Cost Limit | 25.00 |
| Conjugate Gradient Iterations | 15 |
| PCPO Searching Steps | 200 |
| Step Fraction | 0.80 |

| CUP | Value |
| --- | --- |
| Discount Factor $\gamma$ | 0.99 |
| Target KL | 0.02 |
| GAE $\lambda$ | 0.95 |
| Number of SGD Iteration | 40 |
| Training Batch Size | 20000 |
| Actor Learning Rate | 0.0003 |
| Critic Learning Rate | 0.0003 |
| Cost Limit | 25.00 |
| Clip Coefficient | 0.20 |
| CUP $\lambda$ | 0.95 |
| CUP $\nu$ | 2.00 |

| FOCOPS | Value |
| --- | --- |
| Discount Factor $\gamma$ | 0.99 |
| Target KL | 0.02 |
| GAE $\lambda$ | 0.95 |
| Number of SGD Iteration | 40 |
| Training Batch Size | 20000 |
| Actor Learning Rate | 0.0003 |
| Critic Learning Rate | 0.0003 |
| Cost Limit | 25.00 |
| Clip Coefficient | 0.20 |
| FOCOPS $\lambda$ | 1.50 |
| FOCOPS $\nu$ | 2.00 |

| MACPO | Value |
| --- | --- |
| Discount Factor $\gamma$ | 0.99 |
| Target KL | 0.01 |
| GAE $\lambda$ | 0.95 |
| Number of SGD Iteration | 15 |
| Training Batch Size | 10000 |
| Actor Learning Rate | None |
| Critic Learning Rate | 0.0005 |
| Cost Limit | 25.00 |
| Conjugate Gradient Iterations | 10 |
| MACPO Searching Steps | 10 |
| Step Fraction | 0.50 |

**Lagrangian Multiplier Settings.** Lagrangian-based methods are sensitive to hyperparameters. We present the following detailed description of the settings for both the naive and the PID-controlled Lagrangian multiplier.

- Lagrangian Initial Value: The initial value of the Lagrangian multiplier. It impacts the early-stage performance of the Lagrangian-based methods. A higher initial value promotes safer exploration but may impede task completion. Conversely, a lower initial value delays the agent's exploration of safe policies.

- Lagrangian Learning Rate: The learning rate of the Lagrangian multiplier. A high learning rate induces excessive oscillations, impedes convergence speed, and hinders the algorithm's ability to attain the desired solution. Conversely, a low learning rate slows down convergence and adversely affects training.

- PID Controller Kp: The PID controller's proportional gain determines the output's response to changes in the episodic costs. If `pid_kp` is too large, the Lagrangian multiplier oscillates, and performance deteriorates. If `pid_kp` is too small, the Lagrangian multiplier updates slowly, also impacting performance negatively.

- PID Controller Kd: The PID controller's derivative gain governs the output's response to changes in the episodic costs. If `pid_kd` is too large, the Lagrangian multiplier becomes excessively sensitive to noise or changes in the episodic costs, leading to instability or oscillations. If `pid_kd` is too small, the Lagrangian multiplier may not respond quickly or accurately enough to changes in the episodic costs.

- PID Controller Ki: The PID controller's integral gain determines the controller's ability to eliminate the steady-state error by integrating the episodic costs over time. If `pid_ki` is too large, the Lagrangian multiplier may become overly responsive to previous errors, adversely affecting performance.

## A.2 Performance Table of `Safety-Gymnasium`

Table 5: The performance of `SafePO` algorithms on `Safety-Gymnasium`. All experimental outcomes were derived from 10 assessment iterations encompassing multiple random seeds and under the experimental setting of `cost_limit=25.00`. The ↑ indicates higher rewards are better, while the ↓ indicates lower costs (when beyond the threshold of 25.00) are better. *Gray* and *Black* depicts breach and compliance with the `cost_limit`, while *Green* represents the optimal policy, maximizing reward within safety constraints.

(a) The performance of `SafePO` single-agent algorithms on `Safety-Gymnasium`.

| Safety Navigation | PPO $J^R$ | PPO $J^C$ | PPO-Lag $J^R$ | PPO-Lag $J^C$ | CPPO-PID $J^R$ | CPPO-PID $J^C$ | TRPO-Lag $J^R$ | TRPO-Lag $J^C$ | RCPO $J^R$ | RCPO $J^C$ | CPO $J^R$ | CPO $J^C$ | PCPO $J^R$ | PCPO $J^C$ | CUP $J^R$ | CUP $J^C$ | FOCOPS $J^R$ | FOCOPS $J^C$ |
|---|---|---|---|---|---|---|---|---|---|---|---|---|---|---|---|---|---|---|
| AntButton1 | 38.70 | 110.60 | 3.63 | 21.60 | 4.06 | 17.45 | 8.93 | 48.70 | 6.16 | 51.70 | 4.50 | 100.30 | 1.27 | 25.35 | 1.26 | 4.25 | 0.22 | 11.55 |
| AntButton2 | 36.15 | 95.00 | 2.72 | 14.85 | 2.86 | 28.70 | 8.66 | 49.45 | 8.66 | 37.40 | 4.63 | 35.60 | 3.04 | 27.50 | 1.60 | 32.90 | -0.04 | 6.80 |
| AntCircle1 | 94.04 | 420.30 | 74.31 | 63.90 | 64.90 | 47.50 | 61.02 | 26.30 | 59.42 | 26.00 | 43.74 | 26.80 | 26.47 | 46.85 | 56.77 | 20.50 | 2.27 | 30.50 |
| AntCircle2 | 84.80 | 736.00 | 65.72 | 22.45 | 64.49 | 39.85 | 66.75 | 22.75 | 63.04 | 19.00 | 53.74 | 43.90 | 16.41 | 15.85 | 42.65 | 10.80 | 4.78 | 66.30 |
| AntGoal1 | 82.02 | 45.30 | 21.33 | 23.60 | 38.79 | 48.55 | 20.64 | 18.50 | 23.38 | 19.60 | 15.35 | 13.80 | 7.31 | 10.50 | 27.98 | 33.25 | 6.99 | 16.75 |
| AntGoal2 | 86.14 | 165.60 | 1.01 | 0.00 | 0.10 | 0.00 | 4.44 | 13.45 | 6.27 | 54.00 | 0.85 | 4.60 | 0.02 | 0.00 | 0.76 | 1.15 | 0.08 | 1.15 |
| AntPush1 | 0.46 | 47.55 | 0.06 | 0.00 | 0.06 | 0.00 | 0.14 | 0.00 | 0.15 | 0.00 | 0.08 | 0.00 | 0.03 | 0.00 | 0.09 | 0.00 | -0.14 | 0.70 |
| AntPush2 | 0.77 | 139.20 | 0.01 | 0.02 | 0.02 | 0.00 | 0.01 | 0.00 | 0.10 | 0.00 | 0.05 | 0.00 | 0.02 | 0.00 | 0.02 | 0.10 | 0.07 | 0.20 |
| CarButton1 | 15.74 | 398.81 | 0.11 | 11.87 | -1.70 | 10.03 | -0.66 | 26.90 | -3.16 | 43.20 | 1.30 | 43.73 | 0.27 | 47.60 | 0.68 | 137.47 | 0.60 | 30.23 |
| CarButton2 | 19.32 | 333.82 | 1.23 | 46.14 | -1.83 | 26.55 | -2.23 | 17.98 | -0.02 | 27.09 | -0.10 | 36.97 | 0.49 | 38.54 | 0.80 | 154.50 | 0.07 | 53.49 |
| CarCircle1 | 21.92 | 208.73 | 17.91 | 20.62 | 35.71 | 44.87 | 37.42 | 69.30 | 37.78 | 77.77 | 37.10 | 78.23 | 31.37 | 49.80 | 16.89 | 25.88 | 18.63 | 27.98 |
| CarCircle2 | 19.75 | 401.83 | 16.27 | 29.88 | 30.80 | 40.37 | 33.23 | 54.20 | 33.74 | 42.17 | 33.42 | 78.97 | 27.93 | 70.40 | 14.74 | 15.46 | 15.60 | 31.20 |
| CarGoal1 | 32.57 | 58.91 | 14.57 | 9.84 | 1.00 | 61.71 | 27.49 | 27.28 | 18.49 | 21.45 | 26.23 | 40.71 | 20.64 | 35.41 | 6.38 | 15.67 | 17.58 | 23.22 |
| CarGoal2 | 31.59 | 215.74 | 0.59 | 16.81 | 0.12 | 23.09 | 3.27 | 47.18 | 2.61 | 25.45 | 3.55 | 32.63 | 1.83 | 57.82 | 2.45 | 125.80 | 3.28 | 23.01 |
| CarPush1 | 1.13 | 181.04 | 0.49 | 19.60 | 0.03 | 11.83 | 1.48 | 17.60 | 1.19 | 35.50 | 0.89 | 28.50 | 0.68 | 59.03 | 0.34 | 23.86 | 0.31 | 8.96 |
| CarPush2 | 1.03 | 46.87 | 0.54 | 43.32 | 0.57 | 37.37 | 0.43 | 38.63 | 0.12 | 27.57 | 0.15 | 19.03 | 0.29 | 60.10 | 0.41 | 82.20 | -0.28 | 40.42 |
| DoggoButton1 | 27.23 | 189.30 | 0.33 | 0.80 | 0.22 | 1.67 | 0.01 | 31.75 | 0.30 | 2.25 | 0.03 | 3.70 | -0.06 | 6.20 | 0.67 | 11.17 | 1.52 | 91.90 |
| DoggoButton2 | 29.84 | 194.60 | 0.10 | 1.00 | 0.16 | 2.70 | -0.05 | 17.05 | 0.07 | 0.00 | 0.03 | 1.40 | 0.01 | 8.01 | 0.35 | 43.37 | 0.22 | 2.10 |
| DoggoCircle2 | 41.90 | 442.70 | 30.13 | 14.20 | 34.82 | 62.03 | 21.97 | 46.75 | 20.68 | 37.35 | 20.41 | 32.55 | 15.41 | 24.05 | 33.08 | 58.33 | 28.91 | 122.80 |
| DoggoCircle1 | 41.61 | 828.50 | 32.03 | 11.50 | 34.26 | 53.93 | 27.86 | 34.20 | 22.93 | 32.90 | 27.65 | 30.55 | 12.94 | 13.70 | 33.45 | 50.97 | 30.29 | 112.20 |
| DoggoGoal1 | 43.10 | 57.10 | 2.00 | 0.00 | 0.13 | 0.00 | 7.88 | 17.25 | 6.82 | 52.05 | 12.73 | 12.40 | 0.14 | 0.00 | 0.16 | 22.47 | 1.88 | 31.80 |
| DoggoGoal2 | 42.04 | 123.30 | 0.06 | 0.00 | 0.09 | 0.00 | 0.02 | 0.00 | 0.06 | 0.00 | 0.03 | 0.00 | 0.06 | 0.00 | 0.28 | 3.33 | 0.08 | 0.00 |
| DoggoPush2 | 0.82 | 32.70 | -0.02 | 0.00 | 0.08 | 0.00 | 0.16 | 0.00 | 0.18 | 0.00 | 0.54 | 39.08 | 0.14 | 0.00 | 0.22 | 52.70 | 0.52 | 0.00 |
| DoggoPush1 | 0.90 | 32.70 | 0.08 | 0.00 | 0.29 | 11.03 | 0.48 | 19.40 | 0.49 | 38.80 | 0.41 | 0.00 | 0.32 | 0.00 | 0.27 | 17.10 | 0.58 | 85.10 |
| PointButton1 | 26.10 | 151.38 | 5.83 | 32.98 | -0.12 | 20.88 | 7.13 | 32.31 | 3.01 | 28.14 | 3.20 | 40.16 | 2.18 | 54.74 | 4.70 | 31.39 | 6.60 | 38.27 |
| PointButton2 | 27.96 | 166.74 | 0.27 | 31.49 | 0.44 | 30.87 | 4.87 | 24.94 | 7.90 | 53.82 | 5.58 | 47.68 | 1.12 | 41.49 | 3.52 | 61.98 | 1.29 | 26.13 |
| PointCircle1 | 54.57 | 202.54 | 47.00 | 23.28 | 93.84 | 52.23 | 90.87 | 33.83 | 90.65 | 35.53 | 92.10 | 43.50 | 72.81 | 56.53 | 44.98 | 15.50 | 46.06 | 22.36 |
| PointCircle2 | 54.39 | 397.54 | 41.60 | 19.92 | 83.67 | 45.27 | 82.62 | 6.63 | 83.39 | 7.40 | 85.22 | 21.20 | 79.22 | 22.67 | 41.45 | 30.98 | 42.38 | 20.96 |
| PointGoal1 | 26.32 | 48.20 | 12.46 | 37.62 | 8.15 | 26.31 | 18.99 | 22.87 | 13.90 | 24.66 | 20.52 | 27.44 | 18.79 | 20.48 | 11.99 | 18.15 | 14.77 | 32.95 |
| PointGoal2 | 26.43 | 159.28 | 0.59 | 59.43 | -0.56 | 60.37 | 4.18 | 26.80 | 1.84 | 29.19 | 2.65 | 42.40 | 1.32 | 37.66 | 1.00 | 162.97 | 2.71 | 18.63 |
| PointPush1 | 0.82 | 57.80 | 0.80 | 33.18 | 0.29 | 8.87 | 0.70 | 24.93 | 4.35 | 23.47 | 1.82 | 19.90 | 1.41 | 31.33 | 1.90 | 19.98 | 0.93 | 62.64 |
| PointPush2 | 1.39 | 42.82 | 0.52 | 25.90 | 1.01 | 25.87 | 1.05 | 56.07 | 0.54 | 29.83 | 1.50 | 29.17 | 0.59 | 27.57 | 1.26 | 56.08 | 0.44 | 39.24 |
| RacecarButton1 | 8.48 | 343.15 | -0.05 | 48.55 | -1.37 | 51.57 | -0.18 | 44.25 | -0.63 | 29.70 | 0.02 | 60.95 | 0.13 | 45.45 | 0.04 | 130.63 | -0.88 | 84.20 |
| RacecarButton2 | 5.77 | 284.15 | -0.58 | 22.35 | -0.64 | 31.80 | 0.19 | 65.00 | 0.38 | 18.45 | 0.01 | 32.90 | 0.04 | 51.95 | -0.40 | 72.57 | -0.40 | 57.65 |
| RacecarCircle1 | 81.62 | 396.80 | 67.49 | 47.55 | 47.66 | 33.13 | 65.54 | 54.55 | 67.39 | 51.75 | 64.77 | 20.20 | 18.05 | 71.65 | 60.68 | 88.33 | 62.77 | 52.85 |
| RacecarCircle2 | 82.61 | 831.00 | 46.85 | 26.05 | 28.04 | 47.37 | 60.83 | 45.65 | 61.40 | 33.00 | 59.17 | 48.30 | 8.81 | 35.05 | 41.50 | 16.13 | 52.38 | 35.10 |
| RacecarGoal1 | 11.29 | 106.40 | 2.90 | 12.70 | -0.42 | 26.87 | 13.40 | 19.20 | 9.89 | 20.70 | 13.30 | 64.50 | 3.72 | 5.90 | 1.47 | 30.57 | 3.47 | 15.40 |
| RacecarGoal2 | 9.61 | 158.25 | 0.08 | 54.40 | -0.85 | 30.50 | 0.40 | 14.30 | 0.55 | 16.80 | 1.19 | 109.85 | 0.69 | 41.90 | -0.09 | 62.33 | 0.17 | 93.05 |
| RacecarPush1 | 0.50 | 58.45 | -0.20 | 0.00 | -0.42 | 71.83 | 0.37 | 44.75 | 0.29 | 48.00 | 0.47 | 3.30 | -0.08 | 4.50 | -0.03 | 94.70 | 0.15 | 51.00 |
| RacecarPush2 | 0.58 | 213.95 | 0.37 | 43.85 | -0.08 | 24.07 | -0.12 | 56.00 | -0.03 | 0.00 | 0.23 | 9.55 | -0.51 | 49.75 | -1.54 | 101.50 | -0.54 | 56.00 |
| **Safety Velocity** | $J^R$ | $J^C$ | $J^R$ | $J^C$ | $J^R$ | $J^C$ | $J^R$ | $J^C$ | $J^R$ | $J^C$ | $J^R$ | $J^C$ | $J^R$ | $J^C$ | $J^R$ | $J^C$ | $J^R$ | $J^C$ |
| AntVel | 5899.64 | 943.57 | 3221.90 | 5.43 | 3070.67 | 10.23 | 3157.40 | 3.63 | 3087.03 | 14.12 | 3116.77 | 14.10 | 2276.19 | 10.18 | 3297.29 | 23.56 | 3291.30 | 15.07 |
| HalfCheetahVel | 7013.92 | 933.18 | 3025.42 | 0.00 | 3336.80 | 1.09 | 2952.08 | 25.23 | 2520.50 | 13.95 | 2738.36 | 5.68 | 1743.71 | 15.64 | 2765.42 | 4.28 | 2873.14 | 2.88 |
| HopperVel | 2378.23 | 543.14 | 1347.98 | 22.30 | 1709.13 | 11.11 | 1377.89 | 17.67 | 1355.69 | 14.85 | 1713.22 | 12.12 | 1519.59 | 12.79 | 1716.35 | 5.37 | 1538.79 | 7.43 |
| HumanoidVel | 9117.61 | 959.76 | 6586.70 | 18.95 | 6620.69 | 0.00 | 6552.06 | 59.85 | 6236.18 | 20.57 | 6486.40 | 0.22 | 5863.98 | 0.18 | 6181.80 | 19.88 | 6502.90 | 23.23 |
| SwimmerVel | 121.23 | 171.21 | 68.10 | 27.68 | 109.34 | 22.92 | 79.63 | 20.98 | 64.73 | 22.56 | 61.49 | 20.46 | 60.48 | 17.31 | 70.86 | 23.93 | 55.87 | 32.62 |
| Walker2dVel | 6312.27 | 899.82 | 2756.61 | 4.90 | 1704.06 | 8.90 | 3209.78 | 19.18 | 3072.07 | 3.72 | 2440.82 | 20.15 | 1698.31 | 17.73 | 2739.50 | 4.39 | 3116.08 | 3.93 |

(b) The performance of `SafePO` multi-agent algorithms on `Safety-Gymnasium`.

| Safety Velocity | MAPPO $J^R$ | MAPPO $J^C$ | HAPPO $J^R$ | HAPPO $J^C$ | MAPPO-Lag $J^R$ | MAPPO-Lag $J^C$ | MACPO $J^R$ | MACPO $J^C$ |
|---|---|---|---|---|---|---|---|---|
| 2x4AntVel | 4259.52 | 894.06 | 5368.61 | 978.06 | 2423.47 | 0.00 | 2169.23 | 3.39 |
| 4x2AntVel | 4309.05 | 950.33 | 4613.69 | 858.50 | 2171.40 | 0.00 | 2172.31 | 0.17 |
| 2x3HalfCheetahVel | 5057.63 | 975.50 | 5605.98 | 942.56 | 1750.96 | 0.33 | 2470.29 | 32.06 |
| 6x1HalfCheetahVel | 5061.53 | 980.67 | 5540.57 | 943.56 | 1439.38 | 0.61 | 1830.65 | 9.33 |
| 3x1HopperVel | 2115.35 | 564.56 | 2207.50 | 551.33 | 1002.01 | 0.00 | 461.25 | 25.78 |
| 9\|8HumanoidVel | 974.50 | 158.61 | 2718.48 | 429.61 | 526.69 | 21.00 | 512.29 | 32.50 |
| 2x1SwimmerVel | 39.88 | 101.89 | 51.95 | 267.00 | 27.89 | 59.73 | -4.02 | 20.83 |
| 2x3Walker2dVel | 2691.41 | 574.72 | 4183.34 | 841.83 | 1618.98 | 0.33 | 714.18 | 30.22 |

**Experimental Results Analysis.**

During the observation of the experimental results, we have discovered some Insightful findings that are presented below:

- The Lagrangian method is a promising yet constrained baseline approach, successfully optimizing rewards while adhering to constraints. However, its effectiveness heavily relies on hyperparameters configuration, as discussed in Table A.1. Consequently, despite being a dependable baseline, the Lagrangian method is not exempt from limitations.

- Second-order algorithms perform worse in achieving higher rewards in the MuJoCo velocity series but better in navigation series tasks that require higher safety standards, *i.e.*, achieving similar or approximate rewards while minimizing the number and smoothness of cost violations.

# B Details Documentation of Gymnasium-based Learning Environments

## B.1 Single-agent Specification

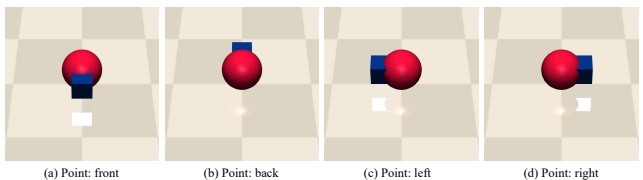

(a) Point: front  (b) Point: back  (c) Point: left  (d) Point: right

Figure 9: A different view of the robot: Point.

Table 6: The overall information of Point

| | |
|---|---|
| Specific Action Space | Box(-1.0, 1.0, (2,), float64) |
| Specific Observation Space | (12, ) |
| Observation High | inf |
| Observation Low | -inf |

Table 7: The specific observation space of Point

| Size | Observation | Min | Max | Name (in XML file) | Joint/Site | Unit |
|---|---|---|---|---|---|---|
| 3 | accelerometer | -inf | inf | accelerometer | site | acceleration (m/s^2) |
| 3 | velocimeter | -inf | inf | velocimeter | site | velocity (m/s) |
| 3 | gyro | -inf | inf | gyro | site | anglular velocity (rad/s) |
| 3 | magnetometer | -inf | inf | magnetometer | site | magnetic flux (Wb) |

Table 8: The specific action space of Point

| Num | Action | Control Min | Control Max | Name (in XML file) | Joint/Site | Unit |
|---|---|---|---|---|---|---|
| 0 | force applied on the agent to move forward or backward | -1 | 1 | x | site | force (N) |
| 1 | velocity of the agent, which is around the z-axis | -1 | 1 | z | hinge | velocity (m/s) |

**Point:** As shown in Figure 9, `Point` operating within a 2D plane is equipped with two distinct actuators: one for rotation and another for forward/backward movement. This decomposed control system greatly facilitates the navigation of the robot. Moreover, there is a small square positioned in front of the robot, aiding in the visual identification of its orientation. Additionally, this square plays a crucial role in assisting the robot, named Point, to effectively push any boxes encountered during its tasks. The overall information of `Point`, the specific action and observation space of `Point` is shown in Table 6, Table 8, Table 7.

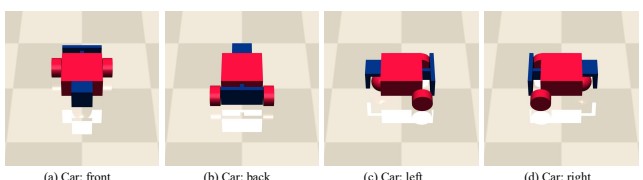

(a) Car: front  (b) Car: back  (c) Car: left  (d) Car: right

Figure 10: A different view of the robot: Car.

Table 9: The overall information of Car

| | |
|---|---|
| Specific Action Space | Box(-1.0, 1.0, (2,), float64) |
| Specific Observation Space | (24, ) |
| Observation High | inf |
| Observation Low | -inf |

Table 10: The specific action space of Car

| Num | Action | Control Min | Control Max | Name (in XML file) | Joint/Site | Unit |
|---|---|---|---|---|---|---|
| 0 | force to applied on left wheel | -1 | 1 | left | hinge | force (N) |
| 1 | force to applied on right wheel | -1 | 1 | right | hinge | force (N) |

Table 11: The specific observation space of Car

| Size | Observation | Min | Max | Name (in XML file) | Joint/Site | Unit |
|---|---|---|---|---|---|---|
| 3 | Quaternions of the rear wheel which are turned into 3x3 rotation matrices | -inf | inf | ballquat_rear | ball | unitless |
| 3 | Angle velocity of the rear wheel | -inf | inf | ballangvel_rear | ball | anglular velocity (rad/s) |
| 3 | accelerometer | -inf | inf | accelerometer | site | acceleration (m/s^2) |
| 3 | velocimeter | -inf | inf | velocimeter | site | velocity (m/s) |
| 3 | gyro | -inf | inf | gyro | site | anglular velocity (rad/s) |
| 3 | magnetometer | -inf | inf | magnetometer | site | magnetic flux (Wb) |

**Car:** As shown in Figure 10, the robot in question operates in three dimensions and features two independently driven parallel wheels, along with a freely rolling rear wheel. This design requires coordinated operation of the two drives for both steering and forward/backward movement. While the robot shares similarities with a basic Point robot, it possesses added complexity. The overall information of Car, the specific action and observation space of Car is shown in Table 9, Table 10, Table 11.

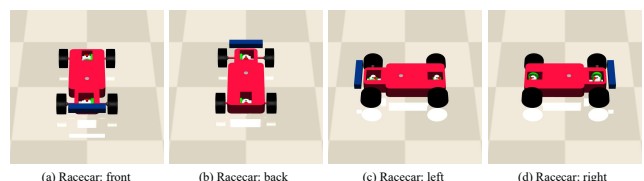

(a) Racecar: front    (b) Racecar: back    (c) Racecar: left    (d) Racecar: right

Figure 11: A different view of the robot: Racecar.

Table 12: The overall information of Racear

| | |
|---|---|
| Specific Action Space | Box([-20. -0.785], [20. 0.785], (2,), float64) |
| Specific Observation Space | (12, ) |
| Observation High | inf |
| Observation Low | -inf |

Table 13: The specific action space of Racecar

| Num | Action | Control Min | Control Max | Name (in XML file) | Joint/Site | Unit |
|---|---|---|---|---|---|---|
| 0 | Velocity of the rear wheels. | -20 | 20 | diff_ring | hinge | velocity (m/s) |
| 1 | Angle of the front wheel. | -0.785 | 0.785 | steering_hinge | hinge | angle (rad) |

Table 14: The specific observation space of Racecar

| Size | Observation | Min | Max | Name (in XML file) | Joint/Site | Unit |
|---|---|---|---|---|---|---|
| 3 | accelerometer | -inf | inf | accelerometer | site | acceleration (m/s^2) |
| 3 | velocimeter | -inf | inf | velocimeter | site | velocity (m/s) |
| 3 | gyro | -inf | inf | gyro | site | anglular velocity (rad/s) |
| 3 | magnetometer | -inf | inf | magnetometer | site | magnetic flux (Wb) |

**Racecar.** As shown in Figure 11, the robot is closer to realistic car dynamics, moving in three dimensions, it has one velocity servo and one position servo, one to adjusts the rear wheel speed to

the target speed and the other to adjust the front wheel steering angle to the target angle. Racecar references the widely known MIT Racecar project's dynamics model. For it to accomplish the specified goal, it must coordinate the relationship between the steering angle of the tires and the speed, just like a human driving a car. The overall information of Racecar, the specific action and observation space of Racecar is shown in Table 12, Table 13, Table 14.

Table 15: The overall information of Ant

| Specific Action Space | Box(-1.0, 1.0, (8,), float64) |
|---|---|
| Specific Observation Space | (40, ) |
| Observation High | inf |
| Observation Low | -inf |

Table 16: The specific action space of Ant

| Num | Action | Control Min | Control Max | Name (in XML file) | Joint/Site | Unit |
|---|---|---|---|---|---|---|
| 0 | torque applied on the rotor between the torso and front left hip | -1 | 1 | hip_1 (front_left_leg) | hinge | torque (N m) |
| 1 | torque applied on the rotor between the front left two links | -1 | 1 | angle_1 (front_left_leg) | hinge | torque (N m) |
| 2 | torque applied on the rotor between the torso and front right hip | -1 | 1 | hip_2 (front_right_leg) | hinge | torque (N m) |
| 3 | torque applied on the rotor between the front right two links | -1 | 1 | angle_2 (front_right_leg) | hinge | torque (N m) |
| 4 | torque applied on the rotor between the torso and back left hip | -1 | 1 | hip_3 (back_leg) | hinge | torque (N m) |
| 5 | torque applied on the rotor between the back left two links | -1 | 1 | angle_3 (back_leg) | hinge | torque (N m) |
| 6 | torque applied on the rotor between the torso and back right hip | -1 | 1 | hip_4 (right_back_leg) | hinge | torque (N m) |
| 7 | torque applied on the rotor between the back right two links | -1 | 1 | angle_4 (right_back_leg) | hinge | torque (N m) |

**Ant.** As depicted in Figure 12, the quadrupedal robot, inspired by the model proposed in [45]. It consists of a torso and four interconnected legs. Each leg is composed of two hinged connecting limbs, which, in turn, are connected to the torso via hinges. To achieve movement in the desired direction, coordination of the four legs is required by applying moments to the eight hinge drivers. For a comprehensive understanding of the robot, please refer to Table 15, Table 16, and Table 17, which provide an overview of the Ant robot, its specific action space, and observation space, respectively.

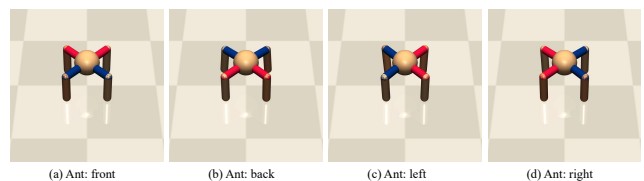

(a) Ant: front    (b) Ant: back    (c) Ant: left    (d) Ant: right

Figure 12: A different view of the robot: Ant.

Table 17: The specific observation space of Ant

| Size | Observation | Min | Max | Name (in XML file) | Joint/Site | Unit |
|---|---|---|---|---|---|---|
| 3 | accelerometer | -inf | inf | accelerometer | site | acceleration (m/s^2) |
| 3 | velocimeter | -inf | inf | velocimeter | site | velocity (m/s) |
| 3 | gyro | -inf | inf | gyro | site | anglular velocity (rad/s) |
| 3 | magnetometer | -inf | inf | magnetometer | site | magnetic flux (Wb) |
| 1 | angular velocity of angle between torso and front left link | -inf | inf | hip_1 (front_left_leg) | hinge | angle (rad) |
| 1 | angular velocity of the angle between front left links | -inf | inf | ankle_1 (front_left_leg) | hinge | angle (rad) |
| 1 | angular velocity of angle between torso and front right link | -inf | inf | hip_2 (front_right_leg) | hinge | angle (rad) |
| 1 | angular velocity of the angle between front right links | -inf | inf | ankle_2 (front_right_leg) | hinge | angle (rad) |
| 1 | angular velocity of angle between torso and back left link | -inf | inf | hip_3 (back_leg) | hinge | angle (rad) |
| 1 | angular velocity of the angle between back left links | -inf | inf | ankle_3 (back_leg) | hinge | angle (rad) |
| 1 | angular velocity of angle between torso and back right link | -inf | inf | hip_4 (right_back_leg) | hinge | angle (rad) |
| 1 | angular velocity of the angle between back right links | -inf | inf | ankle_4 (right_back_leg) | hinge | angle (rad) |
| 1 | z-coordinate of the torso (centre). | -inf | inf | torso | site | position (m) |
| 3 | xyz-coordinate angular velocity of the tors. | -inf | inf | torso | site | angular velocity (rad/s) |
| 2 | sin() and cos() of angle between torso and first link on front left | -inf | inf | hip_1 (front_left_leg) | hinge | unitless |
| 2 | sin() and cos() of angle between torso and first link on front left | -inf | inf | ankle_1 (front_left_leg) | hinge | unitless |
| 2 | sin() and cos() of angle between torso and first link on front left | -inf | inf | hip_2 (front_right_leg) | hinge | unitless |
| 2 | sin() and cos() of angle between torso and first link on front left | -inf | inf | ankle_2 (front_right_leg) | hinge | unitless |
| 2 | sin() and cos() of angle between torso and first link on front left | -inf | inf | hip_3 (back_leg) | hinge | unitless |
| 2 | sin() and cos() of angle between torso and first link on front left | -inf | inf | ankle_3 (back_leg) | hinge | unitless |
| 2 | sin() and cos() of angle between torso and first link on front left | -inf | inf | hip_4 (right_back_leg) | hinge | unitless |
| 2 | sin() and cos() of angle between torso and first link on front left | -inf | inf | ankle_4 (right_back_leg) | hinge | unitless |

## B.2 Multi-agents Specification

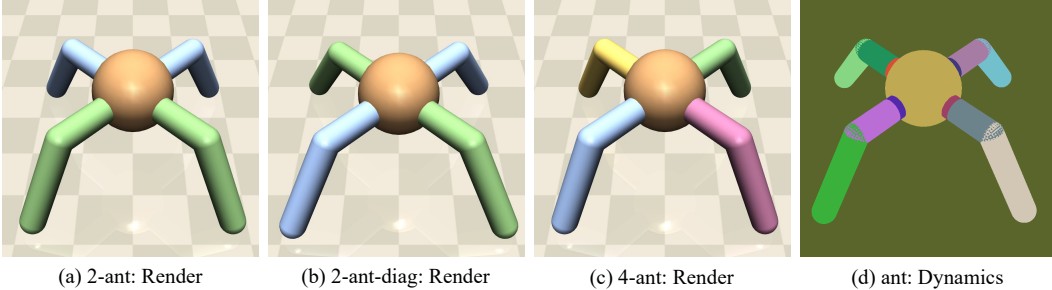

(a) 2-ant: Render    (b) 2-ant-diag: Render    (c) 4-ant: Render    (d) ant: Dynamics

Figure 13: A different view of the MA-Ant.

**2-ant.** The `Ant` is partitioned into 2 parts, the front part (containing the front legs) and the back part (containing the back legs). The action space of agent-0 and agent-1 as shown in Table 18 and Table 19.

Table 18: The specific action space of 2-ant: agent-0

| Num | Action | Control Min | Control Max | Name (in XML file) | Joint | Unit |
|---|---|---|---|---|---|---|
| 0 | Torque applied on the rotor between the torso and front left hip | -1 | 1 | hip_1 (front_left_leg) | hinge | torque (N m) |
| 1 | Torque applied on the rotor between the front left two links | -1 | 1 | angle_1 (front_left_leg) | hinge | torque (N m) |
| 2 | Torque applied on the rotor between the torso and front right hip | -1 | 1 | hip_2 (front_right_leg) | hinge | torque (N m) |
| 3 | Torque applied on the rotor between the front right two links | -1 | 1 | angle_2 (front_right_leg) | hinge | torque (N m) |

Table 19: The specific action space of 2-ant: agent-1

| Num | Action | Control Min | Control Max | Name (in XML file) | Joint | Unit |
|---|---|---|---|---|---|---|
| 0 | Torque applied on the rotor between the torso and front left hip | -1 | 1 | hip_1 (front_left_leg) | hinge | torque (N m) |
| 1 | Torque applied on the rotor between the front left two links | -1 | 1 | angle_1 (front_left_leg) | hinge | torque (N m) |
| 2 | Torque applied on the rotor between the torso and front right hip | -1 | 1 | hip_2 (front_right_leg) | hinge | torque (N m) |
| 3 | Torque applied on the rotor between the front right two links | -1 | 1 | angle_2 (front_right_leg) | hinge | torque (N m) |

**2-ant-diag.** The Ant is partitioned into 2 parts, split diagonally, the front part (containing the front legs) and the back part (containing the back legs). The action space of agent-0 and agent-1 as shown in Table 20 and Table 21.

Table 20: The specific action space of 2-ant-diag: agent-0

| Num | Action | Control Min | Control Max | Name (in XML file) | Joint | Unit |
|---|---|---|---|---|---|---|
| 0 | Torque applied on the rotor between the torso and front left hip | -1 | 1 | hip_1 (front_left_leg) | hinge | torque (N m) |
| 1 | Torque applied on the rotor between the front left two links | -1 | 1 | angle_1 (front_left_leg) | hinge | torque (N m) |
| 2 | Torque applied on the rotor between the torso and back right hip | -1 | 1 | hip_4 (right_back_leg) | hinge | torque (N m) |
| 3 | Torque applied on the rotor between the back right two links | -1 | 1 | angle_4 (right_back_leg) | hinge | torque (N m) |

Table 21: The specific action space of 4-ant: agent-1

| Num | Action | Control Min | Control Max | Name (in XML file) | Joint | Unit |
|---|---|---|---|---|---|---|
| 0 | Torque applied on the rotor between the torso and front right hip | -1 | 1 | hip_2 (front_right_leg) | hinge | torque (N m) |
| 1 | Torque applied on the rotor between the front right two links | -1 | 1 | angle_2 (front_right_leg) | hinge | torque (N m) |
| 2 | Torque applied on the rotor between the torso and back left hip | -1 | 1 | hip_3 (back_leg) | hinge | torque (N m) |
| 3 | Torque applied on the rotor between the back left two links | -1 | 1 | angle_3 (back_leg) | hinge | torque (N m) |

**4-ant.** The Ant is partitioned into 4 parts, with each part corresponding to a leg of the ant. The action space of agent-0, agent-1, agent-2, and agent-3 as shown in Table 22, Table 23, Table 24 and Table 25.

Table 22: The specific action space of 4-ant: agent-0

| Num | Action | Control Min | Control Max | Name (in XML file) | Joint | Unit |
|---|---|---|---|---|---|---|
| 0 | Torque applied on the rotor between the torso and front left hip | -1 | 1 | hip_1 (front_left_leg) | hinge | torque (N m) |
| 1 | Torque applied on the rotor between the front left two links | -1 | 1 | angle_1 (front_left_leg) | hinge | torque (N m) |

Table 23: The specific action space of 2-ant-diag: agent-1

| Num | Action | Control Min | Control Max | Name (in XML file) | Joint | Unit |
|---|---|---|---|---|---|---|
| 0 | Torque applied on the rotor between the torso and front right hip | -1 | 1 | hip_2 (front_right_leg) | hinge | torque (N m) |
| 1 | Torque applied on the rotor between the front right two links | -1 | 1 | angle_2 (front_right_leg) | hinge | torque (N m) |

Table 24: The specific action space of 4-ant: agent-2

| Num | Action | Control Min | Control Max | Name (in XML file) | Joint | Unit |
|---|---|---|---|---|---|---|
| 0 | Torque applied on the rotor between the torso and back left hip | -1 | 1 | hip_3 (back_leg) | hinge | torque (N m) |
| 1 | Torque applied on the rotor between the back left two links | -1 | 1 | angle_3 (back_leg) | hinge | torque (N m) |

Table 25: The specific action space of 4-ant: agent-3

| Num | Action | Control Min | Control Max | Name (in XML file) | Joint | Unit |
|---|---|---|---|---|---|---|
| 0 | Torque applied on the rotor between the torso and back right hip | -1 | 1 | hip_4 (right_back_leg) | hinge | torque (N m) |
| 1 | Torque applied on the rotor between the back right two links | -1 | 1 | angle_4 (right_back_leg) | hinge | torque (N m) |

In addition to the robots mentioned in this paper, we also provide other multi-agent versions of robots. Due to space constraints, we did not elaborate on them extensively in the paper. However, you can refer to `https://www.safety-gymnasium.com/` for more detailed information.

## B.3 Task Representation

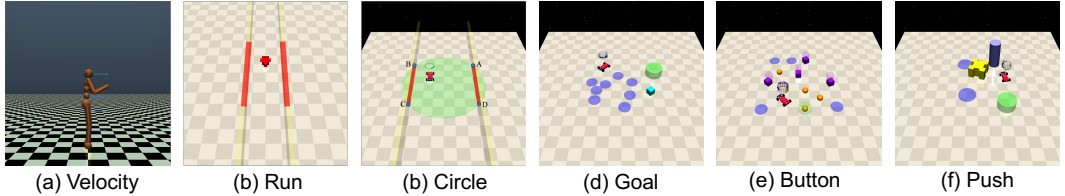

(a) Velocity     (b) Run     (b) Circle     (d) Goal     (e) Button     (f) Push

Figure 14: Tasks of Gymnasium-based Environments.

As shown in Figure 14, the Gymnasium-based learning environments support the following tasks:

**Velocity:** the robot aims to facilitate coordinated leg movement of the robot in the forward (right) direction by exerting torques on the hinges.

**Run:** the robot starts with a random initial direction and a specific initial speed as it embarks on a journey to reach the opposite side of the map.

**Circle:** the reward is maximized by moving along the green circle, and the agent is not allowed to enter the outside of the red region, so its optimal constrained path follows the line segments $AD$ and $BC$. The reward function: $R(s) = \frac{v^T[-y, x]}{1 + |\|[x,y]\|_2 - d|}$, the cost function is $C(s) = \mathbf{1}\left[|x| > x_{\lim}\right]$, where $x, y$ are the coordinates in the plane, $v$ is the velocity, and $d, x_{\lim}$ are environmental parameters.

**Goal:** the robot navigates to multiple goal positions. After successfully reaching a goal, its location is randomly reset while maintaining the overall layout. Achieving a goal position, indicated by entering the goal circle, yields a sparse reward. Additionally, a dense reward encourages the robot's progress by rewarding proximity to the goal.

**Push:** the objective is to move a box to a series of goal positions. Like the goal task, a new random goal location is generated after each successful achievement. The sparse reward is earned when the yellow box enters the designated goal circle. The dense reward consists of two components: one for moving the agent closer to the box and another for bringing the box closer to the final goal.

**Button:** the objective is to activate a series of goal buttons distributed throughout the environment. The agent's goal is to navigate towards and make contact with the currently highlighted button, known as the goal button. Once the correct button is pressed, a new goal button is selected and highlighted while preserving the rest of the environment. The sparse reward is earned upon successfully pressing the current goal button, while the dense reward component provides a bonus for progressing toward the highlighted goal button.

### B.4 Constraint Specification

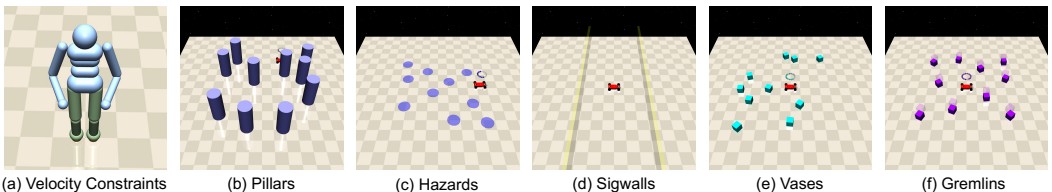

|  |  |  |  |  |  |
|---|---|---|---|---|---|
| (a) Velocity Constraints | (b) Pillars | (c) Hazards | (d) Sigwalls | (e) Vases | (f) Gremlins |

Figure 15: Constraints of Gymnasium-based Environments.

**Velocity-Constraint** consists of a series of safety tasks based on MuJoCo agents [23]. In these tasks, agents, such as `Ant`, `HalfCheetah`, and `Humanoid`, are trained to move faster for higher rewards, while also being imposed a velocity constraint for safety considerations. Formally, for an agent moving on a two-dimensional plane, the velocity is calculated as $v(s,a) = \sqrt{v_x^2 + v_y^2}$; for an agent moving along a straight line, the velocity is calculated as $v(s,a) = |v_x|$, where $v_x, v_y$ are the velocities of the agent in the $x$ and $y$ directions respectively. Then, $cost(s,a) = [v(s,a) > v_{limit}]$, Here, $[P]$ denotes a notation where the value is 1 if the proposition $P$ is true, and 0 otherwise.

**Pillars** are employed to represent large cylindrical obstacles within the environment. In the general setting, contact with a pillar incurs costs.

**Hazards** are utilized to model areas within the environment that pose a risk, resulting in costs when an agent enters such areas.

**Sigwalls** are designed specifically for Circle tasks. They serve as visual representations of two or four solid walls, which limit the circular area to a smaller region. Crossing the wall from inside the safe area to the outside incurs costs.

**Vases** are specifically designed for Goal tasks. They represent static and fragile objects within the environment. Touching or displacing these objects incurs costs for the agent.

**Gremlins** are specifically employed in the Button tasks. They represent moving objects within the environment that can interact with the agent.

### B.5 Vision-only Tasks

In recent years, vision-only SafeRL has gained significant attention as a focal point of research, primarily due to its applicability in real-world contexts [40; 41]. While the initial iteration of Safety Gym offered rudimentary visual input support, there is room for enhancing the realism and complexity of its environment. To effectively evaluate vision-based safe reinforcement learning algorithms, we have devised some more realistic visual tasks utilizing MuJoCo. This enhanced environment facilitates the incorporation of both RGB and RGB-d inputs. More details can be

referred to our online documentation: `https://www.safety-gymnasium.com/en/latest/env ironments/safe_vision.html`.

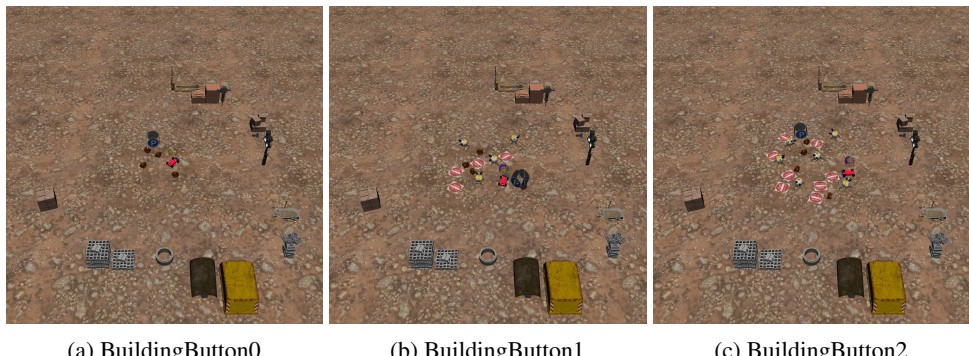

(a) BuildingButton0      (b) BuildingButton1      (c) BuildingButton2

Figure 16: Overview of BuildingButton tasks.

**The Level 0 of BuildingButton** requires the agent to operate multiple machines within a construction site.

**The Level 1 of BuildingButton** requires the agent to proficiently and accurately operate multiple machines within a construction site, while concurrently evading other robots and obstacles present in the area.

**The Level 2 of BuildingButton** requires the agent to proficiently and accurately operate multiple machines within a construction site, while concurrently evading a heightened number of other robots and obstacles in the area.

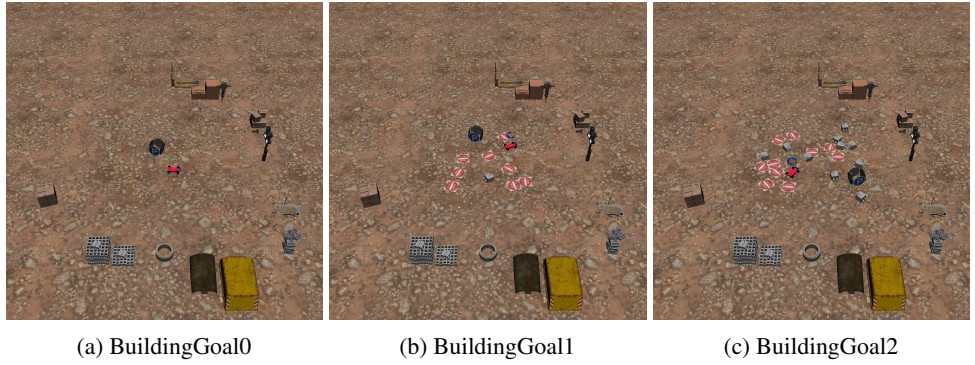

(a) BuildingGoal0      (b) BuildingGoal1      (c) BuildingGoal2

Figure 17: Overview of BuildingGoal tasks.

**The Level 0 of BuildingGoal** requires the agent to dock at designated positions within a construction site.

**The Level 1 of BuildingGoal** requires the agent to dock at designated positions within a construction site while ensuring to avoid entry into hazardous areas.

**The Level 2 of BuildingGoal** requires the agent to dock at designated positions within a construction site, while ensuring to avoid entry into hazardous areas and circumventing the site's exhaust fans.

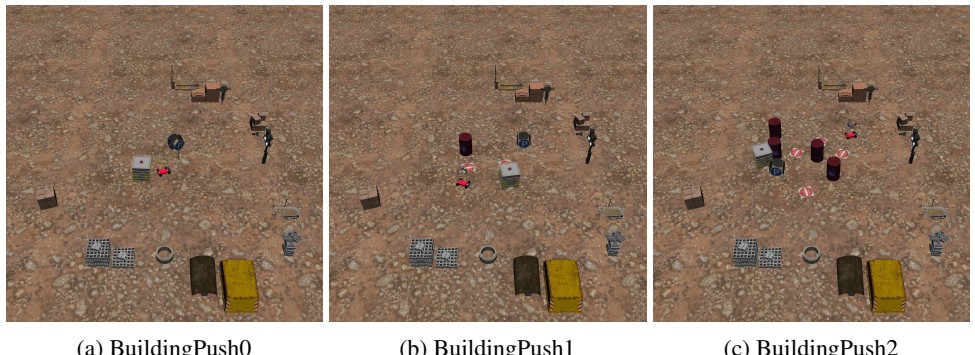

(a) BuildingPush0                    (b) BuildingPush1                    (c) BuildingPush2

Figure 18: Overview of BuildingPush tasks.

**The Level 0 of BuildingPush** requires the agent to relocate the box to designated locations within a construction site.

**The Level 1 of BuildingPush** requires the agent to relocate the box to designated locations within a construction site while avoiding areas demarcated as restricted.

**The Level 2 of BuildingPush** requires the agent to relocate the box to designated locations within a construction while avoiding numerous hazardous fuel drums and areas demarcated as restricted.

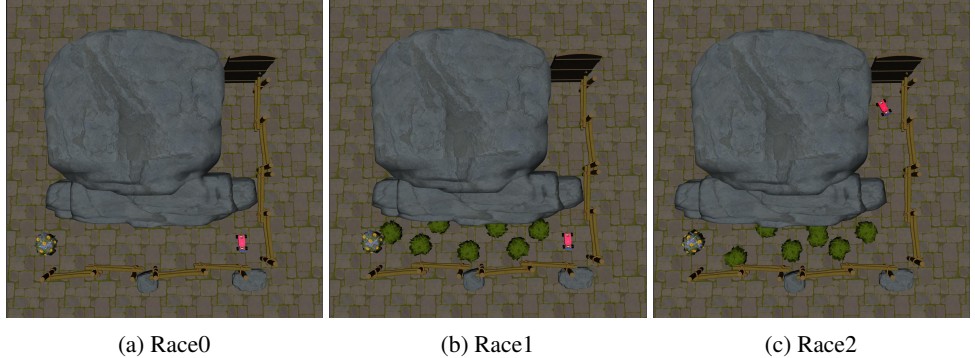

(a) Race0                            (b) Race1                            (c) Race2

Figure 19: Overview of Race tasks.

**The Level 0 of Race** requires the agent to reach the goal position.

**The Level 1 of Race** requires the agent to reach the goal position while ensuring it avoids straying into the grass and prevents collisions with roadside objects.

**The Level 2 of Race** requires the agent to reach the goal position from a distant starting point while ensuring it avoids straying into the grass and prevents collisions with roadside objects.

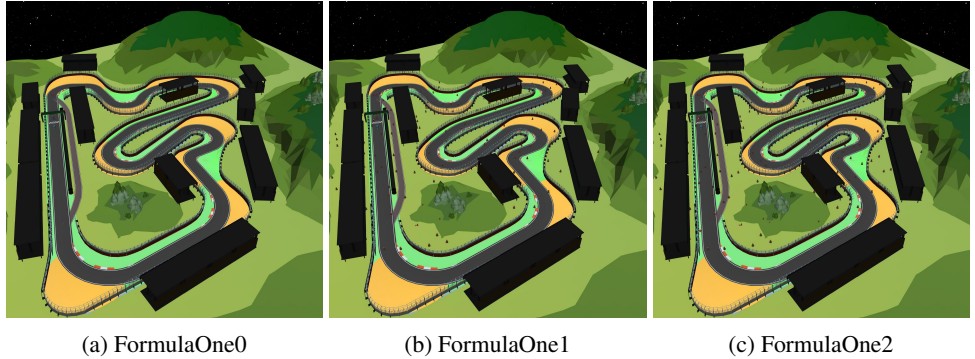

(a) FormulaOne0          (b) FormulaOne1          (c) FormulaOne2

Figure 20: Overview of FormulaOne tasks.

**The Level 0 of FormulaOne** requires the agent to maximize its reach to the goal position. For each episode, the agent is randomly initialized at one of the seven checkpoints.

**The Level 1 of FormulaOne** requires the agent to maximize its reach to the goal position while circumventing barriers and racetrack fences. For each episode, the agent is randomly initialized at one of the seven checkpoints.

**The Level 2 of FormulaOne** requires the agent to maximize its reach to the goal position while circumventing barriers and racetrack fences. For each episode, the agent is randomly initialized at one of the seven checkpoints. Notably, the barriers surrounding the checkpoints are denser.

## B.6 Some Issues about Safety Gym

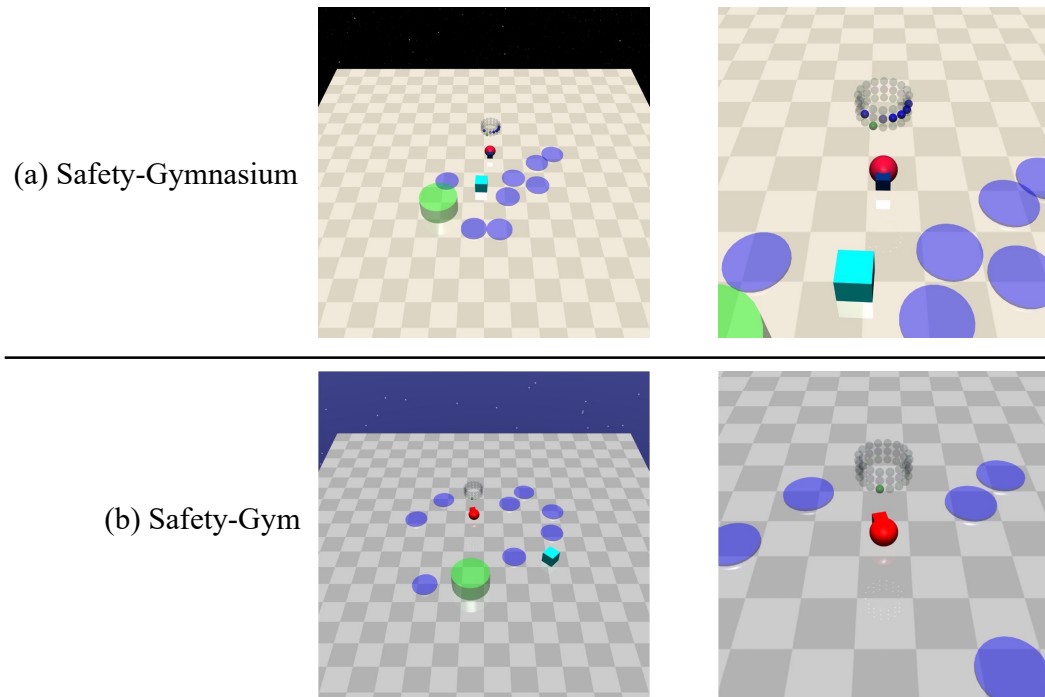

Figure 21: The difference between `Safety-Gymnasium` and Safety Gym.

**The bug of Natural Lidar.** As shown in Figure 21, the original Natural Lidar in Safe-Gym[7] has a problem of not being able to detect low-lying objects, which may affect comprehensive environmental observations.

**The problem of observation space.** In Safety Gym, by default, the observation space is presented as a one-dimensional array. The implementation leads to all ranges in observation space to be $[-\infty, +\infty]$, as shown in the following code:

```
if self.observation_flatten:                                    1
    self.obs_flat_size = sum([np.prod(i.shape) for i in        2
        self.obs_space_dict.values()])
    self.observation_space = gym.spaces.Box(-np.inf, np.inf,   3
        (self.obs_flat_size,), dtype=np.float32)
```

While this representation does not lead to behavioral errors in the environment, it can be somewhat misleading for users. To address this issue, we have implemented the Gymnasium's flatten mechanism in the Safety Gym to handle the representation of the observation space. This mechanism reorganizes the observation space into a more intuitive and easily understandable format, enabling users to process and analyze the observation data more effectively.

```
self.obs_info.obs_space_dict = gymnasium.spaces.Dict(obs_space_dict)   1
                                                                        2
if self.observation_flatten:                                            3
    self.observation_space = gymnasium.spaces.utils.flatten_space(      4
        self.obs_info.obs_space_dict                                    5
    )                                                                   6
else:                                                                   7
    self.observation_space = self.obs_info.obs_space_dict               8
assert self.obs_info.obs_space_dict.contains(                           9
    obs                                                                10
), f'Bad obs {obs} {self.obs_info.obs_space_dict}'                     11
                                                                        12
if self.observation_flatten:                                           13
    obs =                                                              14
        gymnasium.spaces.utils.flatten(self.obs_info.obs_space_dict,
        obs)
    return obs                                                         15
```

**Missing cost information.** In Safety Gym, by default, there are only two possible outputs for the cost: 0 and 1, representing whether a cost is incurred or not.

```
# Optionally remove shaping from reward functions.              1
if self.constrain_indicator:                                    2
    for k in list(cost.keys()):                                3
        cost[k] = float(cost[k] > 0.0)  # Indicator function   4
```

We believe that this representation method loses some information. For example, when the robot collides with a vase and causes the vase to move at different velocities, there should be different cost values associated with it to indicate subtle differences in violating constraint behaviors. Additionally, these costs incurred by the actions are accumulated into the total cost. In typical cases, algorithms use the total cost to update the policy if the total cost generated by different obstacles is limited to only two states 0 and 1, the learning potential for multiple constraints is lost when multiple costs are triggered simultaneously.

**Neglected dependency maintenance leads to conflicts.**

The **numpy =1.17.4** will cause the following problems:

```
ValueError: numpy.ndarray size changed, may indicate binary      1
    incompatibility. Expected 96 from C header, got 80 from PyObject
```

```
AttributeError: module 'numpy' has no attribute 'complex'.       1
```

---

[7]https://github.com/openai/safety-gym

# C  Details of Isaac Gym-based Learning Environments

## C.1  Supported Agents

Safety-DexteroudsHand is based on Bi-DexHands (refer to [42] for more details). Bi-DexHands aims to establish a comprehensive learning framework for two Shadow Hands, enabling them to possess a wide range of skills similar to those of humans. The Shadow Hand's joint limitations are as follows (refer to Table 26). The thumb exhibits 5 degrees of freedom with 5 joints, while the other fingers have 3 degrees of freedom and 4 joints each. The joints located at the fingertips are not controllable. Similar to human fingers, the distal joints of the fingers are interconnected, ensuring that the angle of the middle joint is always greater than or equal to that of the distal joint. This design allows the middle phalange to be curved while the distal phalange remains straight. Additionally, an extra joint (LF5) is located at the end of the little finger, enabling it to rotate in the same direction as the thumb. The wrist comprises two joints, facilitating a complete 360-degree rotation of the entire hand.

Table 26: Finger range of motion.

| Joints | Corresponds to the number of Figure 22 | Min | Max |
|---|---|---|---|
| Finger Distal (FF1,MF1,RF1,LF1) | 15, 11, 7, 3 | 0° | 90° |
| Finger Middle (FF2,MF2,RF2,LF2) | 16, 12, 8, 4 | 0° | 90° |
| Finger Base Abduction (FF3,MF3,RF3,LF3) | 17, 13, 9, 5 | -15° | 90° |
| Finger Base Lateral (FF4,MF4,RF4,LF4) | 18, 14, 10, 6 | -20° | 20° |
| Little Finger Rotation(LF5) | 19 | 0° | 45° |
| Thumb Distal (TH1) | 20 | -15° | 90° |
| Thumb Middle (TH2) | 21 | -30° | 30° |
| Thumb Base Abduction (TH3) | 22 | -12° | 12° |
| Thumb Base Lateral (TH4) | 23 | 0° | 70° |
| Thumb Base Rotation (TH5) | 24 | -60° | 60° |
| Hand Wrist Abduction (WR1) | 1 | -40° | 28° |
| Hand Wrist Lateral (WR2) | 2 | -28° | 8° |

Stiffness, damping, friction, and armature are also important physical parameters in robotics. For each Shadow Hand joint, we show our DoF properties in Table 27. This part can be adjusted in the Isaac Gym simulator.

Table 27: DoF properties of Shadow Hand.

| Joints | Stiffness | Damping | Friction | Armature |
|---|---|---|---|---|
| WR1 | 100 | 4.78 | 0 | 0 |
| WR2 | 100 | 2.17 | 0 | 0 |
| FF2 | 100 | 3.4e+38 | 0 | 0 |
| FF3 | 100 | 0.9 | 0 | 0 |
| FF4 | 100 | 0.725 | 0 | 0 |
| MF2 | 100 | 3.4e+38 | 0 | 0 |
| MF3 | 100 | 0.9 | 0 | 0 |
| MF4 | 100 | 0.725 | 0 | 0 |
| RF2 | 100 | 3.4e+38 | 0 | 0 |
| RF3 | 100 | 0.9 | 0 | 0 |
| RF4 | 100 | 0.725 | 0 | 0 |
| LF2 | 100 | 3.4e+38 | 0 | 0 |
| LF3 | 100 | 0.9 | 0 | 0 |
| LF4 | 100 | 0.725 | 0 | 0 |
| TH2 | 100 | 3.4e+38 | 0 | 0 |
| TH3 | 100 | 0.99 | 0 | 0 |
| TH4 | 100 | 0.99 | 0 | 0 |
| TH5 | 100 | 0.81 | 0 | 0 |

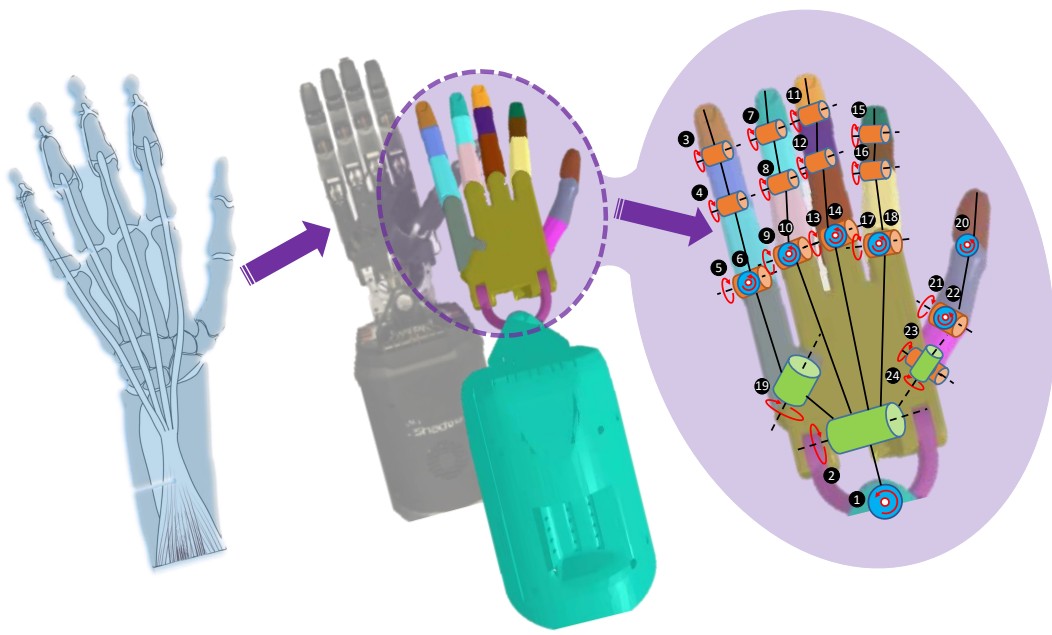

Figure 22: Degree-of-Freedom (DOF) configuration of the Shadow Hand similar to the skeleton of a human hand.

Table 28: Observation space of dual Shadow Hands.

| Index | Description |
|---|---|
| 0 - 23 | right Shadow Hand dof position |
| 24 - 47 | right Shadow Hand dof velocity |
| 48 - 71 | right Shadow Hand dof force |
| 72 - 136 | right Shadow Hand fingertip pose, linear velocity, angle velocity (5 x 13) |
| 137 - 166 | right Shadow Hand fingertip force, torque (5 x 6) |
| 167 - 169 | right Shadow Hand base position |
| 170 - 172 | right Shadow Hand base rotation |
| 173 - 198 | right Shadow Hand actions |
| 199 - 222 | left Shadow Hand dof position |
| 223 - 246 | left Shadow Hand dof velocity |
| 247 - 270 | left Shadow Hand dof force |
| 271 - 335 | left Shadow Hand fingertip pose, linear velocity, angle velocity (5 x 13) |
| 336 - 365 | left Shadow Hand fingertip force, torque (5 x 6) |
| 366 - 368 | left Shadow Hand base position |
| 369 - 371 | left Shadow Hand base rotation |
| 372 - 397 | left Shadow Hand actions |

## C.2  Task Representation

## Hand Over

This scenario encompasses a specific environment comprising two Shadow Hands positioned opposite each other, with their palms facing upwards. The objective is to pass an object between these hands. Initially, the object will randomly descend within the area of the Shadow Hand on the right side. The hand on the right side then grasps the object and transfers it to the other hand. It is important to note that the base of each hand remains fixed throughout the process. Furthermore, the hand initially holding the object cannot directly make contact with the target hand or roll the object towards it. Hence, the object must be thrown into the air, maintaining its trajectory until it reaches the target hand.

In this task, there are 398-dimensional observations and 40-dimensional actions. The reward function is closely tied to the positional discrepancy between the object and the target. As the pose error diminishes, the reward increases significantly. The detailed observation space for each agent can be found in Table 29, while the corresponding action space is outlined in Table 30.

**Observations** The observational space for the Hand Over task consists of 398 dimensions, as indicated in Table 29. However, it is important to highlight that in this particular task, the base of the dual hands remains fixed. Therefore, the observation for the dual hands is compared to a reduced 24-dimensional space, as described in Table 28.

Table 29: Observation space of Hand Over.

| Index | Description |
|---|---|
| 0 - 373 | dual hands observation shown in Table 28 |
| 374 - 380 | object pose |
| 381 - 383 | object linear velocity |
| 384 - 386 | object angle velocity |
| 387 - 393 | goal pose |
| 394 - 397 | goal rot - object rot |

**Actions** The action space for a single hand in the Hand Over task comprises 40 dimensions, as illustrated in Table 30.

Table 30: Action space of Hand Over.

| Index | Description |
|---|---|
| 0 - 19 | right Shadow Hand actuated joint |
| 20 - 39 | left Shadow Hand actuated joint |

**Rewards** Let the positions of the object and the goal be denoted as $x_o$ and $x_g$ respectively. The translational position difference between the object and the goal, represented as $d_t$, can be computed as $d_t = \|x_o - x_g\|_2$. Similarly, we define the angular position difference between the object and the goal as $d_a$. The rotational difference, denoted as $d_r$, is then calculated as $d_r = 2\arcsin(\text{clamp}(\|d_a\|_2, \max = 1.0))$.

The rewards for the Hand Over task are determined using the following formula:

$$r = \exp(-0.2(\alpha d_t + d_r)) \tag{2}$$

Here, $\alpha$ represents a constant that balances the rewards between translational and rotational aspects.

## Hand Over Catch

This environment is made up of a half Hand Over, and Catch Underarm [42], the object needs to be thrown from the vertical hand to the palm-up hand.

**Observations** The observational space for this combined task encompasses 422 dimensions, as illustrated in Table 31.

Table 31: Observation space of Hand Over Catch.

| Index | Description |
|---|---|
| 0 - 397 | dual hands observation shown in Table 28 |
| 398 - 404 | object pose |
| 405 - 407 | object linear velocity |
| 408 - 410 | object angle velocity |
| 411 - 417 | goal pose |
| 418 - 421 | goal rot - object rot |

**Actions** The action space, consisting of 52 dimensions, is illustrated in Table 32, providing a comprehensive representation of the available actions.

Table 32: Action space of Hand Over Catch.

| Index | Description |
|---|---|
| 0 - 19 | right Shadow Hand actuated joint |
| 20 - 22 | right Shadow Hand base translation |
| 23 - 25 | right Shadow Hand base rotation |
| 26 - 45 | left Shadow Hand actuated joint |
| 46 - 48 | left Shadow Hand base translation |
| 49 - 51 | left Shadow Hand base rotation |

**Rewards** Let's denote the positions of the object and the goal as $x_o$ and $x_g$, respectively. The translational position difference between the object and the goal denoted as $d_t$, can be calculated as $d_t = \|x_o - x_g\|_2$. Additionally, we define the angular position difference between the object and the goal as $d_a$. The rotational difference, denoted as $d_r$, is given by the formula $d_r = 2\arcsin(\text{clamp}(\|d_a\|_2, \max = 1.0))$. Finally, the rewards are determined using the specific formula:

$$r = \exp[-0.2(\alpha d_t + d_r)] \tag{3}$$

Here, $\alpha$ represents a constant that balances the translational and rotational rewards.

## C.3 Constraint Specification

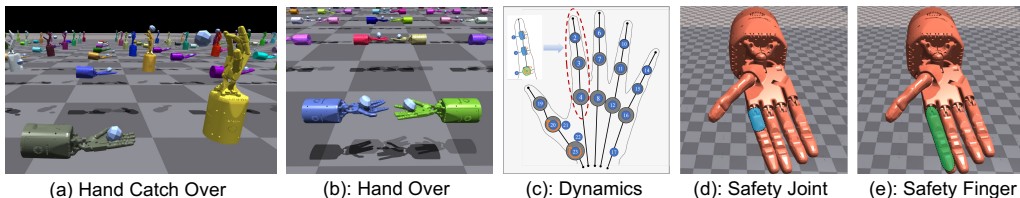

| (a) Hand Catch Over | (b): Hand Over | (c): Dynamics | (d): Safety Joint | (e): Safety Finger |

Figure 23: Tasks of Safety-DexterousHands.

**Safety Joint** constrains the freedom of joint ④ of the forefinger (please refer to Figure 23 (c) and (d)). Without the constraint, joint ④ has freedom of $[-20°, 20°]$. The safety tasks restrict joint ④ within $[-10°, 10°]$. Let ang_4 be the angle of joint ④, and the cost is defined as:

$$c_t = \mathbb{I}(\text{ang\_4} \notin [-10°, 10°]). \tag{4}$$

**Safety Finger** constrains the freedom of joints ②, ③ and ④ of forefinger (please refer to Figure 23 (c) and (e)). Without the constraint, joints ② and ③ have freedom of $[0°, 90°]$ and joint ④ of $[-20°, 20°]$. The safety tasks restrict joints ②, ③, and ④ within $[22.5°, 67.5°]$, $[22.5°, 67.5°]$, and $[-10°, 10°]$ respectively. Let ang_2, ang_3, ang_4 be the angles of joints ②, ③, ④, and the cost is defined as:

$$c_t = \mathbb{I}(\text{ang\_2} \notin [22.5°, 67.5°], \text{ or ang\_3} \notin [22.5°, 67.5°], \text{ or ang\_4} \notin [-10°, 10°]). \tag{5}$$

