# OpenReview forum: "Safety Gymnasium: A Unified Safe Reinforcement Learning Benchmark"
_NeurIPS.cc/2023/Track/Datasets_and_Benchmarks — NeurIPS 2023 Datasets and Benchmarks Poster_

### Official Review · Reviewer_EjZD · 2023-07-19
**Safe Reinforcement Learning Benchmark.**

**Rating:** 6
**Confidence:** 4
**Clarity:** This paper is easy to follow.

**Strengths:**

The availability of Safety-Gymnasium and SafePO library, as benchmark tools and validation resources, respectively, promotes collaborative research and empowers researchers to work collectively towards safer AI development.
The broad scope of tasks covered in Safety-Gymnasium and its acceptance of different input types make it suitable for diverse applications, enhancing its relevance to various research domains.

**Additional Feedback:**

1. The paper seems only to provide the performance of the algorithms on their implementation but does not compare it to the performance achieved by other well-known implementations of the same algorithms.
2. It appears to draw heavily from previous works[1,2] without introducing substantial new ideas. Utilizing existing work as one's own contribution does not align with the standard of research expected at NeurIPS. Originality and innovation are fundamental criteria for evaluating submissions, and direct appropriation of existing contributions undermines the integrity of the scientific process.

[1] Towards Human-Level Bimanual Dexterous Manipulation with Reinforcement Learning.Thirty-sixth Conference on Neural Information Processing Systems Datasets and Benchmarks Track
[2] Safe multi-agent reinforcement learning for multi-robot control. Artificial Intelligence

**Correctness:**

The claims made in the submission regarding SafePO's advantages appear to be exaggerated and misleading, as highlighted in the "Opportunities for Improvement" section, specifically points 1 and 2.

**Documentation:**

Missing description of visual tasks.

**Ethics:**

No.

**Limitations:**

The paper primarily focuses on benchmarking SafeRL algorithms using the proposed environment suite. However, the transferability of the results to complex real-world safety-critical applications may be limited. Future work should explore the adaptability and performance of SafePO in more challenging and diverse real-world scenarios.
The paper could benefit from a more extensive evaluation of the performance metrics used to assess the algorithms' safety and efficiency. Incorporating additional evaluation criteria, such as worst-case scenario analysis or robustness to adversarial perturbations, would provide a more comprehensive understanding of the SafeRL algorithms' capabilities.

**Opportunities For Improvement:**

1. Code Organization and Dependencies:
One area where the paper can be improved is in the organization and dependencies of the Safe Policy Optimization (SafePO) library. The authors claim that their library is clean and comparable to cleanRL, known for its simple and easy-to-use codebase with minimal coupling. However, it is evident that SafePO suffers from significant code dependencies and intricate inter-module calling, particularly in the multi-agent constraint code. This code structure might give the impression of integrating existing algorithms hastily, rather than providing a cohesive and streamlined implementation. To enhance the library's usability and maintainability, the authors should consider revisiting the code organization, minimizing dependencies, and ensuring a more modular design.
2. Missing Description of Visual Tasks:
The authors state that they developed realistic visual environment, but I could not find relevant descriptions in Appendix A.5. If these visual tasks have not been completed, it is essential to clearly indicate them as part of future work in the paper. Including details about these visual tasks and the challenges they pose would provide valuable insights into the comprehensiveness of the Safety-Gymnasium environment suite.
3. Baseline Algorithm Performance and Consistency:
The experimental section presents various baseline algorithms, but there seem to be substantial oscillations in satisfying constraints. This disparity in constraint satisfaction contradicts previous literature and raises concerns about the efficacy and consistency of SafePO's implementation of these algorithms. It is crucial for the authors to address these inconsistencies and provide a thorough analysis to ensure the reliability and accuracy of their SafeRL algorithms.

**Relation To Prior Work:**

Authors are candid about their relation to prior works. However, the motivation for designing a new benchmark on safe RL is unclear to me.

**Summary And Contributions:**

The authors introduce a significant contribution to the field, comprising two key components: Safety-Gymnasium, an extensive environment suite, and Safe Policy Optimization (SafePO) library, consisting of 16 state-of-the-art SafeRL algorithms.
Safety-Gymnasium serves as an all-encompassing environment suite, encompassing safety-critical tasks in both single and multi-agent scenarios, and accepting vector and vision-only inputs.
The Safe Policy Optimization (SafePO) library is a comprehensive collection of 16 cutting-edge SafeRL algorithms.

---

> ### Author Response · Authors · 2023-08-21
> **Rebuttal by Author (1/5)**
>
> > **Re: (Opportunities For Improvement # 1)** Code Organization and Dependencies: One area where the paper can be improved is in the organization and dependencies of the Safe Policy Optimization (SafePO) library. The authors claim that their library is clean and comparable to cleanRL, known for its simple and easy-to-use codebase with minimal coupling. However, it is evident that SafePO suffers from significant code dependencies and intricate inter-module calling, particularly in the multi-agent constraint code. This code structure might give the impression of integrating existing algorithms hastily, rather than providing a cohesive and streamlined implementation. To enhance the library's usability and maintainability, the authors should consider revisiting the code organization, minimizing dependencies, and ensuring a more modular design.
>
> I'm sorry that the implementation standard of SafePO didn't meet the criteria set by CleanRL. We made efforts to address this during the rebuttal process. As you mentioned: CleanRL is known for its simple and easy-to-use codebase with minimal coupling, making it a concise single-file style implementation. We often use CleanRL for our RL research, which also motivated us to develop SafePO. Our goal was for SafePO to have a positive impact on the community similar to CleanRL.
>
> After receiving your critical feedback, we made the following improvements:
>
> 1. We reviewed the code structure and removed redundant class interactions. We refactored the multi-agent part of the code to follow a single-file style. In commit https://github.com/PKU-Alignment/Safe-Policy-Optimization/commit/c1b8d4e9535c202f5ab1ae4c389453ca89314a7e, we deleted 6389 lines of code and made changes to 60 files. We continued to optimize the code in subsequent commits.
>
> 2. We standardized the training and evaluation processes for both single-agent and multi-agent scenarios. We unified the entire workflow, from training execution, parameter saving, and curve plotting to model evaluation, for both single-agent and multi-agent algorithms.
>
> 3. We added detailed documentation on how to [implement a new algorithm](https://safe-policy-optimization.readthedocs.io/en/latest/usage/implement.html) in SafePO and how to [quickly evaluate baseline performance](https://safe-policy-optimization.readthedocs.io/en/latest/usage/make.html) in each environment.
> We hope these improvements bring SafePO closer to the simplicity and usability associated with CleanRL. Thank you for your feedback, and we're committed to enhancing the quality of our implementation.
>
> > **(Opportunities For Improvement # 2)** Missing Description of Visual Tasks: The authors state that they developed realistic visual environment, but I could not find relevant descriptions in Appendix A.5. If these visual tasks have not been completed, it is essential to clearly indicate them as part of future work in the paper. Including details about these visual tasks and the challenges they pose would provide valuable insights into the comprehensiveness of the Safety-Gymnasium environment suite.
>
> > **(Documentation)** Missing description of visual tasks.
>
> We sincerely apologize for any confusion. The visual environment aspect has already been developed by us, though we failed to adequately clarify this in the initial paper. In the revised version, we have enhanced the description of our visual environment in Section 4.1.1 of the main text. We have also included both RGB and RGB-D visual inputs (depicted in Figure 4 and Figure 5 in the paper). Additionally, we have provided more comprehensive details in Appendix B.5 and have elaborated further on the vision-only task in our online documentation.
> Vision tasks in Online Documentation: https://www.safety-gymnasium.com/en/latest/environments/safe_vision.html

---

> ### Author Response · Authors · 2023-08-21
> **Rebuttal by Authors (2/5)**
>
> > **(Opportunities For Improvement # 3)** Baseline Algorithm Performance and Consistency: The experimental section presents various baseline algorithms, but there seem to be substantial oscillations in satisfying constraints. This disparity in constraint satisfaction contradicts previous literature and raises concerns about the efficacy and consistency of SafePO's implementation of these algorithms. It is crucial for the authors to address these inconsistencies and provide a thorough analysis to ensure the reliability and accuracy of their SafeRL algorithms.
>
> > **(Additional Feedback # 1)** The paper seems only to provide the performance of the algorithms on their implementation but does not compare it to the performance achieved by other well-known implementations of the same algorithms.
>
> Your feedback is highly meaningful. Regarding the performance and consistency of the baseline algorithm, we've invested substantial effort. During the rebuttal phase, we heeded your suggestion and introduced comparisons with other publicly available experiments.
>
> To ensure the benchmark's **accuracy and reliability**, we've taken careful steps. **Firstly**, we meticulously replicate each algorithm, strictly adhering to the original paper's specifications. This involves ensuring that the gradient flow aligns with the original paper's methodology and similar considerations. **Secondly**, in cases where established open-source codebases exist for certain algorithms, we conduct meticulous line-by-line comparisons between our implementation and those existing codes. This thorough examination serves to cross-verify the accuracy of our work. **Finally**, we subject SafePO to comparisons with established benchmarks like Safety-Starter-Agents[1] and RL-Safety-Algorithms[2]. The results indicate that SafePO surpasses the performance of other existing implementations, affirming its effectiveness as a benchmark. And we provide their training curves under the same settings: https://safe-policy-optimization.readthedocs.io/en/latest/algorithms/comparision.html#.
>
> [1] Safety-Starter-Agents: https://github.com/openai/safety-starter-agents
>
> [2] RL-Safety-Algorithms: https://github.com/SvenGronauer/RL-Safety-Algorithms
>
> > **(More discussions)** This disparity in constraint satisfaction contradicts previous literature and raises concerns about the efficacy and consistency of SafePO's implementation of these algorithms.
>
> > **(Additional Feedback # 1)** The paper seems only to provide the performance of the algorithms on their implementation but does not compare it to the performance achieved by other well-known implementations of the same algorithms.
>
> After conducting our research and analysis, we observed certain disparities between our replication results and those reported in previous literature. These differences pertain to aspects of performance and cost convergence. Following extensive discussions and meticulous examination, we have concluded that these differences primarily stem from inconsistencies in the environment. **During the process of designing our simulation environment, we gained a crucial insight that the initial states of diverse environments and the distinct approaches various simulation engines take towards randomness may exhibit notable variations.**
>
> Furthermore, we undertook comparisons with several open-source implementations and the publicly available code from the original authors. As illustrated in Table 1, SafePO demonstrates comparable performance with other implementations. Based on these comparisons, we have reached the conclusion that, despite the challenges introduced by environmental differences, our replication work remains consistent in terms of overall performance with existing research.

---

> ### Author Response · Authors · 2023-08-21
> **Rebuttal by Authors (3/5)**
>
> > **(Limitations)** The paper primarily focuses on benchmarking SafeRL algorithms using the proposed environment suite. However, the transferability of the results to complex real-world safety-critical applications may be limited. Future work should explore the adaptability and performance of SafePO in more challenging and diverse real-world scenarios. The paper could benefit from a more extensive evaluation of the performance metrics used to assess the algorithms' safety and efficiency. Incorporating additional evaluation criteria, such as worst-case scenario analysis or robustness to adversarial perturbations, would provide a more comprehensive understanding of the SafeRL algorithms' capabilities.
>
> We deeply appreciate your valuable suggestions, "However, the transferability of the results to complex real-world safety-critical applications may be limited." We wholeheartedly agree with this perspective and have indeed incorporated it into the "Limitations" section of the revised paper. Your insights have been immensely valuable, and should the opportunity arise, we intend to express our gratitude to you (reviewer EjZD) for your constructive feedback on the acknowledgments.
>
> Additionally, we have undertaken several other revisions in line with your recommendations. Allow me to share the specifics:
>
> 1. In subsequent endeavors, SafePO will embark on relevant research pertaining to real-world safety control. Within SafePO, we have established an interface for both multi-agent and single-agent algorithms within the Isaac Gym environment. Our aim is to explore the capabilities of SafeRL in tasks of greater practical significance, such as dexterous hand manipulation and robotic arm control.
>
> 2. Our paper's Section 6 now offers a more profound analysis. We present visually insightful outcomes of SafeRL algorithms depicting constraint violations during the training process in Figure 8. Moreover, we introduce new evaluation criteria and address oscillations, while conducting a comprehensive analysis of diverse algorithm types based on experimental findings.
>
> We trust that these adjustments align well with your insightful feedback and sincerely hope that you find our revised work to be satisfactory. Your input has been instrumental in enhancing the quality and relevance of our research.

---

> ### Author Response · Authors · 2023-08-21
> **Rebuttal by Authors (4/5)**
>
> > **(Relation To Prior Work)** Authors are candid about their relation to prior works. However, the motivation for designing a new benchmark on safe RL is unclear to me.
>
> Our original motivation in designing Safety-Gymnasium was to address certain usability issues with the existing Safety-Gym. This environment has been gaining traction within the SafeRL community. Building upon this, we crafted a new architecture based on MuJoCo, encompassing all tasks of Safety-Gym, and made it open-source for the community. Presently, the SafePO and Safety-Gymnasium repository on GitHub has garnered nearly 400 stars.
>
> **Notably**, recent endeavors such as the CMU team's development of an Offline Dataset for SafeRL[1] have also employed Safety-Gymnasium as a foundation.
>
> [1] Datasets and Benchmarks for Offline Safe Reinforcement Learning
>
> Moreover, the standout feature of Safety-Gymnasium lies in its incorporation of both multi-agent safety tasks and visual safety tasks. This comprehensive integration positions it as a versatile benchmark environment for safety-oriented reinforcement learning. Throughout the implementation of this framework, we prioritized scalability, encouraging users to engage in secondary development to introduce captivating tasks tailored to SafeRL. Closely tied to our work is Safety Gym, and we have elaborated extensively on the distinctions between Safety-Gymnasium and Safety Gym, as outlined below (see **Compare to Safety Gym** (In the introduction) in the revised version of the paper):
>
> **(1). Physics Engine Upgrade.**
> While Safety Gym used `mujoco-py`, Safety-Gymnasium has been updated to use the newest `MuJoCo` 2.3 and above. This change was needed because `mujoco-py` stopped getting updates after October 18, 2021, and it no longer worked with `MuJoCo` now. This caused issues when installing Safety Gym, requiring us to think about compatibility with older versions, including downgrading Numpy and GCC.
> In contrast, Safety-Gymnasium has moved away from relying on `mujoco-py` and instead went through significant changes to take advantage of the latest features of MuJoCo (which was a big technical challenge). This simplification has made it easier to get started, as now users only need to use one simple command: `pip install safety-gymnasium` to set up the basic Gymnasium-based safety tasks. Additionally, using MuJoCo's new features has enabled us to run Safety-Gymnasium on both Mac and Windows PCs, improving how quickly environments are displayed (around 25% faster), making the visuals more accurate, and enhancing support for visual elements like importing Obj files.
>
> **(2). Diverse Agent and Task Expansion.**
>
>   - Added single-agent: Rececar. Compared with the agent car, it is more real and has a corresponding physical machine [MIT Racecar](https://racecar.mit.edu/). Reviewer w71G highly appreciated the introduction of this agent and suggested that we extend the agent to FormulaOne (Drive-Track) tasks.
>
> - Added single-agent: Ant. The original safety gym agent is mainly considered from the perspective of simplifying the difficulty. It includes three agents that are designed not to fall, so they are more suitable for exploration, but this is too simple. We introduced Ant to a higher degree of freedom which iterates and rolls during the exploration process, which can increase the difficulty of agent exploration and the diversity of data.
>
> > **Safety Gym Origin paper**: It is designed such that a uniform random policy should keep the robot from falling over and generate some travel.
>
> **(3). Extend to Multi-agent Safety Tasks.**
>
> - MaMuJoCo-based, which splits the agent in MuJoCo into multiple joints and controls it through a multi-agent algorithm. This is widely accepted in the MARL community and has been adopted by many famous algorithms, such as HATRPO/PPO[1], MARLlib[2] et al. Although this setting is rather strange, it is acceptable as a simple and quick verification of the algorithm.
>
> - Another type is the Multi-Ant scenario, where Ant can be replaced by other agents such as Point, Car, etc. Multiple agents cooperate to complete some tasks. We have implemented this task in Safety Gymnasium and added it to the document. For details, please refer to: https://www.safety-gymnasium.com/en/latest/environments/safe_multi_agent.html
>
>   [1] Trust Region Policy Optimisation in Multi-Agent Reinforcement Learning
>
>   [2] MARLlib: A Scalable Multi-agent Reinforcement Learning Library
>
> **(4). Extend to Complex and High-dimensional Scenes.**
>
> DexterousHands focuses on the Bimanual Dexterous Manipulation, and the task difficulty is even more difficult (state space dimension 350+, action space dimension 40+). We extend its task to the safe field and choose two typical constraints, Safe Joint and Safe Finger. More details can be referred to: https://www.safety-gymnasium.com/en/latest/environments/safe_multi_agent/freight_franka_close_drawer.html

---

> ### Author Response · Authors · 2023-08-21
> **Rebuttal by Authors (5/5)**
>
> > **(Additional Feedback # 2)** It appears to draw heavily from previous works[1,2] without introducing substantial new ideas. Utilizing existing work as one's own contribution does not align with the standard of research expected at NeurIPS. Originality and innovation are fundamental criteria for evaluating submissions, and direct appropriation of existing contributions undermines the integrity of the scientific process.
> [1] Towards Human-Level Bimanual Dexterous Manipulation with Reinforcement Learning.Thirty-sixth Conference on Neural Information Processing Systems Datasets and Benchmarks Track [2] Safe multi-agent reinforcement learning for multi-robot control. Artificial Intelligence
>
> We sincerely apologize for the misunderstanding. Both works [1] and [2] were carried out by the same team. In [1], the focus was on introducing a multi-agent environment without any inherent safety constraints. Building upon this foundation, we expanded the concept and introduced a distinctive heterogeneous multi-agent scenario by reconfiguring resource files, incorporating instances from the repository.
>
> Another work, [2], also stems from our team. It presents a multi-agent safe reinforcement learning algorithm with theoretical guarantees. We did not directly reuse the environment used in that work. Our key improvements include 1. Engine substitution, enabling the removal of high-intensity dependencies like GCC, and offering greater customization through environment components that were not feasible in [2]. 2. Expansion of additional task components.
>
> In essence, our aim was to create a comprehensive benchmark environment for safety-oriented reinforcement learning. In the paper, we emphasize the inspirational influence of these two works and adhere to the licenses from [1] and [2] in our open-source code. We have been maintaining the repositories SafePO and Safety-Gymnasium since last year, amassing nearly **400 stars** on GitHub. Addressing community inquiries, we have diligently responded to around **25+ issues and discussions** (see https://github.com/PKU-Alignment/safety-gymnasium/discussions/50).
>
> Respected reviewer, kindly trust our fervent commitment to the community. We have a strong passion for open-source initiatives, hoping they facilitate community growth and ease of research.

---

> ### Author Response · Authors · 2023-08-24
> **Hope to get your reply**
>
> Dear Reviewer EjZD,
>
> The deadline for review is nearing and we noticed that you have not replied to our rebuttal. We spend a great deal of effort during rebuttal to respond carefully to each of your questions, including the additional experiments, environmental design, and paper revisions. We sincerely hope you to reference both our rebuttal content and our latest revised submission, in the hope that you will consider reevaluating your assessment based upon these updates.
>
> Other references:
>
> Demo Site: https://sites.google.com/view/safety-gymnasium
>
> Safety-Gymnasium Online Documentation: https://www.safety-gymnasium.com
>
> SafePO Online Documentation: https://safe-policy-optimization.readthedocs.io

---

> > ### Comment · Reviewer_EjZD · 2023-08-28
> > **Response to Authors**
> >
> > I want to express my appreciation for the effort you put into addressing the concerns raised during the rebuttal phase. I have carefully reviewed your responses and the revisions made to the manuscript. However, I still have some reservations：
> >
> > Firstly, your comparison of the SafePO algorithm with other open-source implementations such as PPO-Lag, TRPO-Lag, CPO, and FOCOPS is thorough and commendable. However, the significant oscillations observed in the reward and constraint curves of these algorithms within the Mujoco speed limit scenario, particularly in SafePO, raise substantial concerns (FOCOPS page shows "This set of panels contains runs from a private project, which cannot be shown in this report"). The presence of these pronounced oscillations and shadows in the reward and constraint makes me hesitant to consider SafePO as a viable option for use in my future research endeavors. This includes using it as a baseline or building upon it for the development of new algorithms. I anticipate that other reviewers would also express skepticism about the robustness of my own implementation if I were to incorporate SafePO into my new paper.
> >
> > For Safety-Starter-Agents, because it directly inherits the implementation of PPO and TRPO in spinningup, it lacks some standard tricks (such as observation normalization and normalized value regression targets) and cannot perform well on many tasks (its implementation cannot be used for research comparison as mentioned in spinningup [1]). Safety-Starter-Agents can also perform well with some tricks, and I prefer to use my own re-implementation instead of SafePO in my research.
> >
> > In benchmark papers, it's important to make sure that the benchmarks are usable by researchers to validate their algorithm implementations and extensions. It's also important that other reviewers like the benchmarks. The impact of a benchmark paper goes beyond the number of stars it gets on platforms like GitHub. It should be a foundation for the community, helping with progress and agreement.
> >
> > Given these considerations, I believe it is prudent for me to maintain my current score for the paper.
> >
> > [1] https://spinningup.openai.com/en/latest/spinningup/bench.html

---

> > > ### Author Response · Authors · 2023-08-28
> > > **More Statements for Reviewer (1/2)**
> > >
> > > Dear reviewer EjZD,
> > >
> > > We haven't given up at all and we're still hoping you might reconsider. We went through each and every one of your responses carefully and provided detailed replies. We truly hope you can come to appreciate our efforts even more.
> > >
> > > > FOCOPS page shows "This set of panels contains runs from a private project, which cannot be shown in this report"
> > >
> > > We apologize for the oversight. In this rebuttal, we actually allocated a significant amount of additional computational resources and conducted a considerable number of aligned experiments. The comparison report with FOCOPS has now been published. See: https://safe-policy-optimization.readthedocs.io/en/latest/algorithms/comparision.html
> > >
> > > > However, the significant oscillations observed in the reward and constraint curves of these algorithms within the Mujoco speed limit scenario, particularly in SafePO, raise substantial concerns (FOCOPS page shows "This set of panels contains runs from a private project, which cannot be shown in this report").
> > >
> > > Dear Reviewer, regarding the oscillation phenomenon, here's our explanation:
> > >
> > > First off, we took great care to ensure that our implementation closely followed the original paper or official codebase. Additionally, we did a line-by-line comparison with the Safety-Starter-Agent to ensure consistency. As for the oscillation issue, it's not an inherent problem with our baseline implementation. For example, many existing Lagrangian-based methods use a simple Lagrange multiplier update like this:
> > > \begin{split}
> > > &initial: \nu = 0 \\\\
> > > &\nu = max(0, \nu + \alpha * (J_{c} - b))
> > > \end{split}
> > >
> > > where $\alpha$ represents the update step size for the Lagrange multiplier.
> > >
> > >
> > > The oscillation in the algorithm mostly occurs during this step. To address this, we made some adjustments. In most publicly available implementation, $\nu$ is initialized as 0 ([FOCOPS](https://github.com/ymzhang01/focops/blob/main/focops_main.py#L299)) or 1 ([Safety-Starter-Agent](https://github.com/openai/safety-starter-agents/blob/master/safe_rl/pg/run_agent.py#L39)), and $\alpha$ is set to 0.05 ([Safety-Starter-Agent](https://github.com/openai/safety-starter-agents/blob/master/safe_rl/pg/run_agent.py#L40C36-L40C36)) or 0.01 ([FOCOPS](https://github.com/ymzhang01/focops/blob/main/focops_main.py#L301)).
> > >
> > > For instance, a smaller $\alpha$ like in Safety-Starter-Agent results in more stable learning, which means, as you pointed out, less oscillation. However, this also means that if the initial policy is infeasible, it might struggle to converge quickly to a feasible solution. We also experimented with increasing $\alpha$, which causes more oscillation in policy learning. The good news is that when the initial policy is infeasible, it converges to a valid solution faster.
> > >
> > > Additionally, **some effective standard tricks** to mitigate excessive oscillation include:
> > >
> > > 1. Setting an upper bound for the Lagrange multiplier, which has been adopted in FOCOPS[1] and CUP[2], where $\nu_{max}$ is a hyperparameter, typically set to 2.0.
> > > \begin{split}
> > > &initial: \nu = 0 \\\\
> > > &\nu = max(0, \nu + \alpha * (b - J_{c})) \\\\
> > > &\nu = min(\nu, \nu_{max})
> > > \end{split}
> > >
> > > [1] First Order Constraint Optimization in Policy Space (NeurIPS 2020)
> > > [2] Constrained Update Projection Approach to Safe Policy Optimization (NeurIPS 2022)
> > >
> > > 2. Using a more stable Lagrange multiplier update method, such as **combining PID and Lagrangian updates**, as in PID_lagrangian [3]. This method stabilizes the update process by incorporating historical values of the Lagrange multiplier. We've implemented this method in SafePO as well: [SafePO: PID-Lagrangian](https://github.com/PKU-Alignment/Safe-Policy-Optimization/blob/main/safepo/common/lagrange.py#L108).
> > >
> > > [3] Responsive Safety in Reinforcement Learning by PID Lagrangian Methods
> > >
> > > 3. Other standard tricks, like freezing Lagrange multiplier updates for the initial epochs of training and then gradually introducing them, can also enhance stability.
> > >
> > > When determining the default values for our baseline implementation, our priority was for the final converged policy to be safe. Thus, we slightly increased the weight on minimizing algorithm oscillation in SafePO. And there's another reason, just like what you mentioned about the **MuJoCo Speed Limit Scenario.** Unlike the usual Safety-Gym style tasks, this environment has really high rewards on its own (the best reward can go over 8000), plus there's quite a bit of randomness involved. Based on this, the current SafeRL learning on this environment also tends to be quite shaky, and this kind of shakiness isn't unique to SafePO alone.
> > >
> > > We've also looked into our data (https://safe-policy-optimization.readthedocs.io/en/latest/algorithms/comparision.html), and the oscillation seems to be within a reasonable range.

---

> > > > ### Author Response · Authors · 2023-08-28
> > > > **More Statements for Reviewer (2/2)**
> > > >
> > > > (continue)
> > > >
> > > > To sum up, our implementation adheres to the logic of the referenced original paper and, where available, the author's open-source codebase. The oscillation issue you've raised, to some extent, stems from the challenge of finding the right balance between stability and swift convergence to valid solutions, which is a difficult trade-off. We've provided flexible interfaces that allow users to incorporate their own insights while using the algorithm, be it as a baseline or as a foundation for new methods.
> > > >
> > > > > This includes using it as a baseline or building upon it for the development of new algorithms. I anticipate that other reviewers would also express skepticism about the robustness of my own implementation if I were to incorporate SafePO into my new paper.
> > > > > For Safety-Starter-Agents, because it directly inherits the implementation of PPO and TRPO in spinningup, it lacks some standard tricks (such as observation normalization and normalized value regression targets) and cannot perform well on many tasks (its implementation cannot be used for research comparison as mentioned in spinningup [1]). Safety-Starter-Agents can also perform well with some tricks, and I prefer to use my own re-implementation instead of SafePO in my research.
> > > >
> > > > Dear reviewer, allow me to politely explain the positioning of the SafePO, just as you mentioned. We're not claiming that the hyper-parameters we provide are absolutely perfect, or that we've incorporated every potential standard trick. What we're really offering is a flexible framework that lets researchers make modifications based on the existing setup to better suit your specific needs.
> > > >
> > > > **Other comments:**
> > > >
> > > > From your response, we can sense that Reviewer EjZD is also involved in SafeRL research. We're not denying the significant role that Safety-Gym and Safety-Starter-Agent have played in the SafeRL community. In fact, when I first started delving into SafeRL, I heavily relied on these two open-source repositories. I'm quite familiar with them.
> > > >
> > > > Our motivation for this work stems from the fact that Safety-Gym and Safety-Starter-Agent haven't been updated for the past 4 years. This has been a concern for our collaborative research group and others familiar with SafeRL research. The lack of updates has caused issues with dependencies versions, etc. It's not our claim that Safety-Gymnasium and SafePO can replace Safety-Gym and Safety-Starter-Agent. **What's most suitable and familiar is what matters most.** Through our efforts, we hope to provide another option for those engaged in SafeRL research.
> > > >
> > > > I'm not sure if the previous reply was what you were hoping for, but we're really hoping you could consider giving us a positive score. We currently don't have time to do more hyper-parameter analysis of vibration, if the reviewers believe us, we will add a very detailed analysis of vibration in the camera-ready version. If there are any more questions or things you're wondering about, we're totally up for discussing them with you. We'll be keeping an eye out for any messages from you until the review deadline.
> > > >
> > > > After the first round of rebuttal, **Reviewer vCTw acknowledged our work and bumped the score up from 5 to 8. They also mentioned that they are using Safety-Gymnasium for their own research.** Here are their comments:
> > > >
> > > > > Thank you to the authors for their detailed response, additional experimental results, and revisions to the paper. I have utilized this environment for some time in various projects, and it has generally served me well. The rich environments and env wrappers provided, especially the one converting to the standard Gymnasium API, significantly simplify the integration of SafetyGymnasium with my own safe RL library. The authors seem to be pretty active and responsible for maintaining the codebase. I deeply appreciate your efforts.
> > > >
> > > > > My primary concerns previously were about the presentation, specifically the absence of a detailed comparison between the improvements of SafetyGymnasium over the original Safety Gym. These concerns have now been adequately addressed.
> > > >
> > > > > Overall, I believe that the provided environment is an invaluable contribution to the safe RL community, which will likely spur further research in this field. Consequently, I have raised my rating and hope to see this paper accepted.

---

> > > > > ### Comment · Reviewer_EjZD · 2023-08-28
> > > > >
> > > > > As a result of your diligent revisions and responses, I have decided to increase my score for the paper. However,  I strongly recommend including detailed information about the optimal hyperparameters used for each algorithm in the baseline tasks in the camera-ready version of your paper. This addition would greatly assist future researchers who aim to build upon your work and leverage your benchmark setup.

---

> > > > > > ### Author Response · Authors · 2023-08-28
> > > > > > **Thank you for your understanding and recognition**
> > > > > >
> > > > > > Thank you very much for your understanding and recognition. Please trust us, your valuable comments and feedback will be presented in the final version.

---

> ### Comment · Area_Chair_stYK · 2023-08-24
> **Check author's response**
>
> Hi Reviewer,
>
> Can you please come back and check the author's response? Thanks.
>
> Best,
> AC

---

### Official Review · Reviewer_7227 · 2023-07-20
**Review of Safety Gynnasium**

**Rating:** 6
**Confidence:** 4
**Correctness:** Yes
**Clarity:** Yes

**Strengths:**

1. The suite of environments are diverse
2. The implementations of algorithms are clean and correct
3. The experiment results are extensive

**Additional Feedback:**

Overall, the paper addresses an essential topic concerning safe policy optimization, and it presents a comprehensive benchmark to evaluate various algorithms. However, several improvements are needed to enhance the clarity and usefulness of the paper. I think the environments and algorithms are not well organized. As this is a dataset and benchmark track paper, we evaluate the paper not largely on the technical contributions, but in a way about the clearness and in-depth insights for the community.

1. Improved Environment Selection Justification:
The paper lacks sufficient justification for the choice of environments in the benchmark. Rather than presenting a well-reasoned selection process, it appears that the environments were collected merely based on availability. To strengthen the paper's contribution, the authors should identify key aspects relevant to safe policy optimization and then carefully select environments that represent these aspects. This way, the benchmark will have more meaningful and insightful characterizations, aiding researchers in assessing algorithm performance under specific conditions.

2. Comprehensive Analysis of Algorithm Results:
While the authors present results for various algorithms, the analysis in both the main paper and the appendix is insufficient. A more thorough analysis that aligns with the identified key aspects in safe policy optimization should be provided. This will allow users of the benchmark to gain deeper insights into algorithm performance and better understand their strengths and limitations under different scenarios.

3. Inclusion of Failure Case Discussions:
To make the benchmark even more valuable for beginners and researchers in safe policy optimization, I strongly suggest the authors include discussions on failure cases. Understanding failure cases is crucial as they reveal potential areas of unsafety and help identify algorithm weaknesses. By addressing both successful and unsuccessful cases, the benchmark will provide more comprehensive insights, guiding future algorithm development towards greater safety.

Overall, addressing the above points will significantly enhance the paper's impact and value within the research community. The revisions will better support researchers in their quest for safe policy optimization and foster advancements in the field.


————————
Thanks for the author's response. I still feel that the selection of the environments is not well motivated. As a test benchmark, we should make the evaluation as comprehensive as possible. Therefore, a high-level categorization of the different safety constraints is needed, the current justification is not convincing to me. But overall I think the authors address some concerns and this is a good contribution. so I would increase my score to 6.

**Documentation:**

The benchmark is well documented.

**Limitations:**

No. I do not think the authors discuss the limitations of the work. I would suggest the authors to add a detailed discussion about the limitations and the ways to mitigate it.

**Opportunities For Improvement:**

I appreciate the authors' efforts for the suite of environments and algorithms. But I just feel the environments and algorithms are not well organized and more insights can be included. (More details in the additional feedback)

**Relation To Prior Work:**

Yes

**Summary And Contributions:**

This paper provides a suite of environments and a set of implementations of algorithms for safe policy optimization. The environments provided in this paper is diverse and the implementations are clean and correct. Extensive experiments show the correctness of the benchmarks.

---

> ### Author Response · Authors · 2023-08-21
> **Rebuttal by Authors (1/2)**
>
> We greatly appreciate your patience reply. We are also grateful for your recognition of our efforts and the positive evaluation of our work. Once again, we extend our gratitude for your valuable feedback. Your suggestions are of great value in enhancing the quality of our work.
>
> > **Re: (Opportunities For Improvement # 1)** I appreciate the authors' efforts for the suite of environments and algorithms. But I just feel the environments and algorithms are not well organized and more insights can be included. (More details in the additional feedback)
>
> I sincerely apologize for causing any inconvenience in your reading experience. To address your concerns, we've taken steps to enhance the structure of the environment and algorithm organization. Furthermore, we've included additional experimental results and in-depth algorithm analysis. Your feedback regarding the **additional feedback** has been carefully considered. We've taken measures to address all the points you've raised, and we encourage you to take a look at our revised paper. Our aim is that these revisions will help you better appreciate our work and potentially update your score.
>
> > **(Limitations # 1)** I do not think the authors discuss the limitations of the work. I would suggest the authors to add a detailed discussion about the limitations and the ways to mitigate it.
>
> Thank you very much for your suggestion. We have added a paragraph of limitation at the end of the main paper and marked it in orange.
>
> > **(Additional Feedback # 1)** Improved Environment Selection Justification: The paper lacks sufficient justification for the choice of environments in the benchmark. Rather than presenting a well-reasoned selection process, it appears that the environments were collected merely based on availability. To strengthen the paper's contribution, the authors should identify key aspects relevant to safe policy optimization and then carefully select environments that represent these aspects. This way, the benchmark will have more meaningful and insightful characterizations, aiding researchers in assessing algorithm performance under specific conditions.
>
> Your question is very insightful. As you said, we should identify key aspects relevant to safe policy optimization and then carefully select environments that represent these aspects. The selection of each task in this work has been carefully considered. Some of our related reasons are as follows (these contents are not mentioned in the paper, we will add them later in our online documentation):
>
> (1) **Introducing the Velocity task**, we integrated it into existing intelligent agents within the MuJoCo environment. The velocity constraint is, in fact, a subset of the reward components. This constraint introduces feasibility problems (https://en.wikipedia.org/wiki/Mathematical_optimization#Feasibility_problem), stemming from the fact that the objective is roughly proportional to the cost. This constraint limits the maximum achievable reward in the problem, essentially treating the reward as a constant. Feasibility problems essentially serve as sanity checks for constrained reinforcement learning (RL) problems.
>
> (2) **Introducing the Circle task**, we positioned it as a foundational task to validate algorithm effectiveness. In fact, when designing safety reinforcement learning algorithms, we used this environment for simple validation. The constraint task is well-detailed in CPO (Conservative Policy Optimization) and has inspired a series of subsequent studies on safety RL algorithms. CPO's citation count has surpassed 1000+, making this environment particularly significant for researchers in SafeRL.
>
> (3) **Introducing the Push, Button, and Goal tasks** occurred systematically in the Safety-Gym article and has been widely embraced by the SafeRL community. Numerous meaningful works have been designed based on these three tasks. Incorporating these tasks offers researchers a smooth transition from using Safety Gym to our environment.
>
> (4) **The inclusion of joint constraints for the dexterous hands** is closely related to challenges encountered in our real-world environment. Our research team has acquired physically dexterous hands and found that strategies excelling in the simulation environment often lead to significant damage due to excessive control during practical operations. Considering this issue, we introduced the concepts of "SafetyJoint" and "SafetyFinger" for the "Dexterous Hands." As the experiments progressed, we realized the immense practical significance of SafeRL. This approach has demonstrated significant potential in real-world experiments, offering a promising solution to risks associated with excessive control and ensuring the safety of physical mechanical entities.
>
> > **The Picture of  Constraints in the  Physical Machine DexterousHands**: https://sites.google.com/view/safety-gymnasium#h.n9jzi7iqif7t

---

> ### Author Response · Authors · 2023-08-21
> **Rebuttal by Authors (2/2)**
>
> > **(Additional Feedback # 2)** Comprehensive Analysis of Algorithm Results: While the authors present results for various algorithms, the analysis in both the main paper and the appendix is insufficient. A more thorough analysis that aligns with the identified key aspects in safe policy optimization should be provided. This will allow users of the benchmark to gain deeper insights into algorithm performance and better understand their strengths and limitations under different scenarios.
>
> Thank you very much for your valuable feedback. Regarding a more detailed algorithm analysis, we have made the following incremental and improved efforts:
>
> 1. We have expanded the experimental scope by conducting experiments in a broader range of environments. For the single-agent navigation task, we increased the agents from the original two (Point and Car) and tasks from three (Button, Circle, Goal) to five agents and four tasks. Each experiment was evaluated using at least three random seeds for performance validation. In our paper's Section 6, we present a table with results from the additional experiments for comparison. Complete results are available in Appendix A. Training curves can be found in Appendix A and the online documentation. Additionally, we have openly shared all original experimental data for further community analysis and utilization: https://drive.google.com/file/d/1jlfqvcKHRXkEi--xrrSFcl8an9w7YDG0/view?usp=sharing.
>
> **Online Documentation**: https://safe-policy-optimization.readthedocs.io/en/latest/algorithms/curve.html
>
> 2. We conducted further analysis of the algorithm's performance across three dimensions: Reward, Cost, and Oscillation, during the training process. For specific details, you can refer to Figure 8 in the paper. With these three indicators and more comprehensive experimental data, we offer a more detailed analysis in Section 6 (Experiments and Analysis), including a comparison between naive Lagrangian and PID-controlled Lagrangian methods, as well as a comparison between safety projection and Lagrangian methods. Based on these experimental analyses, we provide reference recommendations tailored to users' safety requirements for the environment.
>
> > **(Additional Feedback # 3)** Inclusion of Failure Case Discussions: To make the benchmark even more valuable for beginners and researchers in safe policy optimization, I strongly suggest the authors include discussions on failure cases. Understanding failure cases is crucial as they reveal potential areas of unsafety and help identify algorithm weaknesses. By addressing both successful and unsuccessful cases, the benchmark will provide more comprehensive insights, guiding future algorithm development towards greater safety.
>
> Thank you very much for your feedback. We have incorporated your suggestions into the revised version, which now includes more detailed analyses. Based on experimental analyses of the Safety-Gymnasium tasks, the SafeRL algorithm demonstrates satisfactory average performance in the velocity task, aligning with safety requirements. However, the navigation task presents instances of failure due to its inherent randomness. This randomness induces oscillations during the training of the SafeRL algorithm, resulting in notable performance disparities across diverse tasks.
>
> In Section 6 of the paper, titled "Randomness and Oscillation," we initiate our exploration from a task-oriented perspective, examining the interplay between randomness and oscillation. Proceeding to an algorithmic viewpoint, we delve into the distinctions between Lagrangian-based methods, Projection-based methods, and PID-Lagrangian-based methods concerning constraint satisfaction and task completion. Moreover, we furnish analyses pertaining to failure cases as encountered.
>
> Drawing from empirical data obtained via experiments on the Safety-Gymnasium platform, we present the distribution of failure cases for each algorithm in Figure 8 of Section 6. Tables 2 and 3 highlight the outcomes of failure, success, and optimal results achieved by the SafeRL algorithm across the Safety-Gymnasium task suite. These comprehensive insights are intended to guide subsequent researchers in their pursuits.

---

> ### Author Response · Authors · 2023-08-24
> **Hope to get your reply**
>
> Dear Reviewer 7227,
>
> The deadline for review is nearing and we noticed that you have not replied to our rebuttal. We spend a great deal of effort during rebuttal to respond carefully to each of your questions, including the additional experiments, environmental design, and paper revisions. We sincerely hope you to reference both our rebuttal content and our latest revised submission, in the hope that you will consider reevaluating your assessment based upon these updates.
>
> Other references:
>
> Demo Site: https://sites.google.com/view/safety-gymnasium
>
> Safety-Gymnasium Online Documentation: https://www.safety-gymnasium.com
>
> SafePO Online Documentation: https://safe-policy-optimization.readthedocs.io

---

> ### Comment · Area_Chair_stYK · 2023-08-24
> **Check Author's response**
>
> Hi Reviewer,
>
> Can you please come back and check the author's response and revision?
>
> Best,
> AC

---

### Official Review · Reviewer_vCTw · 2023-07-20

**Rating:** 8
**Confidence:** 4
**Correctness:** Seems correct
**Clarity:** Yes

**Strengths:**

(1) The benchmark contains a diverse and rich set of safety-critical tasks, constraints, and robot models that require solving challenging SafeRL problems. This diversity enables rigorous evaluation.

(2) The procedural generation and Gymnasium framework allow for easy customization and expansion to new tasks, making this a flexible long-term platform.

(3) Seamless installation, comprehensive documentation, and simple usage examples make Safety-Gymnasium welcoming to new users and researchers.

**Additional Feedback:**

(1) Are there plans to transfer policies trained in Safety-Gymnasium to physical robots? How might sim-to-real transfer affect safety guarantees?

(2) Could you compare Safety-Gymnasium to other SafeRL environments on unique capabilities or tasks provided? What gaps does it fill?

(3) Could the authors provide any insights on which algorithms perform best on which environment types and task constraints?

**Documentation:**

Yes

**Limitations:**

Yes

**Opportunities For Improvement:**

(1) While simulation environments are common in RL research, training only in simulation may limit the applicability and safety of learned policies in the real world.

(2) The contrast to prior SafeRL environments like Safety Gym could be expanded. What are the detailed improvements?

(3) More analysis could be provided characterizing the difficulty of different tasks and associations between algorithms and environment types.

**Relation To Prior Work:**

Detail relationship between this benchmark to the SafetyGym environment could be clarified.

**Summary And Contributions:**

This paper introduces Safety-Gymnasium, a benchmark for safe reinforcement learning (SafeRL) research. The key contribution is a diverse set of environments and safety constraints for training and evaluating SafeRL algorithms. Environments span single-agent and multi-agent scenarios, with both vector and vision-based inputs. The benchmark improves upon prior SafeRL environments like Safety Gym in several ways, including more tasks, easier usage, and better documentation. Thorough experimentation with 16 SafeRL algorithms demonstrates the benchmark's utility. Overall, Safety-Gymnasium seems like a comprehensive and inviting benchmark that can help drive SafeRL research forward.

---

> ### Author Response · Authors · 2023-08-21
> **Rebuttal by author (1/3)**
>
> Thank you for taking the time to read and review our paper! Thank you very much for your recognition of the high scalability of Safety-Gymnasium and your foresight regarding its potential as a flexible long-term research platform. This is exactly what motivates us, and we sincerely appreciate your support.
>
> Please see our responses below; we have started working on the corresponding changes (see uploaded revised version) and will incorporate all of them in the final revision. If you have any further concerns, we would be keen to address them as well.
>
> >**Re: (Opportunities For Improvement # 1)** While simulation environments are common in RL research, training only in simulation may limit the applicability and safety of learned policies in the real world.
>
> > **(Additional Feedback # 1)** While simulation environments are common in RL research, training only in simulation may limit the applicability and safety of learned policies in the real world. Are there plans to transfer policies trained in Safety-Gymnasium to physical robots? How might sim-to-real transfer affect safety guarantees?
>
> Thank you very much for your insights. Your deep understanding of this field is clearly evident. Given that RL requires interaction with the environment, it's apparent that most current research relies on simulation environments. This reliance poses challenges when it comes to applying learned policies in real-world scenarios. Our SafeRL project aims to introduce safety constraints to enhance the effectiveness of RL algorithms in real environments. The purpose behind designing this environment is primarily for the initial validation of various algorithms.
>
> Additionally, we are actively pursuing the transition of virtual experiments into real-world settings, as part of our exploration into bridging the gap between simulation and reality. Furthermore, our research group is making dedicated efforts to port specific tasks from Safety-Gymnasium onto physical machines. Through this process, we aim to validate the practical effectiveness of SafeRL algorithms.
>
> **Notes**: Additionally, the setting of the Safety Joint and Safety Finger for the safety dexterous bimanual manipulation task, as described in this paper, is derived from the lessons we learned during physical machine experiments. We encountered instances where excessive control led to the snapping of nylon cords in the fingers, prompting the need for safety constraints.
>
> **The Picture of  Constraints in the  Physical Machine DexterousHands**: https://sites.google.com/view/safety-gymnasium#h.n9jzi7iqif7t
>
> The establishment of the platform has proven to be extremely challenging, and the transfer of policies is equally complex. As a result, we found it difficult to provide an exhaustive analysis in the current paper. However, in the "Limitations and Future Works" section of the paper, we engage in an in-depth discussion of the sim-to-real environment transfer concern that you've expressed. This issue will be thoroughly addressed in the final revised version of the paper. It also represents a future direction for our work.

---

> ### Author Response · Authors · 2023-08-21
> **Rebuttal by Authors (2/3)**
>
> >**Re: (Opportunities For Improvement # 2)** The contrast to prior SafeRL environments like Safety Gym could be expanded. What are the detailed improvements?
>
> > **(Additional Feedback # 2)** Could you compare Safety-Gymnasium to other SafeRL environments on unique capabilities or tasks provided? What gaps does it fill?
>
> Compared to Safety Gym, the improvements and advancements in Safety-Gymnasium can be categorized into the following four aspects (see **Compare to Safety Gym** (In the introduction) in the revised version of the paper):
>
> **(1) Physics Engine Upgrade.**
>
> While Safety Gym used `mujoco-py`, Safety-Gymnasium has been updated to use the newest `MuJoCo` 2.3 and above. This change was needed because `mujoco-py` stopped getting updates after October 18, 2021, and it no longer worked with `MuJoCo` now. This caused issues when installing Safety Gym, requiring us to think about compatibility with older versions, including downgrading Numpy and GCC.
>
> In contrast, Safety-Gymnasium has moved away from relying on `mujoco-py` and instead went through significant changes to take advantage of the latest features of `MuJoCo` (which was a big technical challenge). This simplification has made it easier to get started, as now users only need to use one simple command: `pip install safety-gymnasium` to set up the basic Gymnasium-based safety tasks. Additionally, using `MuJoCo`'s new features has enabled us to run Safety-Gymnasium on both Mac and Windows PCs, improving how quickly environments are displayed (around 25% faster), making the visuals more accurate, and enhancing support for visual elements like importing Obj files.
>
> **(2) Diverse Agent and Task Expansion.**
>
> - **Added single-agent: Rececar.** Compared with the agent car, it is more real and has a corresponding physical machine [MIT Racecar](https://racecar.mit.edu/). Reviewer w71G highly appreciated the introduction of this agent and suggested that we extend the agent to FormulaOne (Drive-Track) tasks. The current display of the environment is as follows, which is very meaningful for visual tasks.
>
> - **Added single-agent: Ant.** The original safety gym agent is mainly considered from the perspective of simplifying the difficulty. It includes three agents that are designed not to fall, so they are more suitable for exploration, but this is too simple. We introduced Ant to a higher degree of freedom which iterates and rolls during the exploration process, which can increase the difficulty of agent exploration and the diversity of data.
>
> > **Safety Gym Origin paper:** It is designed such that a uniform random policy should keep the robot from falling over and generate some travel.
>
> > **the video of the ant's fall out:** https://user-images.githubusercontent.com/26274945/208245842-cde9b3a5-631b-4d53-8abf-278a355197d1.gif
>
> **(3). Extend to Multi-agent Safety Tasks.**
>   - **MaMuJoCo-based,** which splits the agent in MuJoCo into multiple joints and controls it through a multi-agent algorithm. This is widely accepted in the MARL community and has been adopted by many famous algorithms, such as HATRPO/PPO[1], MARLlib[2] et al. Although this setting is rather strange, it is acceptable as a simple and quick verification of the algorithm.
> Another type is the Multi-Ant scenario suggested by Review w71G, where Ant can be replaced by other agents such as Point, etc. Multiple agents cooperate to complete some tasks. We have implemented this task in Safety Gymnasium and added it to the document. For details, please refer to: https://www.safety-gymnasium.com/en/latest/environments/safe_multi_agent.html
>
>   [1] Trust Region Policy Optimisation in Multi-Agent Reinforcement Learning
>
>   [2] MARLlib: A Scalable Multi-agent Reinforcement Learning Library
>
> **(4). Extend to Complex and High-dimensional Scenes.**
>
> DexterousHands focuses on the Bimanual Dexterous Manipulation, and the task difficulty is even more difficult (state space dimension 350+, action space dimension 40+). We extend its task to the safe field and choose two typical constraints, Safe Joint and Safe Finger.
>
> > During our physical experimentation with the bimanual hands, we frequently encountered issues of excessive joint control, leading to the breakage of joint nylon ropes, as illustrated in the image below.
> > **The Picture of  Constraints in the  Physical Machine DexterousHands:** https://sites.google.com/view/safety-gymnasium#h.n9jzi7iqif7t
>
> At the same time, inspired by the automated warehouse, we have expanded on this basis. Through the combination of Fetch+Hand, we have designed a heterogeneous multi-agent collaboration scenario, more details can be referred to https://www.safety-gymnasium.com/en/latest/environments/safe_isaac_gym.html.
>
> **(5). Evidence from some discussions in the open source community.**
>
> Researchers extend a small piece of our code to implement their own environment (Combine components like Pillars Hazards Sigwalls):https://github.com/PKU-Alignment/safety-gymnasium/discussions/66

---

> ### Author Response · Authors · 2023-08-21
> **Rebuttal by Authors (3/3)**
>
> > **Re: (Opportunities For Improvement # 3)** More analysis could be provided characterizing the difficulty of different tasks and associations between algorithms and environment types.
>
> > **(Additional Feedback # 2)** Could the authors provide any insights on which algorithms perform best on which environment types and task constraints?
>
> Thank you very much for your feedback. We have incorporated your suggestions into the revised version, which now includes more detailed analyses. Based on experimental analyses of the Safety-Gymnasium tasks, the SafeRL algorithm demonstrates satisfactory average performance in the velocity task, aligning with safety requirements. However, the navigation task presents instances of failure due to its inherent randomness. This randomness induces oscillations during the training of the SafeRL algorithm, resulting in notable performance disparities across diverse tasks.
>
> In Section 6 of the paper, titled "Randomness and Oscillation," we initiate our exploration from a task-oriented perspective, examining the interplay between randomness and oscillation. Proceeding to an algorithmic viewpoint, we delve into the distinctions between Lagrangian-based methods, Projection-based methods, and PID-Lagrangian-based methods concerning constraint satisfaction and task completion. Moreover, we furnish analyses pertaining to failure cases as encountered.
>
> Drawing from empirical data obtained via experiments on the Safety-Gymnasium platform, we present the distribution of failure cases for each algorithm in Figure 8 of Section 6. Tables 2 and 3 highlight the outcomes of failure, success, and optimal results achieved by the SafeRL algorithm across the Safety-Gymnasium task suite. Based on these experimental analyses, we provide reference recommendations tailored to users' safety requirements for the environment.

---

> > ### Comment · Reviewer_vCTw · 2023-08-28
> >
> > Thank you to the authors for their detailed response, additional experimental results, and revisions to the paper. I have utilized this environment for some time in various projects, and it has generally served me well. The rich environments and env wrappers provided, especially the one converting to the standard Gymnasium API, significantly simplify the integration of SafetyGymnasium with my own safe RL library. The authors seem to be pretty active and responsible for maintaining the codebase. I deeply appreciate your efforts.
> >
> > My primary concerns previously were about the presentation, specifically the absence of a detailed comparison between the improvements of SafetyGymnasium over the original Safety Gym. These concerns have now been adequately addressed.
> >
> > Regarding the benchmarking results with SafePO, I wonder why the authors chose not to use OmniSafe. I'm slightly unclear about the parallels and distinctions between these two projects, given they are both from the same organization. Moreover, I suggest considering some of the more recent safe RL works with varied optimization styles for inclusion in the paper's discussion [1-4].
> >
> > Overall, I believe that the provided environment is an invaluable contribution to the safe RL community, which will likely spur further research in this field. Consequently, I have raised my rating and hope to see this paper accepted.
> >
> > [1] Sootla, Aivar, et al. "Sauté rl: Almost surely safe reinforcement learning using state augmentation." International Conference on Machine Learning. PMLR, 2022.
> > [2] Liu, Zuxin, et al. "Constrained variational policy optimization for safe reinforcement learning." International Conference on Machine Learning. PMLR, 2022.
> > [3] Yu, Haonan, Wei Xu, and Haichao Zhang. "Towards safe reinforcement learning with a safety editor policy." Advances in Neural Information Processing Systems 35 (2022): 2608-2621.
> > [4] Chen, Baiming, et al. "Context-aware safe reinforcement learning for non-stationary environments." 2021 IEEE International Conference on Robotics and Automation (ICRA). IEEE, 2021.

---

> > > ### Author Response · Authors · 2023-08-28
> > > **Re:  Official Comment by Reviewer vCTw**
> > >
> > > Thank you very much for your recognition and improving the score,
> > >
> > > > Regarding the benchmarking results with SafePO, I wonder why the authors chose not to use OmniSafe.
> > >
> > > Omnisafe is like a tightly wrapped package, which might not be the easiest for beginners to dive into. We've received feedback and thought it would be great to offer a simpler baseline library. So, with SafePO, we're aiming for a more streamlined approach. In this implementation, we haven't piled on a bunch of layers of complexity, and each algorithm is housed in its single file.
> > >
> > > > Moreover, I suggest considering some of the more recent safe RL works with varied optimization styles for inclusion in the paper's discussion [1-4].
> > >
> > > Thank you so much for your suggestion. We'll definitely try to incorporate these varied optimization styles algorithms as well.

---

### Official Review · Reviewer_w71G · 2023-07-26
**Review for Safety Gymnasium**

**Rating:** 5
**Confidence:** 4
**Clarity:** Yes.

**Strengths:**

Given the high importance of safe reinforcement learning and the development of new algorithms, a safe RL focused environment and algorithm library is a good contribution for rapid experimentation and benchmarking. The proposed platform contains state and vision observations (proposing both RGB and RGBD sensors  for example) and a higher variety of environments/constraints that can be programmed.

**Additional Feedback:**

N/A

**Correctness:**

Yes. The experiments seem generally well set up and do present results from multiple runs.

**Documentation:**

Yes, the repositories do seem to come with sufficient documentation on setup as well as evaluating/implementing algorithms etc.

**Limitations:**

There is no real discussion of limitations except for a few technical ones mentioned in the appendix. While it is useful to note some of the issues with dependencies, the technical details of observations etc., it would have been more informative to discuss the limitations of the tasks (i.e., do these sufficiently represent realistic tasks? Are they complex enough to push the limits of current Safe RL algorithms), as well as sensor models (sensor noise, fidelity etc.). It would also be good to discuss how easy it is to implement new algorithms  within the SafepO library.

**Opportunities For Improvement:**

I think the set of improvements over something like safety gym is not very extensive. A few things come to mind as limitations or avenues for further enhancements:

1. The visual fidelity and the complexity of tasks is fairly low. For example, the visual appearance is really not that different from Safety Gym (according to the comparison in Figure 9 in the appendix), and tasks like Run/Circle/Goal are fairly simple. There seem to be a few incremental improvements over Safety Gym with regards to stability but the delta is not very high. While constraints like Pillars, Gremlins etc. add a new dimension, conceptually they are still fairly simple constructs.

For instance, Safety Gym contains a 'racecar' agent - it would have been interesting to have a track environment where the car needs to drive and stay on the track, and potentially avoid hitting some cones at the boundaries of the racetrack etc. This would have introduced some more realistic tasks that could be challenging for vision-only RL algorithms.

2. It is unclear why agents like ant, half cheetah are being treated as multi-agent scenarios. If anything, this is just a higher state space to control, whereas something like the two-hands controlling a ball is a good multi agent scenario. It might have been better to introduce some actual multi agent scenarios similar to robocup, or maybe a simpler cooperative environment for multiple ants etc.

3. It is also unclear how well the RGBD observations work. Is the depth a pixel space observation corresponding to the RGB, or is it more sparse? There is a mention of a 'natural LiDAR" in the appendix, but it does not seem to appear in the main paper.

**Relation To Prior Work:**

It is not very clear how much of a significant contribution this platform brings over Safety Gym and its equivalents. Primarily it would have been good to shed light on why the contributions discussed in the paper require an entirely new library as opposed to making changes / additions on top of Safety Gym.

**Summary And Contributions:**

In this paper, the authors present a Safe RL benchmark platform named Safety Gymnasium. Specifically, Safety Gymnasium is composed of A) a suite of environments for safe RL single and multi-agent tasks, and B) a set of algorithms comprised of 16 state of the art RL algorithms. Safety Gymnasium is aimed to be a Gym-equivalent, with an emphasis on easy setup; and also contains a more diverse set of constraints and visually more appealing environments, along with several types of agents. On the SafePO side, the library of algorithms contain TRPO and PPO styled algorithms integrated with Tensorboard/WandB for easy analysis and visualization.

---

> ### Author Response · Authors · 2023-08-21
> **Rebuttal by Authors (1/5)**
>
> Thank you for taking the time to read and review our paper! Please see our responses below and we have started working on the corresponding changes (see uploaded revised version) and will incorporate all of them in the final revision. If you have any further concerns, we would be keen to address them as well.
>
> > **(Opportunities For Improvement #1)** The visual fidelity and the complexity of tasks is fairly low. For example, the visual appearance is really not that different from Safety Gym (according to the comparison in Figure 9 in the appendix), and tasks like Run/Circle/Goal are fairly simple. There seem to be a few incremental improvements over Safety Gym with regards to stability but the delta is not very high. While constraints like Pillars, Gremlins etc. add a new dimension, conceptually they are still fairly simple constructs.
>
> > For instance, Safety Gym contains a 'racecar' agent - it would have been interesting to have a track environment where the car needs to drive and stay on the track, and potentially avoid hitting some cones at the boundaries of the racetrack etc. This would have introduced some more realistic tasks that could be challenging for vision-only RL algorithms.
>
> According to your suggestion, we have made the following adjustments:
>
> 1. Add the task of Racecar
>
>   We have created the FormulaOne task series based on your intriguing insights. Ideally, this task requires the agent to navigate through the entire racetrack, avoiding walls and obstacles accurately. The complexity of the environment and its visual components pose elevated demands on the algorithm. Moreover, this task is versatile, allowing for the substitution of the Racecar with other intelligent agents such as Point, Car, and Ant. For more specific details, please refer to the documentation at https://www.safety-gymnasium.com/en/latest/environments/safe_vision/formula_one.html, and you can also watch our provided video demo for a clearer understanding.
>
>   **The Video Demo URL of FormulaOne Task**: https://sites.google.com/view/safety-gymnasium#h.tlzuhrzbqrur
>
> 2. Now the tasks are diverse, among which Level2 is challenging
>
> Safe Navigation focus on designing environment suites suitable for most hardware to serve the community to quickly verify and compare algorithms. Increasing the complexity of the task results in greater performance overhead and may require additional engineering of the algorithm to obtain meaningful policies on it. In addition, existing tasks such as Run/Circle/Goal, although tasks like Circle0, Goal0 appear too simple, by introducing more obstacles and greater randomness (e.g., from SafetyAntGoal0 to SafetyAntGoal2, the state space Increased from 56 to 88, the random range of positions doubled, the randomly initialized objects increased from 2 types (Agent, Goal) to 4 types (Agent, Goal, Hazards, Vases), and the constraints changed from 0 to 3 types (hazards_area, vases_contact, vases_velocity), the number of objects increased from 2 to 22. For more details, please refer to the document https://www.safety-gymnasium.com/en/latest/environments/safe_navigation/goal.html).
>
> 3. The state-action dimension of DexterousHand based on Issac-gym is very large.
>
> For example, the number of state observations of ShadowHands is 398, and the number of actions can reach 52. The constraints on its joints come from the insights we obtained during the operation of the real machine.
> During our physical experimentation with the bimanual hands, we frequently encountered issues of excessive joint control, leading to the breakage of joint nylon ropes, as illustrated in the image below.
> The Picture of  Constraints in the  Physical Machine DexterousHands: https://sites.google.com/view/safety-gymnasium#h.n9jzi7iqif7t
>
> 4. Added many new task meshes to improve their realism, which is of great benefit to the vision task
>
> In Safe Vision, we have also added the Building series of tasks, which involve the sub-tasks of the agent avoiding moving robots in the construction site, navigating to the designated location, manipulating the machine, and moving objects. Simple tasks are supplemented at the level of visual complexity and environmental complexity. For more information, please refer to our documentation https://www.safety-gymnasium.com/en/latest/environments/safe_vision.html and online demo: https://sites.google.com/view/safety-gymnasium#h.tlzuhrzbqrur

---

> ### Author Response · Authors · 2023-08-21
> **Rebuttal by Authors (2/5)**
>
> > **(Opportunities For Improvement #2)** It is unclear why agents like ant, half cheetah are being treated as multi-agent scenarios. If anything, this is just a higher state space to control, whereas something like the two-hands controlling a ball is a good multi agent scenario. It might have been better to introduce some actual multi agent scenarios similar to robocup, or maybe a simpler cooperative environment for multiple ants etc.
>
> Thank you very much for your comments on the revisions. In order to make the multi-agent environment more meaningful, we have made the following two modifications according to your comments:
>
>   1. Inspired by the actual scene, we combined Freight + Hand and designed the safety constraint task of pick and place. This is a very meaningful heterogeneous multi-agent environment with a higher state space.
>
>   2. We also designed a simple multi-ant cooperation environment. The agent can be an Ant or other agents such as Point and Racecar.
> Regarding the MaMuJoCo-based environment, it splits the agent within the MuJoCo framework into multiple joints, subsequently governing it through a multi-agent algorithm. This approach has garnered widespread acceptance within the MARL (Multi-Agent Reinforcement Learning) community, finding its integration into numerous renowned algorithms like HAPPO and MARLlib, among others. While this configuration might appear unconventional at first glance, it serves as a simple and expedient  method for the verification of the algorithm.
>
> **The demo can be seen in**: https://sites.google.com/view/safety-gymnasium#h.ckao1y2dt9hh and https://sites.google.com/view/safety-gymnasium#h.fturpu4dmqhj
>
> > **(Opportunities For Improvement #3)** It is also unclear how well the RGBD observations work. Is the depth a pixel space observation corresponding to the RGB, or is it more sparse?
>
> I apologize for any confusion caused. We have incorporated additional information regarding the RGBD input style and other inputs in the main text. This can be observed in Figure 4 of the main paper. Additionally, we have provided demonstrations for both RGB and RGBD inputs for the same scenario. Demo can be found in: https://sites.google.com/view/safety-gymnasium#h.w9lcup3877w
>
> In comparison to RGB, RGBD is akin to grayscale images. Each pixel value in RGBD represents the actual distance of an object from the sensor. Generally, RGB and RGBD are aligned, maintaining a one-to-one correspondence between pixels. By utilizing depth maps, we can reconstruct a three-dimensional point cloud structure of the scene for each observation. This capability holds significant importance in various applications requiring dexterous manipulation [1].
>
> [1] RLAfford: End-to-End Affordance Learning for Robotic Manipulation
>
> [2] http://www.open3d.org/docs/latest/tutorial/Basic/rgbd_image.html
>
> > **(Opportunities For Improvement #3)** There is a mention of a 'natural LiDAR" in the appendix, but it does not seem to appear in the main paper.
>
> Natural LiDAR is implemented using MuJoCo's own ray mechanism, which is closer to the real radar mechanism, while the information of pseudo lidar is calculated based on the relative relationship between the object and the agent by traversing the position of the object. For the tasks of the Safe Navigation series, there is not much difference between the two, because the agent mainly navigates and explores on one plane. One difference worth mentioning is that when an object moves out of the observation field of view of a certain lidar, it is manually For some smoothing mechanisms designed, pseudo lidar will provide smooth transition information, while natural lidar will generate jump signals, which will have some impact on the learning of neural networks. In addition, the information provided by lidar can be regarded as a dimensionality reduction of the information in RGB-D. For example, the distance information provided by natural lidar is essentially the same as the depth information in RGB-D, but the distance information of lidar is usually more sparse.
>
> This is the part about lidar in our documentation https://www.safety-gymnasium.com/en/latest/components_of_environments/objects.html#lidar-mechanism

---

> ### Author Response · Authors · 2023-08-21
> **Rebuttal by Authors (3/5)**
>
> > **(Limitations #1)** There is no real discussion of limitations except for a few technical ones mentioned in the appendix. While it is useful to note some of the issues with dependencies, the technical details of observations etc., it would have been more informative to discuss the limitations of the tasks (i.e., do these sufficiently represent realistic tasks?
>
> Thank you very much for your suggestion. We have added a paragraph of limitation at the end of the main paper and marked it in orange.
>
> > **(Limitations #2:)** Are they complex enough to push the limits of current Safe RL algorithms), as well as sensor models (sensor noise, fidelity etc.).
>
> The randomness of our existing tasks (the positions and angles of various objects and agents), the complexity of multiple objects, and multiple constraints really challenge the capabilities of current algorithms. More information can refer to table 4 in Appendix A.3
>
> For example, Safe Navigation focus on designing environment suites suitable for most hardware to serve the community to quickly verify and compare algorithms. Increasing the complexity of the task results in greater performance overhead and may require additional engineering of the algorithm to obtain meaningful policies on it. In addition, existing tasks such as Run/Circle/Goal, although tasks like Circle0, Goal0 appear too simple, by introducing more obstacles and greater randomness (e.g., from SafetyAntGoal0 to SafetyAntGoal2, the state space Increased from 56 to 88, the random range of positions doubled, the randomly initialized objects increased from 2 types (Agent, Goal) to 4 types (Agent, Goal, Hazards, Vases), and the constraints changed from 0 to 3 types (hazards_area, vases_contact, vases_velocity), the number of objects increased from 2 to 22. For more details, please refer to the document https://www.safety-gymnasium.com/en/latest/environments/safe_navigation/goal.html).
>
> Considering the significance of promoting the transfer of research to the real world, we introduce realistic complex robots in the Safe Isaac Gym category and introduce more realistic constraints.
>
> From the perspective of certain specific research domains, both Safe Vision and Safe Multi-Agent pose challenges to advancing the field of Safe Reinforcement Learning (SafeRL). For instance, within the context of Safe Multi-Agent scenarios, the case of multiple ants presents a more practically meaningful endeavor compared to scenarios involving the partitioning of a single robot. Additionally, building upon the exploration of heterogeneous multi-agent interactions on FreightFranka, we aspire to introduce novel insights to the multi-agent community.
> Considering the practical significance of these tasks, Safe Isaac Gym introduces intricate robotic environments, while Safe Vision extends algorithms tailored for vector-based information input from simple geometric settings to more intricate real-world scenes, such as FormulaOne. Furthermore, we incorporate more complex object modeling for a considerable number of tasks, leading to more intricate collisions and an expansion of the task scope, exemplified by the Building series.
>
> > **(Limitations #3)** as well as sensor models (sensor noise, fidelity etc.).
>
> Regarding the simulation fidelity of sensors, our current sensor categories encompass lidar, touch, joint position, joint velocity, accelerometer, velocimeter, gyro, magnetometer, and more. Many of these sensors rely on the support provided by MuJoCo. Thanks to MuJoCo's capabilities, much of the low-level information from various physics engines can be transformed into the corresponding sensor format, which is then made accessible to users. This advancement greatly facilitates simulating real-world robots, as we no longer need to directly extract global information from the physics engine as observational input. Instead, we can encapsulate this information within the sensor observations, reflecting it in the form of localized robot observations. This not only enhances realism but also more closely evaluates the algorithm's ability to extract information. When combined with the inherent randomness prevalent in tasks, we believe this setup better scrutinizes the algorithm's practical generalization capabilities.
>
> For a comprehensive list of sensor types supported by MuJoCo, please refer to their documentation: https://mujoco.readthedocs.io/en/latest/modeling.html#sensors. As for the fidelity of a specific sensor, within the physics engine, it can be considered to be faithfully represented. To simulate the inaccuracy of real-world sensors, noise can be introduced, but currently, this feature is disabled. This is because the presence of randomness and multiple constraints already poses a significant challenge for algorithms, particularly those not emphasizing noise robustness.

---

> ### Author Response · Authors · 2023-08-21
> **Rebuttal by Authors (4/5)**
>
> > **(Limitations)** It would also be good to discuss how easy it is to implement new algorithms within the SafepO library.
> Thank you very much for your suggestion. It is very important to improve the quality of our paper, we restructured the SafePO documentation and added a tutorial on how to implement a new algorithm in the SafePO Library.
>
> Related URLs can refer to: https://safe-policy-optimization.readthedocs.io/en/latest/usage/implement.html

---

> ### Author Response · Authors · 2023-08-21
> **Rebuttal by Authors (5/5)**
>
> > **(Relation To Prior Work # 1)** It is not very clear how much of a significant contribution this platform brings over Safety Gym and its equivalents. Primarily it would have been good to shed light on why the contributions discussed in the paper require an entirely new library as opposed to making changes / additions on top of Safety Gym.
>
> Our original motivation in designing Safety-Gymnasium was to address certain usability issues with the existing Safety-Gym. This environment has been gaining traction within the SafeRL community. Building upon this, we crafted a new architecture based on MuJoCo, encompassing all tasks of Safety-Gym, and made it open-source for the community. Presently, the SafePO and Safety-Gymnasium repository on GitHub has garnered nearly 400 stars.
>
> **Notably**, recent endeavors such as the CMU team's development of an Offline Dataset for SafeRL[1] have also employed Safety-Gymnasium as a foundation.
>
> [1] Datasets and Benchmarks for Offline Safe Reinforcement Learning
>
> Moreover, the standout feature of Safety-Gymnasium lies in its incorporation of both multi-agent safety tasks and visual safety tasks. This comprehensive integration positions it as a versatile benchmark environment for safety-oriented reinforcement learning. Throughout the implementation of this framework, we prioritized scalability, encouraging users to engage in secondary development to introduce captivating tasks tailored to SafeRL. Closely tied to our work is Safety Gym, and we have elaborated extensively on the distinctions between Safety-Gymnasium and Safety Gym, as outlined below (see **Compare to Safety Gym** (In the introduction) in the revised version of the paper):
>
> **(1). Physics Engine Upgrade.**
> While Safety Gym used `mujoco-py`, Safety-Gymnasium has been updated to use the newest `MuJoCo` 2.3 and above. This change was needed because `mujoco-py` stopped getting updates after October 18, 2021, and it no longer worked with `MuJoCo` now. This caused issues when installing Safety Gym, requiring us to think about compatibility with older versions, including downgrading Numpy and GCC.
> In contrast, Safety-Gymnasium has moved away from relying on `mujoco-py` and instead went through significant changes to take advantage of the latest features of MuJoCo (which was a big technical challenge). This simplification has made it easier to get started, as now users only need to use one simple command: `pip install safety-gymnasium` to set up the basic Gymnasium-based safety tasks. Additionally, using MuJoCo's new features has enabled us to run Safety-Gymnasium on both Mac and Windows PCs, improving how quickly environments are displayed (around 25% faster), making the visuals more accurate, and enhancing support for visual elements like importing Obj files.
>
> **(2). Diverse Agent and Task Expansion.**
>
>   - Added single-agent: Rececar. Compared with the agent car, it is more real and has a corresponding physical machine [MIT Racecar](https://racecar.mit.edu/). Reviewer w71G highly appreciated the introduction of this agent and suggested that we extend the agent to FormulaOne (Drive-Track) tasks.
>
> - Added single-agent: Ant. The original safety gym agent is mainly considered from the perspective of simplifying the difficulty. It includes three agents that are designed not to fall, so they are more suitable for exploration, but this is too simple. We introduced Ant to a higher degree of freedom which iterates and rolls during the exploration process, which can increase the difficulty of agent exploration and the diversity of data.
>
> > **Safety Gym Origin paper**: It is designed such that a uniform random policy should keep the robot from falling over and generate some travel.
>
> **(3). Extend to Multi-agent Safety Tasks.**
>
> - MaMuJoCo-based, which splits the agent in MuJoCo into multiple joints and controls it through a multi-agent algorithm. This is widely accepted in the MARL community and has been adopted by many famous algorithms, such as HATRPO/PPO[1], MARLlib[2] et al.
>
> - Another type is the Multi-Ant scenario, where Ant can be replaced by other agents such as Point, Car, etc. Multiple agents cooperate to complete some tasks. We have implemented this task in Safety Gymnasium and added it to the document. For details, please refer to: https://www.safety-gymnasium.com/en/latest/environments/safe_multi_agent.html
>
>   [1] Trust Region Policy Optimisation in Multi-Agent Reinforcement Learning
>
>   [2] MARLlib: A Scalable Multi-agent Reinforcement Learning Library
>
> **(4). Extend to Complex and High-dimensional Scenes.**
>
> DexterousHands focuses on the Bimanual Dexterous Manipulation, and the task difficulty is even more difficult (state space dimension 350+, action space dimension 40+). We extend its task to the safe field and choose two typical constraints, Safe Joint and Safe Finger. More details can be referred to: https://www.safety-gymnasium.com/en/latest/environments/safe_multi_agent/freight_franka_close_drawer.html

---

> ### Author Response · Authors · 2023-08-24
> **Hope to get your reply**
>
> Dear Reviewer w71G,
>
> The deadline for review is nearing and we noticed that you have not replied to our rebuttal. We spend a great deal of effort during rebuttal to respond carefully to each of your questions, including the additional experiments, environmental design, and paper revisions. We sincerely hope you to reference both our rebuttal content and our latest revised submission, in the hope that you will consider reevaluating your assessment based upon these updates.
>
> Other references:
>
> Demo Site: https://sites.google.com/view/safety-gymnasium
>
> Safety-Gymnasium Online Documentation: https://www.safety-gymnasium.com
>
> SafePO Online Documentation: https://safe-policy-optimization.readthedocs.io

---

> ### Comment · Area_Chair_stYK · 2023-08-24
> **Check the author's responses**
>
> Hi Reviewer,
>
> Can you please come back and check the author's response and update?
>
> Best,
> AC

---

> ### Author Response · Authors · 2023-08-28
> **Hope to get your feedback**
>
> Dear Reviewer w71G,
>
> The review cycle is almost over, and we noticed that you have not responded to our rebuttal. We spend a lot of effort during the rebuttal process to carefully answer each of your questions, including additional experiments, environment design, and paper revisions. We sincerely hope that you refer to our rebuttal and our latest revised submission and that you consider reassessing your assessment in light of these updates.
>
> And we have actively resolved most of the reviewers' concerns during the rebuttal process, and they have given high recognition to our work, for example, we have resolved the concerns of the reviewer EjZD to a certain extent, and he has improved his own score. At the same time, Reviewer vCTw highly recognized our work and praised our contribution to the SafeRL community, and raised its score from 5 to 8
>
>
> We sincerely look forward to your reply.
>
> Other references:
>
> Demo Site: https://sites.google.com/view/safety-gymnasium
>
> Safety-Gymnasium Online Documentation: https://www.safety-gymnasium.com
>
> SafePO Online Documentation: https://safe-policy-optimization.readthedocs.io

---

### Official Review · Reviewer_2t22 · 2023-07-28
**Safety Gymnasium Review**

**Rating:** 7
**Confidence:** 3
**Correctness:** Seems correct.
**Clarity:** Fairly clear.

**Strengths:**

Availability of dataset environment through Gym and MuJoCo interface and easy-installation.
Comprehensive validation and comparison of 16 state-of-the-art reinforcement learning algorithms.

**Additional Feedback:**

Nothing particular.

**Documentation:**

The link to the website is provided.

**Ethics:**

Not applicable.

**Limitations:**

The key contribution of this paper should be clearly stated.
It would be great if authors include more information in documentation explaining how someone can reproduce results in the paper with the scripts provided.

**Opportunities For Improvement:**

See Limitations.

**Relation To Prior Work:**

The paper discussed and compared with the prior work sufficiently.

**Summary And Contributions:**

This paper proposes a reinforcement learning simulation framework with specific focus on safety. Compared with existing framework, the task scope is expanded to include vision and multi-agent application scenarios. 16 state-of-the-art algorithms are compared based on this framework.

---

> ### Author Response · Authors · 2023-08-21
> **Rebuttal by Authors (1/1)**
>
> Thank you for taking the time to read and review our paper! Please see our responses below and we have started working on the corresponding changes (see uploaded revised version) and will incorporate all of them in the final revision. If you have any further concerns, we would be keen to address them as well.
>
> > **Re: (Limitations #1)** The key contribution of this paper should be clearly stated.
>
> Thanks a lot for your valuable advice. In response to the issue of unclear contributions raised in the first version, we have redefined the key contributions of this paper in the Introduction (highlighted in orange). These revisions will be incorporated into the final revised version.
>
> 1. **Environmental Components.** We provide an extensive range of safety-oriented tasks under the umbrella of Safety-Gymnasium. These tasks encompass single-agent, multi-agent, and vision-based challenges, each with varying constraint sets. Our environments are categorized into two primary types: Gymnasium-based, featuring agents of escalating complexity for algorithm verification and comparison; and Issac-Gym-based, incorporating sophisticated agents that harness the parallel processing power of Issac-gym's GPU. This empowers researchers to explore security reinforcement learning algorithms in complex scenarios. Further details can be found in Section 4.
>
> 2. **Algorithm Components.** We offer the SafePO algorithm library, which comprises a single-file style housing 16 diverse algorithms. These algorithms encompass both single-agent and multi-agent approaches, along with first-order and second-order variants, as well as Lagrangian-based and Projection-based methods. Through meticulous decoupling, each algorithm's code resides in an individual file. A more in-depth exploration of SafePO is presented in Section 5.
>
> 3. **Insights and Analysis.** Combining Safety-Gymnasium and SafePO, we conduct a detailed analysis of existing algorithms. Our analysis encompasses 16 algorithms across 54 distinct environments, covering various scenarios such as single-agent and multi-agent setups with varying constraint complexities. This scrutiny delves into the strengths, constraints, and avenues for enhancement for each algorithm. We provide open access to all metadata, fostering community verification, and encouraging further research and analysis. Further details can be found in Section 6.
>
> > **Re: (Limitations #2)** It would be great if authors include more information in documentation explaining how someone can reproduce results in the paper with the scripts provided.
>
> Thank you for your suggestion. In the new version of the documentation, we have made content additions and provided key scripts to ensure the reproducibility of experiments. The specific updates are as follows:
>
> 1. We have supplied a reproducible Makefile script that users can employ on any machine equipped with CPU/GPU. This facilitates the effortless replication of the experimental outcomes outlined in the paper. More details can be referred to: https://safe-policy-optimization.readthedocs.io/en/latest/usage/make.html
>
> 2. We have conducted a more in-depth analysis of the existing experimental results and incorporated updates in Section 5: Experiments and Analysis of the main paper as well as Section A of the appendix. Moreover, we have utilized the WandB platform to showcase the experimental outcomes, elegantly presenting the algorithmic achievements in the online documentation of SafePO. More details can be referred to: https://safe-policy-optimization.readthedocs.io/en/latest/algorithms/curve.html
>
> 3. All experimental results have been uploaded to `Google Drive`, encompassing data from various environments, algorithms, and different seeds. This mode of data sharing enables the broader academic community to more directly access and leverage the data, whether for result validation or further research endeavors. We not only provide the original training data but also offer a quick visualization tool to validate these results. In the `Python` environment, execute `pip install -r requirements.txt`, then run the scripts `plot_sa.py` and `plot_ma.py` respectively. This will generate training curve graphs in the figure folder.
>
> `Safety Gymnasium & SafePO Data Package`: https://drive.google.com/file/d/1jlfqvcKHRXkEi--xrrSFcl8an9w7YDG0/view?usp=sharing

---

### Author Response · Authors · 2023-08-21
**To all reviewers**

We thank the reviewers (Reviewer-2t22, Reviewer-w71G, Reviewer-vCTw, Review-7227, Reviewer-EjZD) for their valuable feedback. We addressed all the reviewer comments below and will incorporate them into the revision. If this rebuttal addresses the concerns, we earnestly and kindly ask the reviewers to consider raising the rating and supporting us for acceptance.
The revised version primarily includes the following significant updates (with the modified sections marked in **orange**):

  1. In the introduction section, we added a detailed discussion on the improvement and expansion of Safety Gymnasium compared to Safety Gym. (Received attention from Reviewer w71G, vCTw, EjZD)

  2. We list the three key contributions of this work and update them in the introduction. --> (Received attention from Reviewer 2t22, EjZD)

  3. We added a comparison between SafePO and other open-source implementations, focusing on the comparison of reward and cost under the same settings, and updated it in section 5. --> (Received attention from Reviewer w71G, EjZD)

  4. Based on the original tasks, we have added richer visual tasks, such as racecar-based tracking tasks, multi-agent Ant tasks, and FreightFranka-related tasks on Isaac Gym (see [Online Documentation](https://www.safety-gymnasium.com/en/latest/environments/safe_isaac_gym/freight_franka_close_drawer.html)). --> (Received attention from Reviewer w71G, EjZD)

  5. We have added a description and comparison of visual information input about RGB and RGB-D (see Figure 5), as well as the description of vision-only tasks in the Appendix B.5 and Online Documentation. --> (Received attention from Reviewer w71G, EjZD)

  6. Added a description of SafePO, described the effectiveness of SafePO from the four perspectives of Correctness, Extensibility, Logging and Visualization, and Documentation, and added the figure of SafePO's architecture (see Figure 7).--> (Received attention from Reviewer EjZD)

  7. In Section 6 Experiments and Analysis, we added more experimental results and experimental analysis and added very detailed experimental performance tables (see Table 2 and Table3), as well as the performance analysis figure (see Figure 8), and we uploaded all the data to [Google Drive](https://drive.google.com/file/d/1jlfqvcKHRXkEi--xrrSFcl8an9w7YDG0/view?usp=sharing) for community use and verification. --> (Received attention from Reviewer vCTw, 7227, EjZD)

  8. Newly added Limitations and Future Work to further analyze the limitations and future work of this paper. --> (Received attention from Reviewer 2t22, vCTw, 7227)

---

### Comment · Area_Chair_stYK · 2023-08-29
**Possible dual submission and the overlap among SafePO, omnisafe, and Safety Gymnasium**

Dear Authors,

Since this submission is single-blinded, AC and all the reviewers can check the submission and associated libraries and related resources as due diligence. AC found several papers and libraries from the authors have similar keywords like infrastructure, benchmark, and library or so all related to SafeRL, specifically, SafetyPO (https://gengyiran.github.io/pdf/safepo.pdf), ominisafe (https://arxiv.org/abs/2305.09304), and this one submission.

In particular, AC found this SafePO paper (https://gengyiran.github.io/pdf/safepo.pdf) is in a submission template format as well as evidenced by the first author's website (https://gengyiran.github.io/), saying it is Under Review 2023. If true, it is in a concurrent submission with this Safety Gymnasium submission. One of the key contributions of this Safety-Gymnasium submission is SafePO (in abstract: "Additionally, we offer a library of algorithms named Safe Policy Optimization (SafePO)".

So we need to figure out what's going on here. If there are concurrent submissions or even possible dual submissions, it violates the code of the NeurIPS submission, and this submission should be desk rejected. Authors are highly suggested to explain it and claim any concurent submissions. Also please do not delete all the above evidence as they are already screen-shotted by the AC.

The AC will also urge all the reviewers to consider this case of overlaps in the final recommendation. Please let me know what you think.

AC

---

> ### Author Response · Authors · 2023-08-29
> **A responsible response about dual submission by authors**
>
> Dear PC, SAC, AC, and Reviewers,
>
> We **strictly adhere to scientific principles and have not engaged in any form of dual submission.**
>
> **Last year, SafePO was submitted independently to IJCAI, as indicated in the content found at https://gengyiran.github.io/pdf/safepo.pdf (upload at [2023.04.05](https://github.com/GengYiran/GengYiran.github.io/commit/2b7d3da9909e63d312c921c694c55ecfde48559e)), using the IJCAI template**. However, due to the fact that IJCAI does not accept this type of benchmark (**Not a gift for IJCAI**), our paper was rejected. **Since then, SafePO has not been submitted again.** Additionally, during the period from February (2023.02) to the NeurIPS DB submission deadline (2023.06.14), SafePO has been updated several times. For details, please refer to [GitHub Commit History](https://github.com/PKU-Alignment/Safe-Policy-Optimization/commits/main).
>
> Simultaneously, up until the NeurIPS DB submission, we think that **SafePO's scientific research contribution is not enough to support a paper.** **Therefore, we integrated SafePO into Safety-Gymnasium (as one of the scientific contributions of this paper).**
>
> OmniSafe is also developed by our team. It is a baseline collection, that contains many types of safe reinforcement learning algorithms (such as Model-free, Model-based, and Offline) because it involves many algorithms in different domains. Considering compatibility, we have done a lot of layer encapsulation involving some high-level syntax of Python. It does not involve any environment itself, and it supports mainstream environments with constraints, such as [MetaDrive](https://github.com/PKU-Alignment/omnisafe/pull/263) and so on. However, after we open-sourced, we received more feedback that this high-level encapsulation is not suitable for beginners. **Therefore, we position SafePO as a single-file style algorithm library, which does not involve too many high-level packages and is more suitable for beginners to use.**
>
> In summary, the core contribution of this work ("Safety Gymnasium: A Unified Safe Reinforcement Learning Benchmark") revolves around the Safety-Gymnasium environment component itself. On the other hand, SafePO represents our further expansion. Combined these two components can effectively promote SafeRL research.
>
> It is the basic requirement of every scientific researcher to strictly adhere to every scientific submission standard. ****The above is our responsible statement, please review by PC, SAC, AC, and Reviewers.****

---

> > ### Comment · Area_Chair_stYK · 2023-08-30
> > **Thanks.**
> >
> > Dear Authors,
> >
> > Thanks for the explanation. I will take that into consideration and work things out.
> >
> > AC

---

> > > ### Author Response · Authors · 2023-08-30
> > > **re-confirmation**
> > >
> > > Dear AC,
> > >
> > > Thanks for your prompt response. We would like to re-confirm and re-emphasize that our contribution of `Safety-Gymnasium` is unique and it has ***no overlap with any existing papers*** that are currently under review. The version of SafePO mentioned by AC was rejected by IJCAI, here we further provide the evidence of the  [CMT screenshot](https://github-production-user-asset-6210df.s3.amazonaws.com/73586554/264403971-b9b9fc64-ea38-471e-bba3-5dbf0b701bbe.png)). In our current submission `Safety-Gymnasium`, significant updates on top of the SafePO have been made (including a list of versatile safe RL tasks, single file-style PO algorithms collection, detailed documentation, comprehensive experimental with insights) , which also have been credited by the reviewers.
> > >
> > > We firmly believe that AC would make a fair judgemental decision. Regardless of the final result (acceptance or rejection), we will choose to disclose the entire rebuttal process to show our ***integrity and innocence***, which we always value the most.

---

> ### Comment · Reviewer_vCTw · 2023-08-30
>
> Dear AC,
>
> I took a closer look at the two libraries: SafePO and OmniSafe, since I raised similar questions in my comment thread. After delving deep into these two repositories afterwards, I am confident to say that these two projects are quite different in terms of project scope, targeted users, and codebase design. SafePO's style is more like [CleanRL](https://github.com/vwxyzjn/cleanrl), featuring a straightforward architecture that is friendly for beginners and new researchers in this area. In contrast, OmniSafe uses more advanced code design and syntax for encapsulation, which is more like a production-level codebase and is thus more suitable for senior researchers with strong background in this field. Omnisafe also provides offline safe learning algorithms, while SafePO includes multi-agent safe learning implementations, which I highly appreciate. This differences between SafePO and OmniSafe are similar to the distinction between [Tianshou](https://github.com/thu-ml/tianshou) and [CleanRL](https://github.com/vwxyzjn/cleanrl), from my point of view.
>
> In terms of the dual submission issue, I think the evidence provided by the authors is sufficient and convincing to me, as I also checked the IJCAI official website and found that there is indeed no overlap time period to make dual submission possible. Finally, from my perspective, the open-sourced projects presented in this work are significant to the safe RL community, with the potential to accelerate researches in this important area, so I still lean towards acceptance of this paper.
>
> Best,
>
> Reviewer vCTw

---

### Decision · Program_Chairs · 2023-09-22

**Decision:**

Accept (Poster)

**Comment:**

This submission proposes a safe RL benchmark called Safety Gymnasium. A range of safety-critical autonomy tasks in single and multi-agent scenarios are encompassed. A library of algorithms called SafePO with 16 SOTA safe RL algorithms is also provided as the baselines. The proposed benchmark can facilitate the evaluation and comparison of safe RL algorithms. Detailed documentation and Github repo are provided. The Github repo already has a couple of hundred stars, demonstrating the usage of the proposed benchmark to the community. AC has checked all of them.

After the discussion between reviewers and the authors, the submission received scores of 8, 7, 6, 6, 5, so overall very positive evaluation. AC did some homework in the discussion session to examine the library and raised an issue about duo submission. After clarification from the authors, the issue is resolved. But authors are still suggested to be mindful of the submission policy and what can be put on a personal webpage, to avoid confusion.

Overall, AC appreciates the contribution of the library, the detailed documentation, and the timely responses of the authors in the discussions. AC believes this is a solid contribution to the community. Thus, an acceptance is recommended.